# Molecular and physical characteristics of aerosol at a remote free troposphere site: Implications for atmospheric aging

Simeon K. Schum[1], Bo Zhang[2,a], Katja Dzepina[1,b], Paulo Fialho[3], Claudio Mazzoleni[2,4] and Lynn R. Mazzoleni[1,2]

[1]Department of Chemistry, Michigan Technological University, Houghton, MI, USA
[2]Atmospheric Sciences Program, Michigan Technological University, Houghton, MI, USA
[3]Institute for Volcanology and Risk Assessment – IVAR, University of the Azores, Angra do Heroísmo, Portugal
[4]Department of Physics, Michigan Technological University, Houghton, MI, USA
[a]now at the National Institute of Aerospace, Hampton, VA, USA
[b]now at the Department of Biotechnology, University of Rijeka, Croatia

*Correspondence to:* Lynn R. Mazzoleni (lrmazzol@mtu.edu)

**Abstract.** Aerosol properties are transformed by atmospheric processes during long-range transport and play a key role in the Earth's radiative balance. To understand the molecular and physical characteristics of free tropospheric aerosol, we studied samples collected at the Pico Mountain Observatory in the North Atlantic. The observatory is located in the marine free troposphere at 2225 m above sea level, on Pico Island in the Azores archipelago. The site is ideal for the study of long-range transported free tropospheric aerosol with minimal local influence. Three aerosol samples with elevated organic carbon concentrations were selected for detailed analysis. FLEXPART retroplumes indicated that two of the samples were influenced by North American wildfire emissions transported in the free troposphere and one by North American outflow mainly transported within the marine boundary layer. Ultrahigh resolution Fourier transform ion cyclotron resonance mass spectrometry was used to determine the detailed molecular composition of the samples. Thousands of molecular formulas were assigned to each of the individual samples. On average ~60 % of the molecular formulas contained only carbon, hydrogen, and oxygen atoms (CHO), ~30 % contained nitrogen (CHNO), and ~10 % contained sulfur (CHOS). The molecular formula compositions of the two wildfire influenced aerosol samples transported mainly in the free troposphere had relatively low average O/C ratios ($0.48 \pm 0.13$ and $0.45 \pm 0.11$) despite the 7 - 10 days of transport time according to FLEXPART. In contrast, the molecular composition of North American outflow transported mainly in the boundary layer had a higher average O/C ratio ($0.57 \pm 0.17$) with 3 days of transport time. To better understand the difference between free tropospheric transport and boundary layer transport, the meteorological conditions along the FLEXPART simulated transport pathways were extracted from the Global Forecast System analysis for the model grids. We used the extracted meteorological conditions and the observed molecular chemistry to predict the relative humidity dependent glass transition temperatures ($T_g$) of the aerosol components. Comparisons of the $T_g$ to the ambient temperature indicated that a majority of the organic aerosol components transported in the free troposphere were more viscous and therefore less susceptible to oxidation than the organic aerosol components transported in the boundary layer. Although the number of observations is limited, the results suggest that biomass burning organic aerosol injected into the free troposphere are more persistent than organic aerosol in the boundary layer having broader implications for aerosol aging.

## 1 Introduction

Atmospheric organic aerosol composition and mass concentrations are transformed by atmospheric processes including oxidization (Dunlea et al., 2009; Jimenez et al., 2009; Kroll et al., 2011), cloud processing (Ervens et al., 2008, 2011; Zhao et al., 2013), and wet or dry deposition (Pöschl, 2005). Oxidation of organic aerosol impacts its lifetime, cloud droplet and ice nucleation activity (Massoli et al., 2010; Lambe et al., 2011; China et al., 2017), aerosol morphology, and optical properties

(Pöschl, 2005; China et al., 2015; Laskin et al., 2015 and references therein). As such, the chemistry of atmospheric aerosol oxidation has received much attention (George and Abbatt, 2010; Lee et al., 2011; Kroll et al., 2011). Jimenez et al. (2009) studied the oxidation of anthropogenic organic aerosol emitted from Mexico City as it was transported downwind. They used an aerosol mass spectrometer (AMS) instrument on board an aircraft to measure the magnitude of the $m/z$ 44 fragment as a proxy for the oxidation of organic aerosol. After 6 hours and 63 km of transport, a noticeable increase in the overall chemical oxidation was observed. Rapid oxidation was also observed in studies of biomass burning organic aerosol in Africa (Capes et al., 2008; Vakkari et al., 2014), over the Mediterranean Sea (Bougiatioti et al., 2014), and Hyytiälä, Finland (Corrigan et al., 2013; Vogel et al., 2013). Other studies focused on the oxidation of molecular tracers such as levoglucosan have shown that they can be degraded rapidly after emission, depending on the atmospheric conditions (Lai et al., 2014; Slade et al., 2014; Arrangio et al., 2015; Bertrand et al., 2018). These studies all demonstrate the importance of oxidation to the aging of organic aerosol and provide motivation for studies of long-range transported organic aerosol.

A study of predominately Asian anthropogenic aerosol transported in the free troposphere over the Pacific Ocean found that the oxidation, inferred by the average oxygen to carbon ratio (O/C), continued to increase over the course of roughly a week (Dunlea et al., 2009). In a study of biomass burning aerosol transported in the free troposphere across the North Atlantic Ocean, Dzepina et al. (2015) observed a relatively low O/C ratio ($0.46 \pm 0.13$), considering the aerosol transport time of more than 10 days (Aiken et al., 2008). Dzepina et al. (2015) hypothesized that cloud processing and other oxidative processes led to the formation and subsequent removal of oxidized species, leaving behind the more persistent aerosol species. Other recent studies of long-range transported brown carbon (BrC) from biomass burning in the boundary layer found that the aging of aerosol led to a near complete depletion of BrC within 24 hours (Forrister et al., 2015; Laing et al., 2016). The remaining BrC was found to lead to a 6 % increase of BrC over background levels and may represent the ubiquitous BrC present in the atmosphere far from the source (Forrister et al., 2015). This leftover BrC aerosol could impact large areas globally because much of it is located within the free troposphere and above clouds, due to the typically elevated injection heights of aerosol over wildfires (Val Martin et al., 2008a). These studies indicate that free tropospheric aerosol chemistry is particularly important because this aerosol can have a longer atmospheric lifetime than boundary layer aerosol (Laing et al., 2016) allowing it to be transported over greater distances.

Studies of transported biomass burning aerosol are typically performed using instrumentation either onboard aircrafts (Capes et al., 2008) or located at low altitude (Bougiatioti et al., 2014; Vakkari et al., 2014) and continental mountain (Laing et al., 2016) sites. Aircraft measurements have the advantage of sampling aerosol over wide spatial and altitudinal ranges, but they are limited to short time periods, typically of a few days to a week (Capes et al., 2008; Dunlea et al., 2009). Ground sites are less constricted by time, but low altitude sites are less often affected by pyro-convective wildfire plumes due to the high injection heights of wildfires (Val Martin et al., 2008a). Continental mountain sites typically have seasonally limited access to the free troposphere, because high summer temperatures can lead to convection of the planetary boundary layer (Collaud Coen et al., 2011). Thus, many of the continental mountain sites have long-term access to the free troposphere in the winter, but not in the summer when most wildfire activity occurs. The Pico Mountain Observatory (PMO, see the Supplement for additional information), is located 2225 m above sea level (a.s.l) on the caldera summit of Pico Mountain, on Pico Island in the Azores archipelago in the North Atlantic. The marine boundary layer in the region has been measured and is estimated to range from 500 to 2000 m a.s.l in the summer months (Kleissl et al., 2007; Remillard et al., 2012; Zhang et al., 2017), well below the observatory. This permits access to free tropospheric long-range transported aerosol during the wildfire season. This, in conjunction with negligible local emission sources, makes PMO an ideal site for the study of long-range transported free tropospheric aerosol.

As described by Zhang et al. (2017), the Azores-Bermuda anticyclone causes persistent downward mixing from the upper free troposphere and lower stratosphere, and is the dominant meteorological pattern in this region, and strengthens in the summer. The FLEXible PARTicle dispersion model (FLEXPART) retroplumes discussed in Zhang et al. (2017) show that this site is most commonly impacted by North American outflow (30-40%). In the summer months (June – August), 15% of the intercepted air masses have North American anthropogenic influence, and 7.3% have wildfire influence (Zhang et al., 2017). These factors make PMO an excellent site for the study of North American outflow (Val Martin et al., 2008a). In many of the previous studies at PMO, investigators focused on the North American outflows of $NO_x$, $NO_y$, $CH_4$, non-methane hydrocarbons, and $O_3$ gases (Val Martin et al., 2006; Pfister et al., 2006; Val Martin et al., 2008a; Val Martin et al., 2008b; Helmig et al., 2015) and the physical characteristics of black carbon and mineral dust aerosol and their ice nucleation activity (Fialho et al., 2005; China et al., 2015; China et al., 2017). So far, only Dzepina et al. (2015) has previously looked at the aerosol chemical and molecular characteristics of the organic aerosol collected at the site.

Recent environmental and laboratory studies have shown that under low temperature and low relative humidity (conditions common in the free troposphere), aerosol can be in a solid glassy phase (Zobrist et al., 2008; Virtanen et al., 2010). This observation has been hypothesized to lead to longer atmospheric lifetimes for organic species that are otherwise susceptible to degradation through oxidative processes. As an example, polycyclic aromatic hydrocarbons (PAH) such as benzo[a]pyrene were observed to have a much higher ambient concentration than what could be explained by model simulations without considering the aerosol phase state (Shrivastava et al., 2017). Recently, PAH have been shown to enhance the formation of a viscous phase state in laboratory generated secondary organic aerosol (SOA) (Zelenyuk et al., 2017). Since PAH are common products of biomass burning and anthropogenic emissions, the viscosity could be enhanced in ambient samples as well, leading to a greater likelihood of the occurrence of solid phase aerosol. The solid phase can increase the resistance of aerosol to photodegradation (Lignell et al., 2014; Hinks et al., 2015) and water diffusivity (Berkemeier et al., 2014), which may lead to lower rates of oxidation. Shiraiwa and colleagues developed a set of equations to predict the dry glass transition temperature based on the mass and O/C ratio of organic aerosol components (Shiraiwa et al., 2017; DeRieux et al., 2018). Thus, the phase state of molecular species with respect to ambient conditions can be predicted using the Gordon-Taylor equation (Shiraiwa et al., 2017; DeRieux et al., 2018). Additionally, estimation methods to determine the volatility of organic aerosol were reported by Donahue et al. (2011) and Li et al. (2016). Both the phase state and volatility are important in understanding the processes that affect aerosol during transport and aging. The low oxidation observed by Dzepina et al. (2015) was attributed to the dominance of persistent aerosol that resisted removal mechanisms, however it is possible that the phase state of the aerosol during transport played a significant role. The increased resistance to photodegradation (Lignell et al., 2014; Hinks et al., 2015) and water diffusivity (Berkemeier et al., 2014) of solid phase organic aerosol provide a basis for this hypothesis.

Ultrahigh resolution mass spectrometry (MS) is a necessary tool for determining the molecular characteristics of complex mixtures such as organic aerosol. It has been used to analyze dissolved organic matter (Kido-Soule et al., 2010; Herzsprung et al., 2014), cloud water (Zhao et al., 2013; Cook et al., 2017), fog water (Mazzoleni et al., 2010), sea spray (Schmitt-Kopplin et al., 2012), and organic aerosol (Walser et al., 2007; Mazzoleni et al., 2012; O'Brien et al., 2013; Wozniak et al., 2014; Dzepina et al., 2015). Ultrahigh resolution MS techniques are typically paired with electrospray ionization (ESI) because it is a soft ionization technique with little to no fragmentation of the molecular species being analyzed. Negative mode ESI is sensitive to molecules with acidic functional groups, which is ideal for the analysis of long-range transported organic aerosol due to its generally acidic nature (Bougiatioti et al., 2016). The ultrahigh mass resolution available from Fourier transform ion cyclotron resonance mass spectrometry (FT-ICR MS) and the high field Orbitrap Elite MS instruments (R = 240,000) separates sulfur containing species from carbon, hydrogen, and oxygen containing species (Schmitt-Kopplin et al., 2010), which is

important because sulfur containing species are common in atmospheric aerosol (Schmitt-Kopplin et al., 2010; Mazzoleni et al., 2012; Dzepina et al., 2015).

The observations of Dzepina et al. (2015) raised interesting questions regarding the nature of long-range transported free tropospheric aerosol. To further elucidate the detailed molecular characteristics of free tropospheric aerosol, we analyzed three pollution events using ultrahigh resolution FT-ICR MS. We observed key molecular differences pertaining to the extent of oxidation likely related to the combination of transport pathways and their apparent emission sources. In this paper, we present the detailed molecular characteristics of organic aerosol collected at PMO and use the aerosol chemical composition, FLEXPART model simulations, and physical property estimates to interpret those characteristics and infer implications for long-range transported organic aerosol.

## 2 Methods

### 2.1 Sample collection

PM$_{2.5}$ samples were collected at PMO on 8.5 x 10 in. quartz fiber filter using high volume air samplers (EcoTech HiVol 3000, Warren, RI, USA) operated at an average volumetric flow rate of 84 m$^3$ hr$^{-1}$ for 24 h. Prior to sampling, the filters were wrapped in clean, heavy-duty aluminum foil and baked at 500 $^0$C for ~ 8 hours to remove organic artifacts associated with the filters. Afterward, they were placed in antistatic sealable bags until deployment. We deployed four air samplers at the site, each was set up with a filter simultaneously and programmed to start one day after another, allowing for continuous sample collection for up to four consecutive days. This procedure was used to maximize the number of filters collected. Daily visits and maintenance were prohibited by the time consuming and strenuous hike necessary to reach the site. The sampled filters were removed and returned to the same aluminum wrapper and bag. The samples were then brought down the mountain and stored in a freezer until cold transport back to Michigan Tech where they were stored in a freezer until analysis. Three samples, collected in consecutive years at PMO, on 27-28 June 2013, 5-6 July 2014, and 20-21 June 2015 were analyzed in this study. The sampling time for all samples was 24 hours; on 27-28 June the sampling began at 19:00, on 5-6 July and on 20-21 June the sampling began at 15:00, all local times.

### 2.2 Chemical analyses

Organic carbon and elemental carbon (OC/EC) measurements were performed using an OC/EC analyzer (Model 4, Sunset Laboratory Inc. Tigard, OR, USA) following the NIOSH protocol. Major anions and cations were analyzed using ion chromatography. Anion analysis was performed using a Dionex ICS-2100 instrument (Thermo Scientific) with an AS-17-C analytical and guard column set (Thermo Scientific) using a KOH generator for gradient elution. Cation analysis was performed using a Dionex ICS-1100 instrument with CS-12A analytical and guard column set (Thermo Scientific) and an isocratic 20 mM methanesulfonic acid eluent. The instruments were operated in parallel using split flow from the autosampler. Additional details can be found in the Supplement.

### 2.3 Ultrahigh resolution FT-ICR mass spectrometry analysis

The samples for FT-ICR-MS analysis were selected based on the organic carbon concentration. Selected samples typically had more than 1000 µg of organic carbon per quartz filter. Sample preparation was described in detail in previous studies from our group (Mazzoleni et al., 2010, 2012; Zhao et al., 2013; Dzepina et al., 2015). Briefly, one quarter of the quartz filter was cut into strips, placed in a pre-washed and baked 40 mL glass vial, and then extracted using ultrasonic agitation in Optima LC/MS grade deionized water (Fisher Scientific, Waltham, MA, USA) for 30 minutes. The extract was then filtered using a pre-baked quartz filter syringe to remove undissolved material and quartz filter fragments. The sample filter was then sonicated again in 10 mL of Optima LC/MS grade deionized water for 30 minutes, filtered, and then added to the original 30 mL of filtrate

yielding a total of 40 mL. Ice packs were used during the sonication to ensure the water temperature stayed below 25 °C. The water-soluble organic carbon (WSOC) compounds were then isolated using Strata-X (Phenomenex, Torrance, CA, USA) reversed phase solid phase extraction (SPE) cartridges to remove inorganic salts that can adduct with organic compounds during electrospray ionization. During the reversed phase SPE, losses of highly water soluble, low molecular weight (MW) and hydrophobic, high MW organic compounds are expected. Thus, the resulting WSOC is the SPE-recovered fraction. The cartridges were pre-conditioned with acetonitrile and LC/MS grade water before the 40 mL filtrate was applied to the cartridges at a rate of ~ 1 mL/min. The cartridges were eluted with 2 mL of an aqueous acetonitrile solution (90/10 acetonitrile/water by volume) and stored in the freezer until analysis. The procedural loss of ionic low MW compounds such as oxalate can lead to an underprediction of the organic aerosol O/C and overprediction of the average glass transition temperatures ($T_g$). To investigate this, we used the concentrations of the prominent organic anions measured with ion chromatography to estimate the abundance of these compound relative to the compounds detected by FT-ICR MS. The low MW organic anion corrected average O/C values correlated with the trends of the original O/C values, however the significance of impacts varies with the measured analyte concentrations and the assumptions associated with the uncertain mass fraction of the molecular formula composition (Table SM4). When low MW organic anions were included in the estimated average dry $T_g$ values, they dropped by ≤ 2.5 %, which was deemed relatively insignificant (Table SM5).

Ultrahigh resolution mass spectrometric analysis was done using FT-ICR MS with ESI at the Woods Hole Oceanographic Institution (Thermo Scientific LTQ Ultra). The samples were analyzed using direct infusion ESI in the negative ion mode. Negative polarity is effective for the deprotonation of polar organic molecules (Mazzoleni et al., 2010), which are expected to dominate the organic aerosol mass fraction and were the focus of this study. The spray voltage ranged from 3.15 to 3.40 kV depending on the ionization stability with a sample flow rate of 4 to 5 µL/min. We used a scan range of $m/z$ 100 – 1000 with a mass resolving power of 400,000 (defined at $m/z$ 400) for all samples. The samples were run in duplicate and 200 transient scans were collected. The transients were co-added for each replicate run using the Midas Co-Add tool and molecular formula assignments were made using Composer software (Sierra Analytics), as described in previous studies (Mazzoleni et al., 2012; Dzepina et al., 2015). The resulting molecular formula assignments underwent additional quality assurance (QA) data filtering to remove chemically unreasonable formulas with respect to O/C, hydrogen to carbon ratio (H/C), double bond equivalent (DBE), and absolute PPM error as described in the Supplemental Information of Putman et al. (2012). Molecular formulas in common with the instrument blanks with signal intensity ratios < 3 were removed; meanwhile analytes in common with the field blanks with signal intensity ratios < 3 were flagged. Specifically, two formulas ($C_{17}H_{34}O_4$ and $C_{19}H_{38}O_4$) observed in PMO-1 could not be classified as pertaining only to the field blank and so they were not removed. Further discussion about the blank subtraction is provided in the Supplement. To produce the final data set for each sample, the replicates were aligned and only the molecular formulas found in both replicates after QA were retained.

## 2.4 FLEXPART numerical simulations

FLEXPART was used to determine the sources, ages, and transport pathways of the aerosol samples collected at PMO. FLEXPART backward simulations (also called retroplumes) were driven by meteorology fields from the Global Forecast System (GFS) and its Final Analysis with 3-hour temporal resolution, 1° horizontal resolution, and 26 vertical levels. The output was saved in a grid with a horizontal resolution of 1° latitude by 1° longitude, and eleven vertical levels from the surface to 15,000 m a.s.l. For each simulation, 80 thousand air parcels were released from the receptor and transported backwards for 20 days to calculate a source-receptor relationship (in units of s kg⁻¹, Seibert and Frank, 2004). FLEXPART retroplumes were then multiplied with CO emission inventories (kg s⁻¹) from the Emissions Database for Global Atmospheric Research (EDGAR version 3.2 (Olivier and Berdowski, 2001)) and the Global Fire Assimilation System (Kaiser et al., 2012) to estimate the influence from anthropogenic and wildfire sources, respectively. The FLEXPART CO tracer calculated with this approach

indicates the relative contributions from anthropogenic and biomass burning emissions. Since CO chemistry and dry deposition are not considered in the FLEXPART setup, the absolute FLEXPART CO value does not reproduce the actual CO concentrations at Pico. FLEXPART does not consider the background CO accumulated in the atmosphere. The difference between FLEXPART CO and the actual CO largely depends on these factors. In previous applications of this approach, FLEXPART CO was able to estimate the episodes of CO enhancement due to transport of emissions (e.g., Brown et al., 2009; Stohl et al., 2007; Warneke et al., 2009). This approach has been used in several PMO studies and successfully captured elevated CO periods (e.g., Dzepina et al., 2015; Zhang et al., 2014; 2017) and it is used here to assist in the interpretation of the chemical composition in this work.

In addition to the typical FLEXPART simulations done for the site (e.g., retroplume, CO source apportionment), we extracted the ambient temperature and relative humidity (RH) from the GFS analysis data for model grids along the FLEXPART simulated transport pathways. These parameters were then used to estimate the glass transition temperatures ($T_g$) of the organic aerosol components during transport, based on its molecular composition from ultrahigh resolution MS, using estimation methods recently developed by Shiraiwa et al. (2017) and extended to higher masses by DeRieux et al. (2018). DeRieux et al. (2018) reported an uncertainty of $\pm$ 21 K for the prediction of any single compound, but the uncertainty is expected to decrease when a mixture of compounds is considered. Nonetheless, we assumed an uncertainty range of $\pm$ 21 K on $T_g$ and found that it did not significantly change the $T_g$ trends presented in Section 3.5. Further discussion on the uncertainty of $T_g$ is provided in the Supplement. The distributions of the estimated organic aerosol component $T_g$ values provides new insight for the interpretation of long-range transported aerosol.

## 3 Results and discussion

### 3.1 Overview of the aerosol chemistry: OC/EC and IC

In this study, we present the detailed composition of three individual samples collected for 24 hours on 27-28 June 2013, 5-6 July 2014, and 20-21 June 2015 at the PMO. These samples, referred to as PMO-1, PMO-2 and PMO-3 hereafter, were selected after analysis of organic and elemental carbon (OC/EC) were performed for all 127 aerosol samples collected in this study. The three selected samples all had elevated organic carbon (OC) concentrations (Table 1) representing the capture of a pollution plume. After blank subtraction, the median OC of the samples collected over the summers of 2013-2015 was 0.16 $\pm$ 0.018 µg/m$^3$. The minimum OC level measured was below the average blank concentration and the maximum was 2.07 $\pm$ 0.017 µg/m$^3$ (PMO-1). The most abundant anions and cations in these samples are also shown in Table 1. The presence of these ions is consistent with the results of other studies (Yu et al., 2005; Aggarwal and Kawamura, 2009). Further discussion of the bulk chemical trends will be presented in a future manuscript.

The concentrations of common anions and cations can offer important insight regarding cloud processing and emission sources (Table 1). Specifically, the elevated level of sulfate observed in the PMO-2 sample can be an indicator of anthropogenic influence, cloud processing, or marine influence (Yu et al., 2005). We also observed elevated oxalate concentrations in PMO-1 and PMO-2. Oxalate is known to co-vary with sulfate concentrations in the atmosphere when they are formed by aerosol-cloud processing (Yu et al., 2005; Sorooshian et al., 2007). Thus, the oxalate to sulfate ratio can be an indication of cloud processing (Sorooshian et al., 2007); in general, a higher ratio is the result of increased cloud processing. As described in Sorooshian et al. (2007), the oxalate concentrations increase with cloud processing because there is more time for it to be produced, leading to an increased ratio. PMO-1 had the highest oxalate to sulfate ratio (0.278), followed by PMO-3 (0.124), and PMO-2 (0.084). The observed oxalate to sulfate ratios for these samples are all much higher than what was reported in Sorooshian et al. (2007) suggesting other factors may have impacted the ion concentrations. Specifically, an enrichment of oxalate from biomass combustion plumes (Cao et al., 2017) likely contributed to the observed concentrations of these ions in

PMO-1 and PMO-3. The bulk concentration of oxalate in PMO-2 is similar to PMO-1, but the sulfate in PMO-2 is much higher, leading to a low oxalate to sulfate ratio. Based on FLEXPART simulations it is likely that PMO-2 underwent aqueous phase processing (see Sect. 3.5), but the high concentration of sulfate from possible anthropogenic and marine sources appears to have obscured the oxalate-sulfate relationship (Yu et al., 2005; Sorooshian et al., 2007).

Despite inconsistencies in the replicate potassium measurements for PMO-1, elevated potassium levels were observed, indicating contributions from biomass combustion (Duan et al., 2004). PMO-3 had slightly elevated potassium relative to PMO-2, but not as high as PMO-1. Chloride was also present in PMO-1 and PMO-3, which has been shown in some studies to be a minor product of biomass burning, depending on the fuel burned (Levin et al., 2010; Liu et al., 2017).

The nitrate levels in PMO-2 were significantly lower than what was observed in PMO-1 and PMO-3, which is consistent with the observation that the marine boundary layer promotes the rapid removal of $HNO_3$ (Val Martin et al., 2008b). This is also consistent with removal due to cloud scavenging (Dunlea et al., 2009). The elevated nitrate in PMO-1 and PMO-3 is consistent with the observation of elevated $NO_y$ and $NO_x$ in the plumes of wildfire emissions made in previous studies at PMO (Val Martin et al., 2008a) and a lack of recent cloud scavenging (Dunlea et al., 2009).

Despite the low altitude transport, the major ion concentrations in PMO-2 do not strongly support a major influence from marine sources (Quinn et al., 2015; Kirpes et al., 2017). However, the increased concentration of methane sulfonic acid (MSA) in PMO-2 relative to PMO-1 and PMO-3 suggests some degree of marine influence. To estimate this, we used the non-background subtracted sodium concentration as an upper limit to estimate sea salt sulfate according the method described in Chow et al. (2015), this led to a maximum sea salt sulfate contribution of 25 %. The influence of marine sources supports boundary layer transport. However, the results indicate that marine aerosol is not likely a major component of PMO-2, perhaps because the rate of PMO-2 transport was very fast.

## 3.2 FLEXPART retroplume simulation results

Representative FLEXPART retroplumes for the three samples are shown in Fig. 1, additional time periods are in the Supplement (Figs. S1-S3). PMO-1 was largely influenced by North American outflow transported relatively high in the free troposphere. Based on the FLEXPART carbon monoxide (CO) modeling (Fig. S4), PMO-1 was impacted by wildfire emissions from Canada. The transport time for PMO-1 air masses from North America to PMO was about 7 days. The free tropospheric transport is likely due to the high injection heights (Val Martin et al., 2008a; 2010) of organic aerosol from wildfire events in northwestern Quebec (See Fig. S4, S5). Similar events at PMO have been identified previously by (Val Martin et al., 2006; 2008a). The air masses intercepted during PMO-3 were North American outflows that traveled in the lower free troposphere across the Northern Atlantic Ocean to Western Europe before circling back to PMO. The transport time for the PMO-3 air masses from North America to PMO was roughly 10 days. After a northward transport to Western Europe in the jet stream during the first 4-5 days, the simulated plume turned to the south and west, arriving at PMO from Europe in about 2-4 days. This air mass was most likely influenced by wildfire emissions in western and central Canada (U.S. Air Quality, Smog Blog. alg.umbc.edu). Similar to PMO-1, FLEXPART CO source apportionment (Fig. S4) suggests this sample was influenced by fire, although considering the OC concentration and transport time, it was much more diluted than what was observed in PMO-1. In contrast, the PMO-2 air masses traveled relatively low ($\leq 2$ km) over the "Rust Belt" (Illinois, Indiana, Michigan, Ohio, Pennsylvania, and New York) of the United States and stayed at approximately the same altitude until it reached the observatory 2-4 days later. This transport pattern suggests that the aerosol was predominantly transported in the boundary layer on its way to the PMO and was primarily influenced by a mixture of continental U.S. anthropogenic and biogenic emissions. This was supported by the FLEXPART CO simulations as well (Fig. S4). The height of the boundary layer over the continent

generally ranges from 500-2500 m and is strongly affected by diurnal cycles, seasonal effects, and topography (Liu and Liang, 2010); overall, the continental boundary layer height generally increasing during the day and during the summer months. This suggests that PMO-2 was within the boundary layer over the United States.


### 3.3 Molecular formula oxidation metrics: O/C and $OS_C$

In this section, we describe the detailed molecular formula composition of the three individual samples PMO-1, PMO-2, and PMO-3. Overall, nearly 80% of the observed mass spectral peaks in the ultrahigh resolution mass spectra were assigned molecular formulas, which is comparable to previous studies (Zhao et al., 2013; Dzepina et al., 2015). After removing the

duplicate molecular formulas containing $^{13}C$ or $^{34}S$, a total of 3168 (PMO-1), 2121 (PMO-2), and 1820 (PMO-3) monoisotopic molecular formulas remained. Groups of molecular formulas were assigned based on their elemental composition $C_cH_hN_nO_oS_s$, including: carbon, hydrogen, and oxygen (CHO); carbon, hydrogen, nitrogen, and oxygen (CHNO); and carbon, hydrogen, oxygen, and sulfur (CHOS). The most frequently observed compositions were CHO and CHNO. The reconstructed negative ion mass spectra of the monoisotopic molecular formulas for each of the samples are provided in Fig. 2. Visual comparisons

of the mass spectra indicate that PMO-2, which was likely transported through the North American continental and North Atlantic marine boundary layer, has an increased prevalence of higher O/C ratio formulas compared to the two samples transported through the free troposphere. Considering the ion distribution and normalized relative abundances, PMO-1 and PMO-3 mass spectra look quite similar with a high frequency of individual O/C values < 0.5. This may suggest similar emission sources or aerosol processing. In contrast, PMO-2 has a stronger relative influence of molecular compositions with higher O/C

ratios. The O/C histogram plots in Fig. 2 provide additional evidence for the O/C differences between the two types of samples (free troposphere and boundary layer) due to the difference in the O/C distribution.

The North American boundary layer outflow of organic aerosol captured in PMO-2 was likely influenced by SOA (Zhang et al., 2007) and thus is expected to have a higher initial O/C value compared to pyro-convected wildfire emissions of organic

aerosol (e.g., Aiken et al., 2008; Jimenez et al., 2009; Bougiatioti et al., 2014). Although the initial compositions are unknown, we anticipated that the samples with longer transport times (~ 1 week for PMO-1 and PMO-3) would be at least similar or perhaps more oxidized than PMO-2 which had a much shorter transport time (~ 3 days). This expectation was based on literature describing secondary organic aerosol formation and aging (Volkamer et al., 2006; Jimenez et al., 2009) and the reported molecular composition of continental boundary layer aerosol (Mazzoleni et al., 2012; Huang et al., 2014). The lower

oxidation observed in the free tropospheric samples transported for 7-10 days is consistent with our previous study at this site reported in Dzepina et al. (2015). In fact, when we compared the molecular formula composition of the free tropospheric aerosol sample "9/24" from Dzepina et al. (2015) to the free tropospheric samples in this study (PMO-1 and PMO-3), we observed that 86% and 91% of the molecular formulas are in common. FLEXPART simulations from both studies indicated these samples were all affected by wildfire emissions, contributing to their similarity. In contrast, only 75% of the formulas

found in the boundary layer sample (PMO-2) were common with those in Dzepina et al. (2015). These comparisons are provided in Table S2.

As observed in the mass spectra and histograms presented in Fig. 2, the samples have noticeable differences in the distribution of O/C values. This is also reflected in the abundance weighted mean O/C values for the samples: 0.48 ± 0.13 (PMO-1), 0.57

± 0.17 (PMO-2), and 0.45 ± 0.11 (PMO-3). Note that these O/C values are averages of thousands of individual measurements, as such the standard deviation represents the range of values and not uncertainties (see Figs. S7-S8 for violin plots of the distributions). We also note that the relative abundance of compounds in ESI mass spectra is not directly proportional to their solution concentration, other factors including surface activity and polarity impact the ionization efficiency (Cech & Enke, 2001). Nonetheless, the abundance does differentiate trends between the samples and the assigned molecular formulas

represent a collection of multifunctional isomers (e.g., LeClair et al., 2012). For completeness, both the abundance weighted average values for various metrics of aerosol oxidation and saturation (Table 2) and the unweighted average values (Table S3) are reported. Additional O/C distribution insight was derived from separating the species into CHO, CHNO, CHOS elemental groups. For example, the comparison of the species with CHO formulas in each sample indicates a smaller relative difference between PMO-2 aerosol compared to PMO-1 and PMO-3, with the PMO-2 aerosol having a higher average O/C value ($0.55 \pm 0.17$ (PMO-2) compared to $0.47 \pm 0.14$ (PMO-1) and $0.44 \pm 0.14$ (PMO-3)). Meanwhile 85 - 98% of the CHO species in each sample are present in at least one other sample, with 848 (42 - 78%) of the formulas being found in all three samples, as shown in Fig. S9. This suggests that the CHO composition may be fairly uniform throughout the atmosphere, without a significant abundance of clear marker species after long-range transport, regardless of the source region and transport time. This observation is consistent with other studies which have observed the decay of marker species after ~ 24 hours (Bougiatioti et al., 2014; Forrister et al., 2015).

In contrast, the CHNO molecular formulas demonstrate stronger differences that correlate with the overall O/C ratio. The average O/C value for the CHNO formulas in PMO-2 was $0.59 \pm 0.14$ compared to $0.49 \pm 0.15$ in PMO-1 and $0.49 \pm 0.14$ PMO-3 (Table 2). Differences in the elemental ratios are often visualized using the van Krevelen plot, which shows the correlation of H/C $vs.$ O/C. The van Krevelen plots for the three samples with the unique CHNO formulas present in each sample are shown in Fig. 3. Most of the unique CHNO species in PMO-2 (68%) fall in the more oxidized region of the plot (Tu et al., 2016) with high overall O/C values. This differs from the PMO-1 unique species that are predominantly on the less oxidized, low O/C side of the plot, or the oxidized aromatic region. Another observation from the CHNO species is more identified species in both PMO-1 (1120) and PMO-3 (608) than in PMO-2 (561), despite the higher total number of molecular species in PMO-2 compared to PMO-3. This is potentially due to the enrichment of $NO_x$ and $NO_y$ species as previously observed in wildfire pollution events (Val Martin et al., 2008a), which may in turn lead to an increased nitrogen content in the organic aerosol species. The nitrogen containing species show a distinct difference in terms of the total oxidation between the two sets of samples, more so than the CHO compounds. This implies that much of the distinction between aerosol sources may come from heteroatom containing species.

The difference in O/C is even more evident in the sulfur containing formulas (CHOS). The PMO-2 CHOS species have a much higher average O/C ratio ($0.74 \pm 0.34$) than what is observed in PMO-1 ($0.48 \pm 0.14$). Consistent with the CHNO formulas, the PMO-2 unique CHOS formulas (55% of unique formulas) are present in the oxidized region of the plot, whereas those in PMO-1 are nearly completely in the less oxidized region of the van Krevelen plot (Fig. S10). The Kendrick plot (Fig. S10c) also demonstrates a clear difference between the two samples. Most of the unique CHOS compounds in PMO-2 are located on the lower mass, higher defect side of the plot, while the PMO-1 formulas are on the higher mass, lower defect side. This difference is due to the larger amount of oxygen present in the PMO-2 formulas, which would lead to a greater mass defect than the more reduced CHOS formulas present in PMO-1. Higher oxygen content of PMO-2 aerosol is supported by its higher O/C ratio when compared to PMO-1 as shown in box plots (Fig. S10d). Very few CHOS molecular formulas (N = 29) were identified in PMO-3 and most of them (N = 26 of 29 total) were also present in PMO-1. Due to the small number of identified CHOS formulas in PMO-3, we did not consider it in the comparison between CHOS formulas in the samples. The increased number of sulfur species observed in PMO-2 are likely associated with the anthropogenic emission sources in the North American boundary layer. Overall, the observed differences in the O/C ratios between the boundary layer transported aerosol (PMO-2) compared to the free troposphere transported aerosol (PMO-1 and PMO-3) highlight differences in the aging and lifetime of aerosol relative to its transport pathway and emission source.

Another commonly used metric of aerosol oxidation is the average oxidation state of carbon ($OS_C$) described by Kroll et al. (2011). The average $OS_C$ includes both hydrogen and oxygen for the average oxidation of carbon in each molecular formula. Additionally, we assumed all nitrogen and sulfur were present as nitrate and sulfate functional groups and calculated the $OS_C$ with the appropriate corrections (Equation S1). The average $OS_C$ values (Table 2) for the three samples show again that PMO-2 is more oxidized than the other two samples. The average $OS_C$ values for the CHO formulas in PMO-1 and PMO-2 are very similar (Table 2), but as shown in the histograms in Fig. 4, their relative abundance distributions are quite different. The $OS_C$ *vs.* carbon number plots in Fig. 4 show slight differences between PMO-1 and PMO-2, mostly in the distribution of the sulfur containing formulas. However, the similarity of the PMO-1 and PMO-3 samples and their difference from the PMO-2 sample is quite clear in the visual comparisons of the histograms of the $OS_C$ values with their normalized relative abundances. The observation of an overall lower oxidation in PMO-1 and PMO-3 may support the findings of Aiken et al. (2008) and Bougiatioti et al. (2014) who reported that biomass burning aerosol are less oxidized than other types of aerosol, even after some aging. Conversely, the overall higher oxidation of PMO-2 implies that the sampled aerosol was likely more hygroscopic, included more efficient cloud condensation nuclei (Massoli et al., 2010), or had components of a less volatile nature (Ng et al., 2011) than PMO-1 and PMO-3.

### 3.4 Molecular formula aromaticity and brown carbon

The aromaticity of the samples is also different between the two groups of aerosol samples. Based on the aromaticity index (AI, Eq. S2; $AI_{mod}$, Eq. S3; Koch and Dittmar, 2006; 2016), the free tropospheric aerosol samples (PMO-1 and PMO-3) are more aromatic than the convected boundary layer aerosol (PMO-2; Fig. 5). The presence of more aromatic species in the long-range transported wildfire-influenced aerosol may lead to increased light absorption (Bao et al., 2017) and perhaps an increased resistance to oxidation (Perraudin et al., 2006). Aromatic species can also be associated with the presence of brown carbon (BrC; Desyaterik et al., 2013). Aromaticity is heavily dependent on the H/C ratio and the DBE (Eq. S4), where low H/C and high DBE indicate aromatic structure. Histograms depicting the distribution of H/C and DBE values for the three samples are shown in Fig. S11. As observed previously, PMO-1 and PMO-3 are more similar to each other than compared to PMO-2. Likewise, PMO-1 and PMO-3 exhibit an increase in the number frequency of higher DBE species, which is not observed in PMO-2, supporting the observation of an increased overall aromaticity for these free tropospheric aerosol samples. Many aromatic compounds, such as PAH are known to be carcinogens, and are a product of incomplete combustion biomass burning and anthropogenic emissions (Perraudin et al., 2006; Bignal et al., 2008).

Generally, BrC is considered to be aromatic or olefinic in nature (Bao et al., 2017). In our observations, the two samples influenced by wildfire show the greatest amount of olefinic and aromatic species, which is likely associated with the presence of BrC compounds. Additional evidence for the presence of BrC in PMO-1 comes from aethalometer measurements using the 7 wavelength aethalometer (Magee Scientific Company, Berkeley, California, USA) located at the site, which detected a wavelength-dependent peak with an Ångström exponent of 1.3 during the sampling period. Ångström exponents above 1 suggest the presence of BrC or iron oxides. Based on the retroplume analysis and comparison to similar samples (Dzepina et al., 2015), the detected peak is most likely the result of BrC. Figure S12 contains the aethalometer observations for this event. Difficulties with the instrument prevented similar data from being collected for PMO-3, although based on the retroplumes, ambient conditions, and molecular characteristics similar results seem likely. In addition to the aethalometer response, PMO-1 contained species that were related to BrC in studies by Iinuma et al. (2010) and Lin et al. (2016) (Table S4). This observation provides evidence for the persistence of BrC species, which is contrary to the observations by Forrister et al. (2015) who concluded that BrC is mostly removed within 24 hours. Additionally, the high concentration of OC for this sample makes it seem unlikely that we observed just a minor residual fraction. Perhaps, the lifetime of BrC is dependent on additional ambient conditions that influence aerosol oxidation and phase state.

### 3.5 Phase state, volatility, and cloud processing: Implications for the observed aerosol oxidation

Atmospheric aging processes are influenced by ambient conditions, such as temperature and water vapor, and the concentrations of reactive species. Recently, Shrivastava et al. (2017) reported observations of long-range transported PAH from Asia to North America and suggested an enhanced lifetime due to a probable glassy aerosol phase state during transport. Additionally, model simulations reported by Shiraiwa et al. (2017) indicated that model SOA is predicted to be semi-solid or glassy at altitudes above 2000 m in the northern hemisphere. Since the PMO aerosol was sampled at 2225 m above sea level, we examined the estimated glass transition temperature ($T_g$) of the studied WSOC species in addition to the markers of aqueous phase processing for the three PMO samples. Increased aerosol viscosity has been shown to decrease the rate of photodegradation (Lignell et al., 2014; Hinks et al., 2015) and water diffusivity (Berkemeier et al., 2014). Both photodegradation and water diffusion are expected to strongly affect the oxidation and aging of aerosol species during transport.

In general, lower volatility typically inversely correlates with $T_g$ (Shiraiwa et al., 2017) and viscosity. As such it was important to estimate the volatility of the PMO aerosol. Using the parameters reported by Donahue et al. (2011) and Li et al. (2016), we estimated the volatility of the FT-ICR MS identified organic aerosol molecular compositions (Figs. S13 and S14, respectively). As expected based on the length of transport for the samples, the majority of formulas show extremely low volatility. Interestingly, PMO-2 has a larger number of higher abundance molecular formulas with extremely low volatility and elevated oxidation relative to PMO-1 and PMO-3 (Fig. 6). This highlights the relationship between O/C and volatility, where volatility is expected to decrease as O/C increases when the mass range is constant (Ng et al., 2011); the relationship between oxygen and carbon and its effect on volatility is used by both Donahue et al. (2011) and Li et al. (2016) to estimate volatility. Similarly, lower volatility is expected to lead to lower diffusivity in aerosol even at elevated RH as demonstrated by Ye et al. (2016).

As predicted in earlier studies (Shrivastava et al., 2017; Shiraiwa et al., 2017), particles transported in the free troposphere are likely semi-solid to solid, where the actual particle viscosity depends on the ambient conditions and the composition of the particles. Thus, to better understand the potential phase state associated with the PMO organic aerosol, we first estimated the dry $T_g$ for the identified CHO molecular formulas in each of the PMO aerosol using the estimation method by DeRieux et al. (2018; Eq. S5). We then converted the dry $T_g$ to the RH dependent $T_g$ (below). Currently $T_g$ can only be estimated for CHO species, however the CHO species were the most frequently observed and constituted a major fraction of the total relative abundance in the PMO negative ion mass spectra. Assuming the identified CHO compositions are fairly representative of the total organic aerosol composition, a comparison of the $T_g$ values to the ambient temperature ($T_{amb}$) provides an indication of the likely phase state of the organic aerosol particles. Generally, if $T_g$ exceeds $T_{amb}$, a glassy solid state is predicted, likewise, if $T_g$ is less than $T_{amb}$ then either a semi-solid or liquid state is predicted depending on the ratio magnitude (Shiraiwa et al., 2017; DeRieux et al., 2018). Although the exact composition of the total organic aerosol is yet unknown, the identified water-soluble organic compounds provide a reasonable upper limit for the estimated $T_g$ values. Under this assumption, the CHO molecular formulas in PMO-1 and PMO-3 had higher average dry $T_g$ values than PMO-2 (Table S5, Fig. S16), which implies that they would be more viscous than PMO-2, given similar atmospheric conditions.

Water is known to be a strong plasticizer relative to typical aerosol species (Koop et al., 2011; Shiraiwa et al., 2017; Reid et al., 2018), thus it can decrease $T_g$ and the overall aerosol viscosity. Therefore, it's important to consider the ambient relative humidity when estimating the $T_g$. Using the extracted ambient temperature and RH from the GFS along the FLEXPART retroplumes and the Gordon-Taylor equation (Eqs. S6 - S7), the calculated dry $T_g$ were modified to RH-dependent $T_g$ for the CHO molecular species. The distributions of the $T_g$ values for the three PMO samples based on one standard deviation of the

ambient conditions are shown as boxplots in Fig. 7. The range of ambient temperature and RH extracted from the GFS along the FLEXPART simulated path yields a wide range of $T_g$ values (Figs. 7, S17). The estimates were taken back only 5 days due to the increasing range of possible meteorological conditions associated with the spread in the air masses as shown in Figs. 1 and S1-S3. Overall, the distributions of $T_g$ values in PMO-1 and PMO-3 generally exceed the ambient temperature (Fig. 7), implying that particles containing a majority of these compounds would likely be solid. To account for the low molecular weight organic anions not observed in the FT-ICR mass spectra, their mass concentrations and $T_g$ values (estimated using the Boyer-Kauzmann rule (Koop et al., 2011; Shiraiwa et al., 2017; DeRieux et al., 2018)) are also shown in Fig. 7. The three most prevalent low molecular weight organic acids indicate the potential impact of those compounds on the overall $T_g$ value of a particle that contains them. Oxalic acid was estimated to have a similar $T_g$ value to a majority of the higher MW species identified in PMO-2, but it is slightly lower than the majority of species in PMO-1 and PMO-3. However, the mass fraction of oxalate is 3 times lower in PMO-1 and PMO-3 (2.3 and 3.0 %) compared to PMO-2 (9.4 %).

The results suggest that aerosol in PMO-1 and PMO-3 was overall less susceptible to atmospheric oxidation due to the aerosol phase state during free tropospheric long-range transport than it may have been in the boundary layer with higher ambient RH and temperature. A more viscous phase state during transport may also explain the presence of persistent BrC species in PMO-1, where the BrC species are protected from oxidation similarly to the long-lived PAHs observed by Shrivastava et al. (2017). In contrast to the observations from PMO-1 and PMO-3, much of the PMO-2 $T_g$ distribution falls below the ambient temperature implying a semi-solid or liquid state during the final 5 days of transport. This indicates an increased susceptibility to oxidation processes in the atmosphere (Shiraiwa et al., 2011), such as aqueous phase processing. The possibility of aqueous phase processing is also supported by the extracted GFS RH in Fig. 7, which is above 50% for the last 5 days of PMO-2 transport. The potential for liquid/semi-solid aerosol in the boundary layer is consistent with other studies (Shiraiwa et al., 2017; Maclean et al., 2017) due to the increased RH in the boundary layer and the plasticizing effect of water. Although, we note the PMO-2 average dry $T_g$ values were 4-5° lower than those of PMO-1 and PMO-3. Overall, the estimates of dry $T_g$ and RH-dependent $T_g$ provide an otherwise unattainable upper limit estimate of the aerosol phase state of the sampled free tropospheric aerosol in this study.

As described above, the most obvious difference in the molecular composition of PMO-2 vs. PMO-1 and PMO-3 is the increased extent of oxidation. In fact, most of the unique species observed in PMO-2 are in the highly oxidized region of the van Krevelen plot (Fig. 8). However, the exact oxidation pathways that led to the increased oxidation observed for PMO-2 and its initial composition are unclear. Both gas phase and aqueous phase reactions lead to SOA, where aqueous SOA components can have higher O/C values than gas phase SOA components (Lim et al., 2010; Ervens et al., 2011). The high numbers of CHNO and CHOS molecular formulas observed here are consistent with secondary components associated with an emission plume likely enriched in $SO_2$, NOx, and $O_3$ pertaining to its expected anthropogenic influence. All three of these reactive species have been shown to lead to production and oxidation of SOA in the atmosphere (Hoyle et al., 2016; Bertrand et al., 2018). Cloud and aqueous processing have also been shown to increase the oxidation of atmospheric organic matter (e.g., Ervens et al., 2008; Zhao et al., 2013; Cook et al., 2017; Brege et al., 2018). Comparisons of the detailed molecular composition of the PMO samples with studies of cloud (Zhao et al., 2013; Cook et al., 2017) and fog (Mazzoleni et al., 2010) organic matter indicate that the formulas uniquely common to only PMO-2 have higher O/C, which supports aqueous phase processing during transport. These results are provided in Fig. S19 and Table S6. Studies have shown that the reactive species emitted from anthropogenic plumes ($SO_2$, NOx, $O_3$) can play a role in the oxidation of the organic species that are dissolved in water (Blando and Turpin, 2000; Chen et al., 2008; Ervens et al., 2011); furthermore, studies have shown aerosol liquid water content contributes to aqueous production of SOA (Volkamer et al., 2009; Lim et al., 2010). The elevated RH extracted from the GFS for this plume (Fig. 7) indicates the presence of aerosol liquid water and is consistent with its ubiquitous nature (Nguyen et al.,

2016). Additionally, PMO-2 had a strongly elevated non-sea salt sulfate concentration relative to PMO-1 and PMO-3, which also indicates aqueous phase processing (Crahan et al., 2004; Yu et al., 2005; Sorooshian et al., 2007; Hoyle et al., 2016). Oxalate, another well-known marker of potential aqueous phase processing (Warneck 2003; Crahan et al., 2004; Yu et al., 2005; Sorooshian et al., 2007; Carlton et al., 2007), was also elevated in PMO-2. The organic mass fraction of oxalate was 9.4 % in PMO-2 compared to 2.3 % and 3.0 % in PMO-1 and PMO-3. The nitrate concentration in PMO-2 was very low compared to PMO-1 or PMO-3 (Table 1), supporting aqueous phase processed aerosol in PMO-2. While clearly gas phase SOA cannot be excluded, several lines of evidence suggest that aqueous phase oxidation likely influenced the chemical and physical characteristics of the PMO-2 aerosol to a larger extent than those of PMO-1 and PMO-3 based on the observed molecular characteristics, major ion concentrations (Fig. S20), and the model simulated transport pathways and GFS meteorology.

## 4 Conclusions

Aerosol samples collected on 27-28 June 2013 (PMO-1), 5-6 July 2014 (PMO-2), and 20-21 June 2015 (PMO-3) at the Pico Mountain Observatory were analyzed using ultrahigh resolution FT-ICR mass spectrometry for molecular formula composition determination. FLEXPART retroplumes for the sampled air masses indicate that: (a) PMO-1 and PMO-3 aerosol were transported predominantly through the free troposphere and were primarily influenced by wildfire emissions; and (b) PMO-2 aerosol were transported primarily through the boundary layer over the Northeast continental U.S. and the North Atlantic Ocean and was largely influenced by anthropogenic and biogenic sources. Although elevated levels of OC, sulfate, and oxalate were found in all three samples, PMO-2 had the overall highest mass fractions of oxalate and sulfate indicating a clear influence of aqueous phase processing. The molecular formula assignments indicated differences in the aerosol oxidation between the free troposphere transported aerosol (PMO-1 and PMO-3) and the boundary layer transported aerosol (PMO-2). These observations suggest that the transport pathways, in addition to the emission sources, contributed to the observed differences in the organic aerosol oxidation. The ambient temperature and RH at upwind times were extracted from the GFS analysis in FLEXPART and were used to estimate the glass transition temperatures of the aerosol species during transport. The results suggest that the organic aerosol components extracted from PMO-1 and PMO-3 were considerably more viscous due to lower RH than those from PMO-2 and thus were less susceptible to oxidation. The relationship between aerosol viscosity and its susceptibility to oxidation in the free troposphere is well supported (e.g., Koop et al., 2011; Berkemeier et al., 2014; Lignell et al., 2014; Shiraiwa et al., 2017). This suggests that biomass burning emissions and BrC injected into the free troposphere are more resistant to removal than aerosol transported in the boundary layer, due largely to the ambient conditions in the free troposphere. Although more work is needed to better constrain the molecular composition of long-range transported aerosol and the processes that affect it during transport, the presented results have broader implications for the aging of long-range transported aerosol rapidly convected to the free troposphere.

### Data availability

The molecular formula composition for each sample in this study is available via Digital Commons at the following link: http://digitalcommons.mtu.edu/chemistry-fp/88/.

### Acknowledgements

This project was supported with funding from NSF (AGS-1110059) and DOE (DE-SC0006941). Logistical support for the operation of the Pico Mountain Observatory was provided by the Regional Secretariat of the Environment and the Sea of the Portuguese Regional Government of the Azores. Major equipment cost share and graduate student support associated with this project was provided by the Earth, Planetary, and Space Science Institute at Michigan Technological University. We thank Mike Dziobak, Kendra Wright, Sumit Kumar, Andrea Baccarini, Stefano Viviani, Jacques Huber, and Detlev Helmig for assistance in the field. We thank Manabu Shiraiwa, Sarah Petters, and Marcus Petters for helpful discussion regarding aerosol

phase state. Finally, we thank Melissa Soule and Elizabeth Kujawinski of the Woods Hole Oceanographic Institution (WHOI)

Mass Spectrometry Facility for FT-ICR MS instrument time and assistance with data acquisition (NSF OCE-0619608 and Gordon and Betty Moore Foundation).

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

**Tables**

**Table 1.** Average concentrations ($\mu g/m^3$) of major ions and organic carbon.

| Component | PMO-1 | PMO-2 | PMO-3 |
|---|---|---|---|
| Acetate | 0.0519 ± 0.0001 | 0.004587 ± 0.000005 | 0.0071 ± 0.0002 |
| Formate | 0.0289 ± 0.0003 | 0.00438 ± 0.00007 | 0.0119 ± 0.0001 |
| MSA | 0 | 0.00439 ± 0.00006 | 0.00232 ± 0.00002 |
| Chloride | 0.0104 ± 0.0003 | 0 | 0.0310 ± 0.0001 |
| Nitrate | 0.189 ± 0.002 | 0.0173 ± 0.0004 | 0.3010 ± 0.00028 |
| Sulfate | 0.338 ± 0.004 | 1.07 ± 0.01 | 0.421 ± 0.003 |
| Oxalate | 0.0938 ± 0.00070 | 0.0897 ± 0.00181 | 0.05222 ± 0.00002 |
| Sodium | 0.2101±0.0004* | 0.2560±0.0001* | 0.548±0.005* |
| Ammonium | 0.1364 ± 0.0004 | 0.2394 ± 0.00001 | 0.1193 ± 0.00062 |
| Potassium | 0.0791 ± 0.0020** | 0.0126 ± 0.0002 | 0.0197 ± 0.0003 |
| OC | 2.07 ± 0.02 | 0.478 ± 0.026 | 0.87 ± 0.10 |

*Sodium concentrations were not background subtracted due to inconsistent and high blank levels, they are included to provide an upper limit on the approximate sodium concentrations.

**Replicate measurements of potassium were inconsistent. The sample could not be re-analyzed because there was no remaining sample. Standard deviation was determined by looking at the average standard deviation of 36 potassium measurements in other samples. This sample should only be considered as elevated potassium and only minimal importance placed on the actual measured value.

**Table 2.** Molecular formula composition with abundance weighted average values and the numbers of formulas for each elemental group.

| Sample | Group | $O/C_w$ | $H/C_w$ | $DBE_w$ | $OS_{Cw}$ | Number |
|---|---|---|---|---|---|---|
| PMO-1 | All | 0.48 ± 0.13 | 1.30 ± 0.28 | 7.74 ± 3.38 | -0.42 ± 0.43 | 3168 |
| PMO-2 | All | 0.57 ± 0.17 | 1.36 ± 0.22 | 6.42 ± 2.54 | -0.30 ± 0.46 | 2121 |
| PMO-3 | All | 0.45 ± 0.11 | 1.34 ± 0.41 | 7.45 ± 3.15 | -0.50 ± 0.41 | 1820 |
| PMO-1 | CHO | 0.47 ± 0.14 | 1.31 ± 0.29 | 7.43 ± 3.68 | -0.37 ± 0.44 | 1848 |
| PMO-2 | CHO | 0.55 ± 0.17 | 1.35 ± 0.25 | 6.43 ± 3.66 | -0.26 ± 0.45 | 1281 |
| PMO-3 | CHO | 0.44 ± 0.14 | 1.37 ± 0.31 | 6.93 ± 3.82 | -0.48 ± 0.48 | 1183 |
| PMO-1 | CHNO | 0.49 ± 0.15 | 1.2 ± 0.26 | 9.44 ± 3.09 | -0.50 ± 0.3 | 1120 |
| PMO-2 | CHNO | 0.59 ± 0.14 | 1.25 ± 0.19 | 8.20 ± 2.19 | -0.38 ± 0.29 | 561 |
| PMO-3 | CHNO | 0.49 ± 0.14 | 1.23 ± 0.21 | 9.25 ± 2.41 | -0.52 ± 0.25 | 608 |
| PMO-1 | CHOS | 0.48 ± 0.14 | 1.78 ± 0.35 | 2.87 ± 3.28 | -1.20 ± 0.42 | 200 |
| PMO-2 | CHOS | 0.74 ± 0.34 | 1.57 ± 0.23 | 4.05 ± 2.45 | -0.54 ± 0.51 | 274 |
| PMO-3 | CHOS | 0.40 ± 14 | 1.90 ± 0.47 | 1.60 ± 4.29 | -1.50 ± 0.20 | 29 |

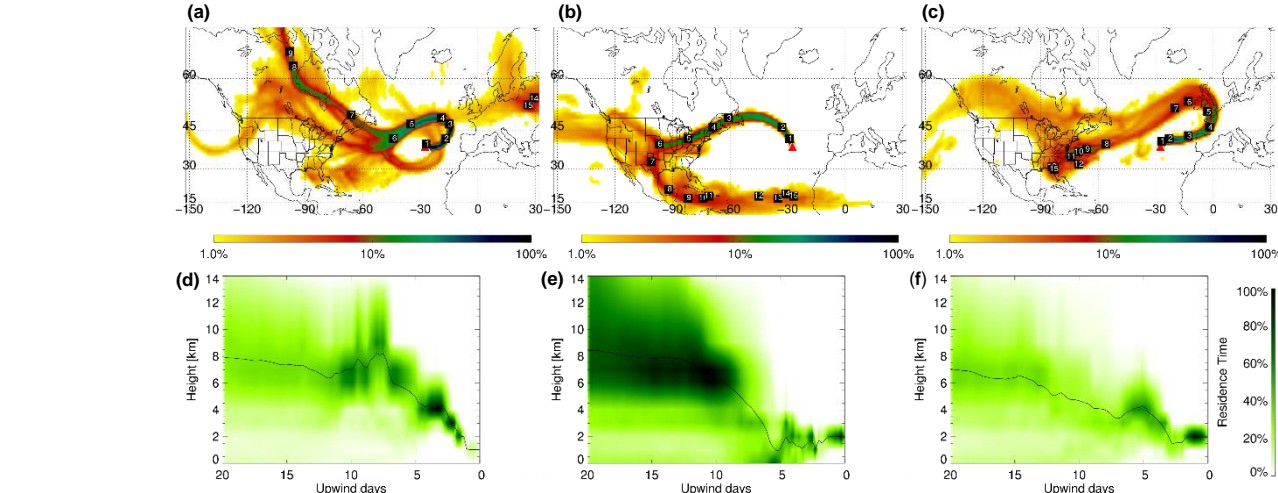

**Figure 1.** FLEXPART retroplumes for 28 June 2013 06:00 (PMO-1, (a, d)), 6 July 2014 03:00 (PMO-2, (b, e)), and 21 June 03:00 2015 (PMO-3, (c, f): column integrated residence time over the 20-day transport time (a-c) and vertical distribution of the retroplume residence time at given upwind times (d-f). The labels indicate the approximate locations of the center of the plume for each of the transport days. Residence time is color coded by logarithmic grades representing its ratio to the location of maximal integrated residence time (100%). The black line in d-f indicates the mean height of the plume during transport.

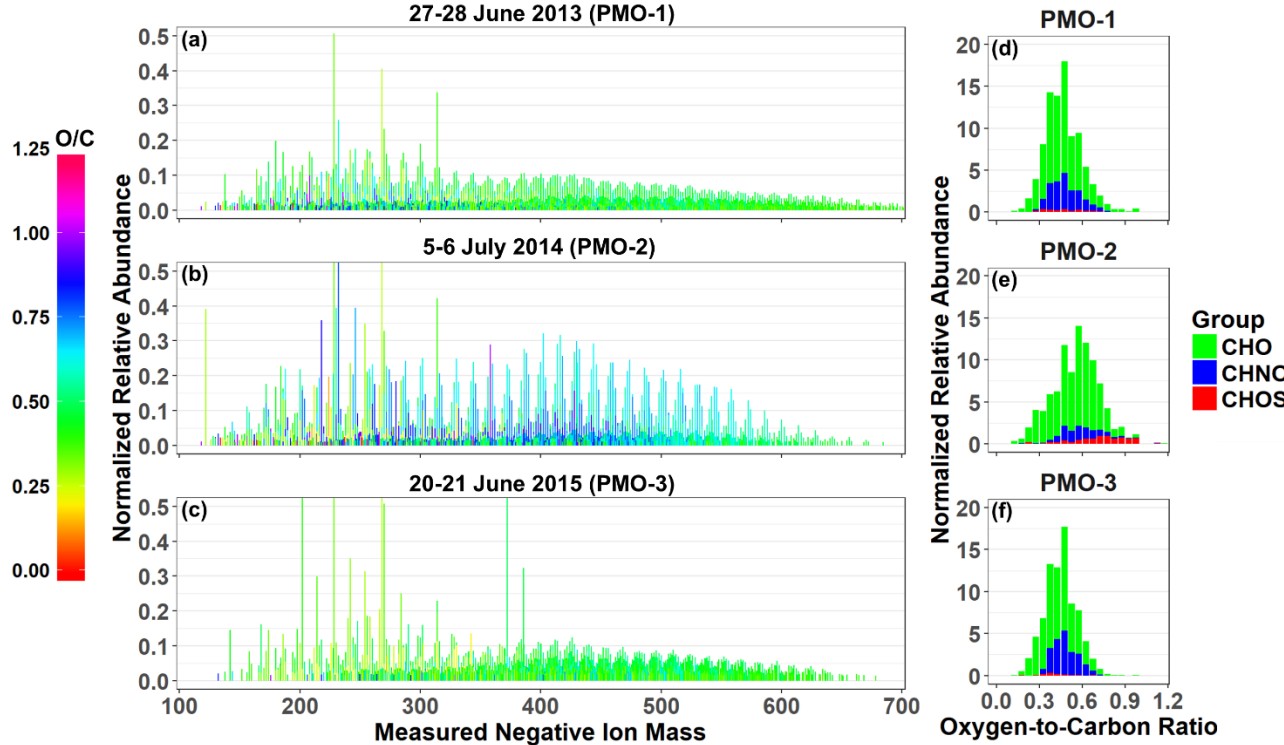

**Figure 2.** Reconstructed negative ion mass spectra (a-c) and O/C histograms (d-f) for the three PMO samples. The color in the mass spectra indicates the O/C value for the molecular formula it represents. The tallest peaks in the mass spectra exceed the range, this was done to improve the visibility of the lower abundance species (see also Fig. S6).

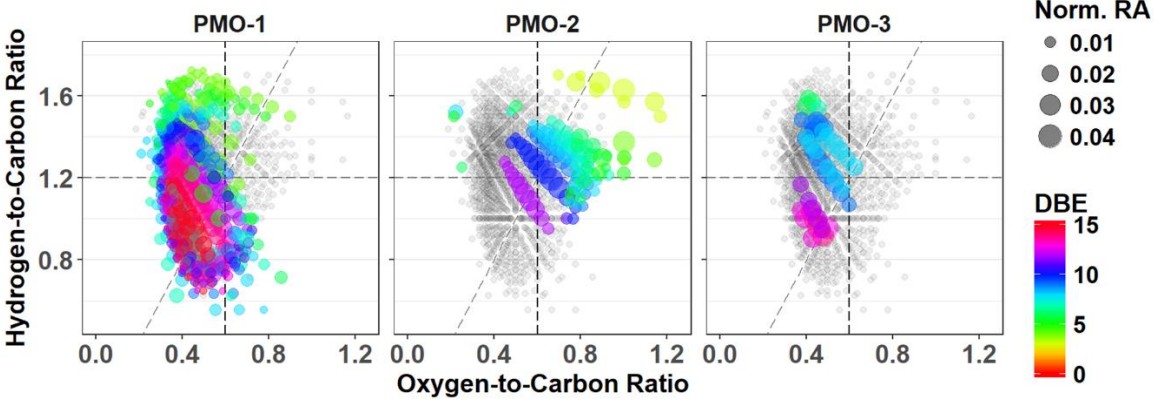

**Figure 3.** Van Krevelen plots for the CHNO species with all identified CHNO species (grey) and unique species (colored). The color represents the number of double bond equivalents in the corresponding molecular formula. The diagonal line is an oxidation line ($OS_C = 0$ for C, H, O elements; Tu et al., 2016), where the more oxidized formulas are on the right side and less oxidized are on the left.

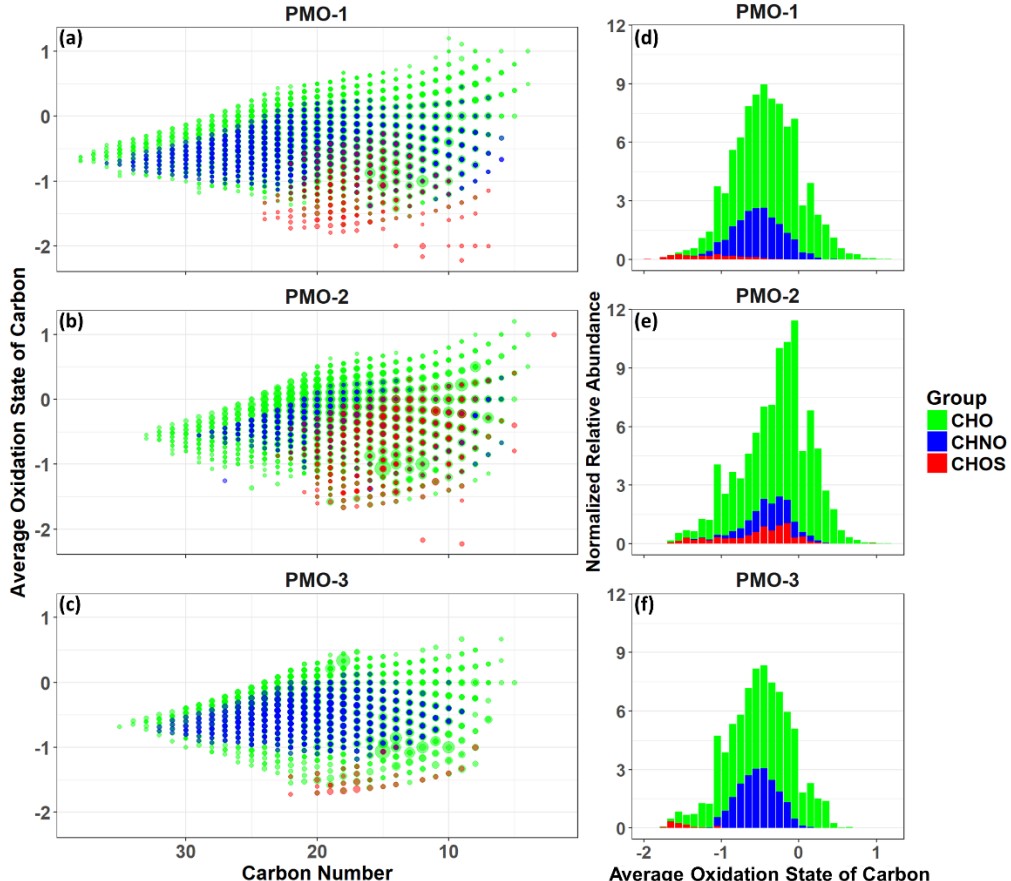

**Figure 4.** Average OS$_C$ vs. carbon number plots for molecular formula identified in each of PMO samples (a-c). The size of the symbols is scaled to the analyte relative abundance and the color represents the elemental group: CHO (green), CHNO (blue), and CHOS (red). The right panel (d-f) contains average OS$_C$ histograms based on the sum of normalized abundance.

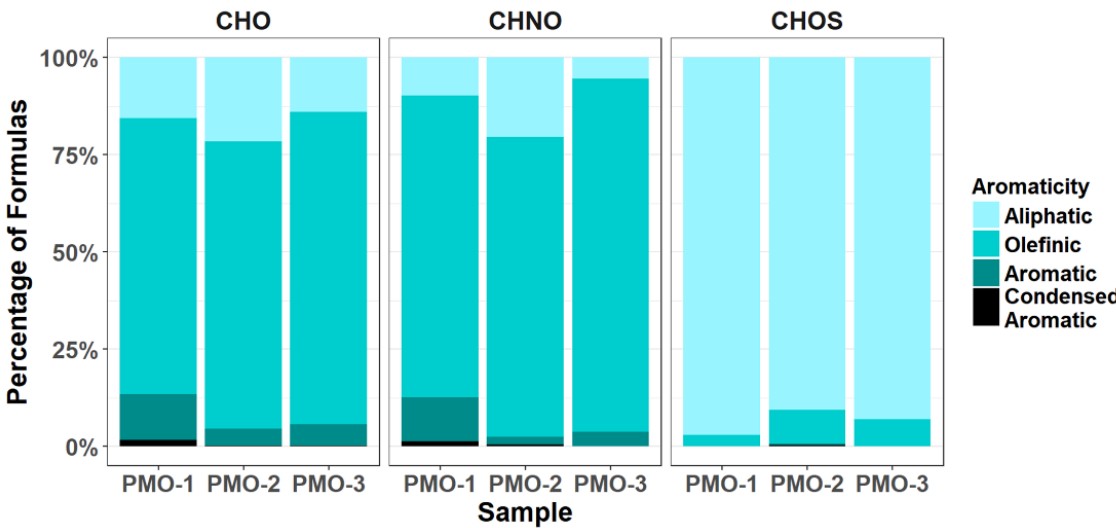

**Figure 5.** Normalized bar charts for the aromaticity of the three PMO samples, calculated using the Koch and Dittmar (2006; 2016) modified aromaticity index (AI$_{mod}$).

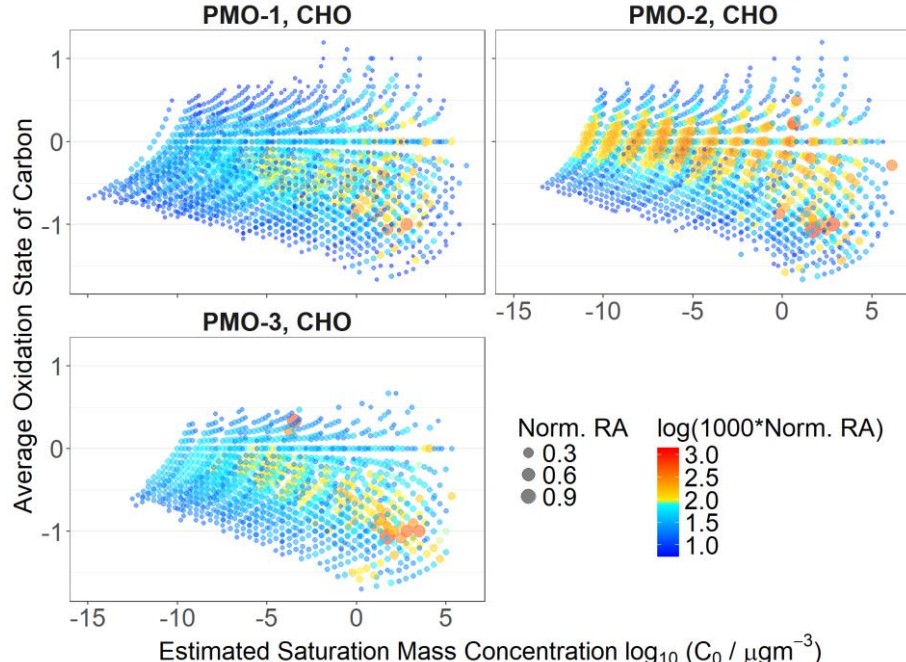

**Figure 6.** $OS_C$ *vs.* volatility estimated using the Li et al. (2016) method for the CHO species in the three samples. The size is determined by the normalized relative abundance and the color is determined by the logarithm of the normalized relative abundance multiplied by 1000.

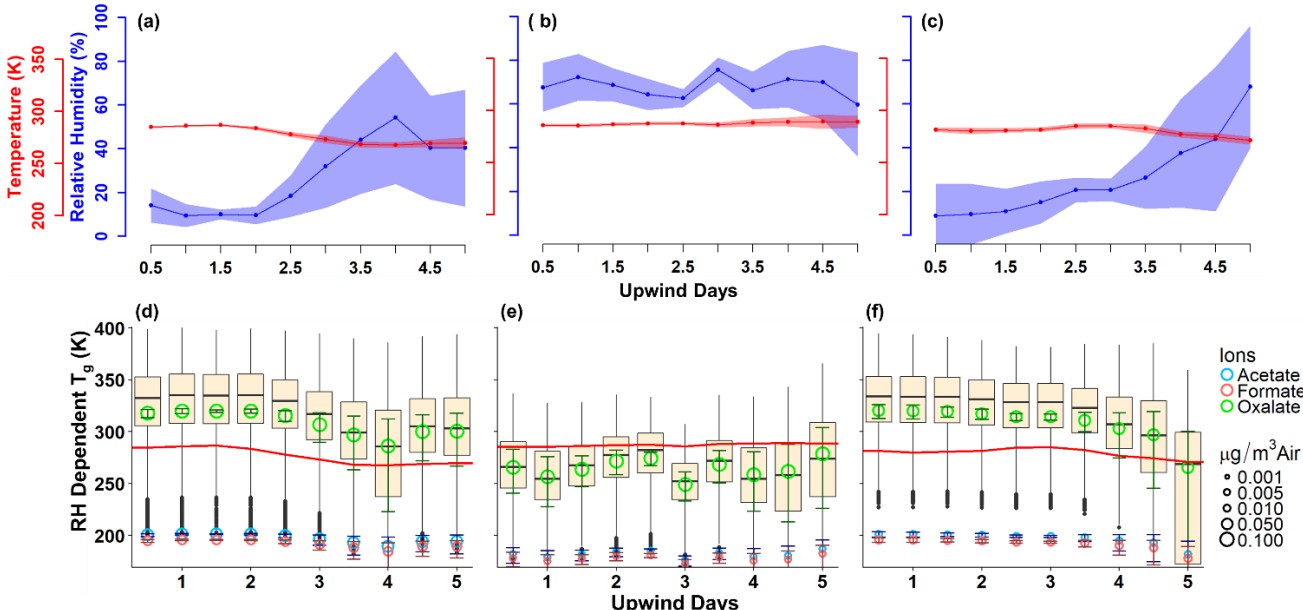

Figure 7. Panels a-c contain the ambient conditions extracted from the GFS analysis along the FLEXPART modeled path weighted by the residence time for PMO-1, PMO-2, and PMO-3, respectively. The line represents the mean value and the shading represents one standard deviation of values. Panels d-f contain the boxplot distributions of the relative humidity dependent $T_g$ values for molecular formulas using the maximum, mean, and minimum RH for PMO-1, PMO-2, and PMO-3, respectively. The $T_g$ values for the full composition of each sample were calculated using the maximum, mean, and minimum RH and then all three sets of data are combined and plotted as a single distribution for each time period. The open circles represent the abundance and Boyer-Kauzmann estimated $T_g$ for the acid forms of the three most abundant low MW organic ions, the bars around the circles represent the range of possible $T_g$ values for those compounds when the range of RH is considered. The red line demonstrates the ambient temperature at each time point, as extracted from GFS. The centerline of the boxplot represents the median, the top and bottom of the "box" represent the third and first quartiles, respectively. The "whiskers" represent Q3 + 1.5* interquartile range (IQR, Q3-Q1) (maximum), and Q1 – 1.5*(IQR) (minimum).

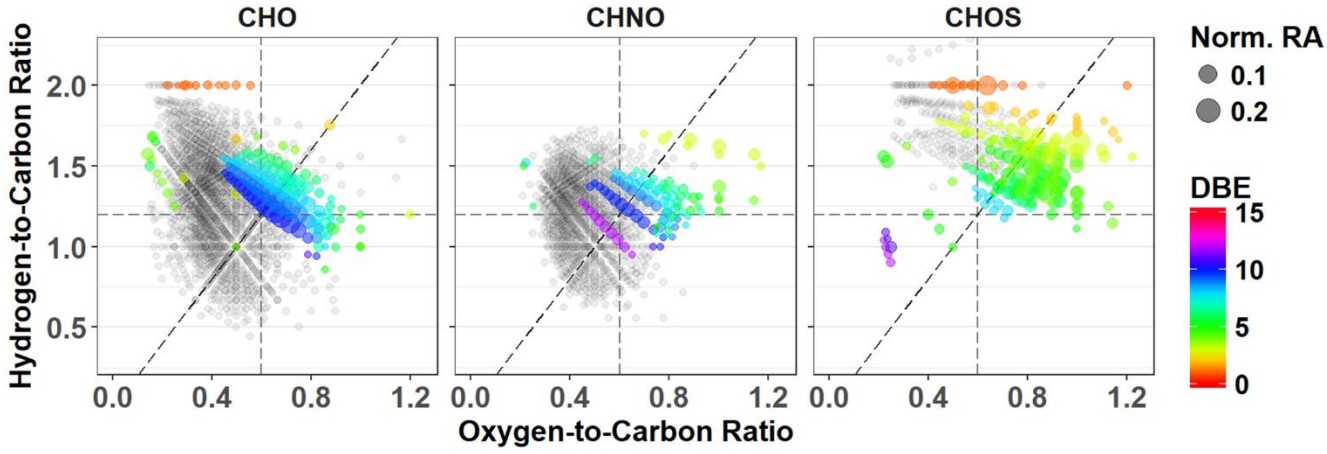

990

**Figure 8.** PMO-2 van Krevelen plots for unique molecular formulas separated by group. Symbols are scaled to indicate the normalized relative abundance. The DBE is indicated for each of unique molecular formulas using colored symbols. Formulas common with other samples are provided in grey for context.