# Peer review of "Molecular and physical characteristics of aerosol at a remote free troposphere site: Implications for atmospheric aging"

_Atmospheric Chemistry and Physics, 2018_

## Referee Comment (RC1) · A. L. Vogel (Referee) · 14 Mar 2018

Review on:

**Schum et al.: "Molecular and physical characteristics of aerosol at a remote marine free troposphere site: Implications for atmospheric aging"**

The authors describe the analysis of three selected filter samples that were collected within a more comprehensive sampling campaign (a total of 127 filters) at the Pico Mountain Observatory (PMO) on Pico Island / Azores. The samples were chosen because of the high organic carbon (OC) concentration. While major small ions and OC of the three filter samples were measured as well, the focus of the manuscript certainly lies on the analysis using direct infusion electrospray ionization ultra-high resolution mass spectrometry in the negative ion mode ((−)ESI/UHRMS). Differences in the mass spectra are discussed with regard to a back-trajectory analysis. The authors observe signals in one sample (out of three) that exhibits higher O/C ratios compared to the other two samples that likely have undergone a longer atmospheric transport time (and thus aging). The authors argue that the two samples with the lower O/C ratio were transported in the free troposphere (FT) to PMO, and thus the particle phase state during transport was likely solid. They conclude that "biomass burning emissions", which are directly injected by pyro-convection into the FT, "are longer-lived than emissions in the boundary layer".

*General comments:*

Overall, the manuscript presents results from an atmospheric measurement station that is certainly very well suited for studying aerosol transformation processes during long-range transport. Furthermore, the authors demonstrate the need for ultra-high resolution mass spectrometry techniques when it comes to ambient measurements of particulate matter. However, I have major concerns regarding a weak reasoning that is used as a basis for their conclusions and implications for atmospheric aging (see my point (1) below). Also, the authors remain too speculative in many cases, or even state arguments that are not supported by their figures (see (2)). Furthermore, I have serious technical concerns that might have an effect on the outcome of the ESI/UHRMS analysis (see (3)).

My major remarks concerning the above mentioned points:

(1) The authors argue that particle phase state is affected by the conditions during atmospheric transport. Their observation of a low O/C in the biomass burning samples (PMO-1 and PMO-3) is reasoned by the phase state of the particles during transport. Although, the authors mention that the PMO-2 sample is originating from another source (from the Eastern United States – dominated by a mix of biogenic and anthropogenic emissions), they argue that the high O/C ratio of this sample is caused by the semi-solid phase state, which allows faster aging during atmospheric transport to PMO. Assuming that the back-trajectory analysis is getting the sources right, the authors don't present a convincing argument why we can use the two different sources (biomass burning organic aerosol (BBOA) vs. anthropogenic/biogenic secondary aerosol (A/BSOA)) as an **identical reference point for the onset of atmospheric aging**!
In the literature it is now well recognized that these two kinds of organic aerosol (BBOA vs SOA from anthropogenic and biogenic VOCs) are already very different on the molecular level at the time when emitted by their sources or formed in the atmosphere: While BBOA is largely composed of lignin- and cellulose-derived

condensed aromatic / polyphenolic structures with low O/C ratios (Lin et al., 2016), numerous studies have shown that the auto-oxidation of (mostly biogenic) VOCs results in highly oxygenated molecules within seconds after the initial attack by an oxidant (Crounse et al., 2013; Ehn et al., 2012; Jokinen et al., 2014). Although it is not yet fully understood what happens to these compounds once they condense, the auto-oxidation mechanism still can explain high aerosol O/C from atmospheric oxidation of VOCs.

My impression is that the authors do not adequately consider or discuss different reasons for their observations and overemphasize the possible link between atmospheric transport and aging efficiency at different altitudes.

To be clear, I am not saying that aerosol phase state does not change aerosol transformation rates, but to extract this effect from ambient observations, one likely needs to consider a larger set of samples (which the authors apparently have).

(2) The authors argue that the aerosol that was captured on the PMO-2 sample travelled at altitudes below 2 km over Eastern U.S. and stayed below 2 km altitude until it reached PMO 2-4 days later (p. 7, l. 251-253). From Figure 1 (e), I cannot see that. For the upwind days 0-5, the mean height of the plume is consistently higher than 2 km. As stated on p.2, l. 65-67, the marine boundary layer (MBL) around PMO ranges between 500 m and 2 km, and thus below the mean height of the plume. However, the authors argue that **PMO-2 air masses travelled within the MBL layer** to PMO, explaining high relative humidity and a semi-solid phase state during transport. Another argument against the transport within the MBL is given by the authors, mentioning that PMO-2 does not reveal any chemical signature from the MBL (p. 6, l. 232-233). Furthermore, the mean height in PMO-3 appears even lower than PMO-2 for the last five days.

(3) The discussed filters were selected because of their high mass loadings of organic carbon (>1 mg OC / quarter filter). After loading the water-soluble (WS) OC extract onto solid phase extraction (SPE) material for purification, the SPE was then eluted by 2 mL of MeCN/$H_2O$ and the extract was used for direct infusion. If we assume that half of the OC is WSOC and assume 100% SPE collection efficiency (neglecting losing the small, polar organic compounds), the concentration of the solution for direct injection ESI would be ~0.25 mg/mL. To me, this appears as a huge concentration in which ion source cluster formation (e.g. x-mers of analytes, clusters with solvents or solvent additives, impurities), can become a serious issue. I understand that SPE was done in order to reduce cluster formation with inorganic ions and that a separation technique was apparently not available. However, the authors could have done straight-forward tests to check the extent of cluster formation in these samples by: (1) sample dilution and checks for non-linear reduction of cluster-signal candidates and (2) MS/MS isolation and recording the fragmentation energy of cluster signal candidates.

Although, I understand that MS/MS cannot be done on all ion signals, there are some "suspicious" signals standing out in Figure 2 (the signals > 0.3 rel. abundance) that should have been checked using MS/MS when doing direct injection.

Ion source cluster formation would introduce a bias on the calculated glass transition temperature ($T_g$) by artificially increasing the average number of carbon per molecule. Furthermore, the overall $T_g$ is already biased toward higher values since small molecules are very likely lost during the SPE procedure (l. 158-159). The manuscript misses in its current form a critical evaluation of these points and its implications on aerosol phase state and aging.

Finally, it would have been interesting to measure also in the positive ESI mode, in which one can observe levoglucosan or nitrogen-heterocycles that are expected in biomass burning aerosol samples.

Moreover, it is not clear how the blank signal of the DI/ESI-UHRMS was determined. I would expect a measurement of a blank filter that undergoes the whole procedure incl. transport from the field site, sample preparation procedure in laboratory (sonication, filtration, SPE, etc). Only a good blank measurement allows determining the significance level at which individual signals are present in the samples and identifying those signals that emerge from sample preparation. Here, it is especially important since the paper discusses the number of identified compounds between the different samples. If a compound was identified as "not present" in one sample, does that mean after blank subtraction? It is not described what were procedures involving a blank filter, nor are mentioned the criteria and the thresholds for this kind of filtering!

Overall, I cannot recommend the article to be published in ACP, since the conclusions reached remain far too speculative and are not convincingly supported by the presented data. I miss a more critical discussion and evaluation of other potentially important processes (both atmospheric and instrumental) throughout the manuscript. The description of the mass spectrometry analysis is not sufficiently complete and leaves the reader with open questions (e.g. What was the workflow of the data analysis? What did they use as blank samples?). Last but not least, the presentation and language is in many cases not precise.

*Specific comments*

p.3, l. 106-109: Please provide a reference stating that long-range transported aerosol is generally acidic in nature. Furthermore, negative ESI is not only sensitive to organic acids, but also to important biomass burning markers (e.g. nitro-phenols (Iinuma et al., 2010)). Have you seen nitro-phenols or similar biomass burning tracers in the biomass burning samples?

p. 5, l. 169: The two references describe different criteria for the molecular formula assignments:
       Dzepina et al.: max. 100 C, 400 H, 100 O, 3 N, and 1 S.
       Mazzoleni et al.: max. 70 C, 140 H, 25 O, 3 N, and 1 S.
The elemental windows for nitrogen and sulfur seem very strict. The used limits exclude for example the identification of nitrogen-heterocycles with four nitrogen (e.g Kampf et al., 2012). Given the clear isotopic signature of sulfur, why was not more sulfur allowed? Were the isotopic patterns used to confirm the molecular formulas in case multiple elemental compositions appeared within the instrumental accuracy limits?

p. 5, l. 172: According to Putman et al., allowing nitrogen for compounds larger than 500 amu results in multiple results within 1 ppm. Does that mean that the number of elements allowed was chosen such strict that only one molecular formula per signal was obtained?

p. 6, l. 205-222: As mentioned by the editor, this paragraph is not well structured and needs rewriting.

p. 8, l. 308: Is the high O/C of the CHNO species potentially driven by organic nitrates?

p. 8, l. 310-311: Does the common number of identified molecules in a certain group (here CHNO) really tell us something about the similarity of samples? This is mentioned several times in the manuscript, and I don't understand why intensity of compounds is not given more weight in the discussions.
Why not visualizing the similarity of two different samples by simple scatter plots of the intensity of all ions of sample A vs intensity of all ions of sample B. This would also allow determining the Pearson correlation coefficient.

p. 8, l. 319: also here: Is the high O/C of the CHOS species potentially driven by organic sulfates? What would be the O/C after accounting for organic sulfates? Is it then still different from the CHO value?

p. 9, l. 334 ff.: The oxygen that originates from organic nitrates and sulfates artificially increases the oxidation state of carbon.

p. 10, l. 394: Was the glass transition temperature determined only for single molecules and not for the intensity weighted population of all ion signals? Obviously the atmospheric particles are mixtures and therefore only the glass transition temperature derived from the whole spectrum is meaningful. Accounting for the fact that small organics are lost during the SPE (which would reduce $T_g$ of the mixture), what would be the effect on $T_g$ of the mixture if one assumes that 10, 20 or 50% of total OC consist of small organics?

Figure 3: The three dots with H/C<0.6 are cyan in PMO-1, which should mean unique in PMO-1. Why are these three signals in PMO-2 and PMO-3 then grey (common signals)?

*Technical corrections*

Figure 1: The Figure quality is not appropriate.

Figure 2: X-axis ticks are missing for (a),(b),(d) and (e). It seems as the highest peaks in (b) and (c) are cut at the top, or are they all the same height?

Figure 4: X-axis ticks are missing for (a), (b), (d) and (e).

Figure 4: The y-label is number of formulas. The caption is not clear on that. What is the "normalized abundance"? On what is it normalized?

Figure 6: X-axis ticks are missing for PMO-1.

*Literature*

Crounse, J. D., Nielsen, L. B., Jorgensen, S., Kjaergaard, H. G., and Wennberg, P. O.: Autoxidation of Organic Compounds in the Atmosphere, Journal of Physical Chemistry Letters, 4, 3513–3520, doi:10.1021/jz4019207, 2013.
Ehn, M., Kleist, E., Junninen, H., Petaja, T., Lonn, G., Schobesberger, S., Dal Maso, M., Trimborn, A., Kulmala, M., Worsnop, D. R., Wahner, A., Wildt, J., and Mentel, T. F.: Gas phase formation of

extremely oxidized pinene reaction products in chamber and ambient air, Atmos. Chem. Phys., 12, 5113–5127, doi:10.5194/acp-12-5113-2012, 2012.

Iinuma, Y., Boege, O., Graefe, R., and Herrmann, H.: Methyl-Nitrocatechols: Atmospheric Tracer Compounds for Biomass Burning Secondary Organic Aerosols, Environ. Sci. Technol., 44, 8453–8459, doi:10.1021/es102938a, 2010.

Jokinen, T., Sipila, M., Richters, S., Kerminen, V.-M., Paasonen, P., Stratmann, F., Worsnop, D., Kulmala, M., Ehn, M., Herrmann, H., and Berndt, T.: Rapid Autoxidation Forms Highly Oxidized RO2 Radicals in the Atmosphere, Angewandte Chemie-International Edition, 53, 14596–14600, doi:10.1002/anie.201408566, 2014.

Kampf, C. J., Jakob, R., and Hoffmann, T.: Identification and characterization of aging products in the glyoxal/ammonium sulfate system - implications for light-absorbing material in atmospheric aerosols, Atmos. Chem. Phys., 12, 6323–6333, doi:10.5194/acp-12-6323-2012, 2012.

Lin, P., Aiona, P. K., Li, Y., Shiraiwa, M., Laskin, J., Nizkorodov, S. A., and Laskin, A.: Molecular Characterization of Brown Carbon in Biomass Burning Aerosol Particles, Environ. Sci. Technol., 50, 11815–11824, doi:10.1021/acs.est.6603024, 2016.

---

## Referee Comment (RC2) · Anonymous Referee #2 · 15 Mar 2018

Schum et al. present a unique dataset collected on Pico Mountain Oberservatory to study the physiochemical properties of aerosol in the remote marine free troposphere. They analyzed three aerosol samples that had elevated organic carbon concentration, and attributed the differences in their molecular and physical characteristics to emission sources as well as their transport pathways. They observed a lower O/C ratio in two samples that they believed were likely from biomass burning plumes that were transported mostly in the free troposphere, and the aerosols were in a solid state that resisted oxidation. Before this work is published in ACP, the authors need to provide careful clarification and further discussion of several important aspects in this manuscript. Please find the comments below.

[Figure]

Major comments

1. The O/C values for PMO-1 and PMO-3 are surprisingly low for particles that had been transported for 7-10 days. In Section 2.3, the authors pointed out that "losses of highly water soluble, low molecular weight organic compounds are expected". Highly water soluble compounds are presumably quite polar and thus should have higher O/C. Authors need to address how the SPE artifacts affect the overall sample O/C. The same issue applies to the artifacts of water extraction that the water-soluble compounds in the samples were preferably collected for the subsequent analysis. Please provide a discussion of possible bias, what is roughly the fraction that had been extracted versus not-extracted, and how it might affect the results of the analysis.

2. The authors use the method developed by DeRieux et al. to estimate particle phase state and heavily rely on the result to explain their findings. However, the authors use this method without further comment and discussion, especially regarding its uncertainty. Solid, semisolid and liquid state are qualitative descriptions which do not provide much insight into diffusion time-scale of water or organic molecules into/out of particles. Diffusion is a key process that determines the evolution of particle composition, and the connection of phase state and diffusivity involves multiple-step estimations with large uncertainties, as shown in a couple of studies [1][2]. Is it possible that the uncertainty of the method is large enough that it changes the major conclusions of this paper? The authors need to provide a much more comprehensive discussion of these issues.

Minor comments

1. In line 20, "This suggests that biomass burning emissions injected into the free troposphere are longer-lived than emissions in the boundary layer." The term "longer-lived" is vaguely used here, as well as in a couple places in the main text. Do the authors mean the particles from biomass burning have lower oxidation state, or the authors are referring to the chemical life time of the compounds from biomass burning that were transported in the free troposphere?

2. In Section 3.1, chloride is presented in Table 1 but not discussed in the main text. Some studies show that biomass burning can produce chlorine-containing particles [3][4].

3. In Figure 1 (c), the air mass spent a couple of days over Europe, and based on (f), the height of the air mass was quite low during those days. Could there be any influence from emissions from Europe on the sample?

4. In Section 3.2, regarding the CO source apportionment in Figure S1, what is the uncertainty associated with the CO modeling?

5. In Section 3.3, line 285-287, 78% of the formulas in PMO-2 are found to be common with sample from the boundary layer aerosol, and PMO-3 has similarity of 76%. Are 78% and 76% significantly different? This piece of information might not be a strong evidence to support the conclusion that PMO-2 was largely influenced by North America outflow transported within the boundary layer while PMO-3 was not.

6. In Figure 2, an obvious difference of the three spectra is the much higher fraction of high molecular weight materials in PMO-2. Little is discussed about the sources of the high molecular weight compounds in the text. Are they from oligomerization? In contrast, Lee et al. [5] observed abundant high molecular weight compounds from biomass burning in Canada using an aerosol mass spectrometer.

7. In section 3.5, line 387, "Volatility can also play a role in the phase state". This expression is vague. Do the authors mean phase state depends on volatility? Or they both relate to structures of molecules in particles? Please make clarification.

8. In section 3.5, line 392, "This highlights the correlation between O/C and volatility, where volatility is expected to decrease as O/C increases." What about fragmentation?

9. Lastly, how generalizable are these findings in the paper in terms of predicting the oxidation state of aerosols having different transport pathways?

References

[1] Frances H Marshall, Rachael EH Miles, Young-Chul Song, Peter B Ohm, Rory M Power, Jonathan P Reid, and Cari S Dutcher. Diffusion and reactivity in ultraviscous aerosol and the correlation with particle viscosity. Chemical Science, 7(2):1298–1308, 2016.

[2] Lindsay Renbaum-Wolff, James W Grayson, Adam P Bateman, Mikinori Kuwata, Mathieu Sellier, Benjamin J Murray, John E Shilling, Scot T Martin, and Allan K Bertram. Viscosity of alpha-pinene secondary organic material and implications for particle growth and reactivity. Proc Natl Acad Sci U S A, 110(20):8014–9, May 2013.

[3] EJT Levin, GR McMeeking, CM Carrico, LE Mack, SM Kreidenweis, CE Wold, H Moosmuller, WP Arnott, WM Hao, JL Collett, et al. Biomass burning smoke aerosol properties measured during fire laboratory at missoula experiments (FLAME). Journal of Geophysical Research: Atmospheres, 115(D18), 2010.

[4] Xiaoxi Liu, L Gregory Huey, Robert J Yokelson, Vanessa Selimovic, Isobel J Simpson, Markus Muller, Jose L Jimenez, Pedro Campuzano-Jost, Andreas J Beyersdorf, Donald R Blake, et al. Airborne measure- ments of western US wildfire emissions: Comparison with prescribed burning and air quality implications. Journal of Geophysical Research: Atmospheres, 122(11):6108–6129, 2017.

[5] Alex KY Lee, Megan D Willis, Robert M Healy, Jon M Wang, Cheol-Heon Jeong, John C Wenger, Greg J Evans, and Jonathan PD Abbatt. Single-particle characterization of biomass burning organic aerosol (BBOA): evidence for non-uniform mixing of high molecular weight organics and potassium. Atmospheric Chemistry and Physics, 16(9):5561–5572, 2016.

---

## Author Comment (AC1) · 15 Mar 2018

Two of the subplots in Figure 1 were inadvertently misplaced in the original version. The corrected version is provided here.

[Figure]

**Figure 1.** FLEXPART retroplumes for 27-28 June 2013 (PMO-1, (a, d)), 5-6 July 2014 (PMO-2, (b, e)), and 20-21 June 2015 (PMO-3, (c, f): column integrated residence time over the 20-day transport time (a-c) and vertical distribution of the retroplume residence time at given upwind times (d-f). The white labels indicate the approximate locations of the center of the plume for each of the transport days. Residence time is color coded by logarithmic grades representing its ratio to the location of maximal integrated residence time (100%). The black line indicates the mean height of the plume during transport.

---

## Author Comment (AC2) · 29 May 2018

**Author Responses to Reviewer #1**

The *Reviewer comments are black italic font* and the Author responses are blue font.

*The authors describe the analysis of three selected filter samples that were collected within a more comprehensive sampling campaign (a total of 127 filters) at the Pico Mountain Observatory (PMO) on Pico Island / Azores. The samples were chosen because of the high organic carbon (OC) concentration. While major small ions and OC of the three filter samples were measured as well, the focus of the manuscript certainly lies on the analysis using direct infusion electrospray ionization ultra-high resolution mass spectrometry in the negative ion mode ((−)ESI/UHRMS). Differences in the mass spectra are discussed with regard to a back-trajectory analysis. The authors observe signals in one sample (out of three) that exhibits higher O/C ratios compared to the other two samples that likely have undergone a longer atmospheric transport time (and thus aging). The authors argue that the two samples with the lower O/C ratio were transported in the free troposphere (FT) to PMO, and thus the particle phase state during transport was likely solid. They conclude that "biomass burning emissions", which are directly injected by pyro-convection into the FT, "are longer-lived than emissions in the boundary layer".*

We thank the reviewer for his comments. We made several changes to both the main paper and the supplemental information. In particular, we made major revisions to section 3.5.

*General comments:*
*Overall, the manuscript presents results from an atmospheric measurement station that is certainly very well suited for studying aerosol transformation processes during long-range transport. Furthermore, the authors demonstrate the need for ultra-high resolution mass spectrometry techniques when it comes to ambient measurements of particulate matter. However, I have major concerns regarding a weak reasoning that is used as a basis for their conclusions and implications for atmospheric aging (see my point (1) below). Also, the authors remain too speculative in many cases, or even state arguments that are not supported by their figures (see (2)). Furthermore, I have serious technical concerns that might have an effect on the outcome of the ESI/UHRMS analysis (see (3)).*

*My major remarks concerning the above mentioned points:*
*(1) The authors argue that particle phase state is affected by the conditions during atmospheric transport. Their observation of a low O/C in the biomass burning samples (PMO-1 and PMO-3) is reasoned by the phase state of the particles during transport. Although, the authors mention that the PMO-2 sample is originating from another source (from the Eastern United States – dominated by a mix of biogenic and anthropogenic emissions), they argue that the high O/C ratio of this sample is caused by the semi-solid phase state, which allows faster aging during atmospheric transport to PMO. Assuming that the back-trajectory analysis is getting the sources right, the authors don't present a convincing argument why we can use the two different sources (biomass burning organic aerosol (BBOA) vs. anthropogenic/biogenic secondary aerosol (A/BSOA)) as an **identical reference point for the onset of atmospheric aging**!*

We thank the reviewer for this comment. We agree that biomass combustion and anthropogenic/biogenic aerosol are not identical reference points for the onset of atmospheric aging. However, these are the pollution events that arrived at the Pico Mountain Observatory and they arrived with very different transport conditions. To provide context regarding our expectations for oxidation, we cited studies that demonstrate both the rapid oxidation of anthropogenic/biogenic aerosol (Jimenez et al. 2009) and the lower average oxidation of biomass combustion aerosol relative to anthropogenic/biogenic aerosol (Bougiatioti et al., 2014) see also lines 30-35, 377-378.

We further strengthened this point to avoid confusion with the following new text (Lines 307-309):
**"Although North American outflow of anthropogenic secondary organic aerosol is expected to have**

**a higher O/C value compared to the wildfire emissions of biomass burning organic aerosol (e.g., Aiken et al., 2008; Jimenez et al., 2009; Bougiatioti et al., 2014) …"**

We also added the following sentence to the manuscript (lines 504-505), to clarify the role played by emission sources: "**These observations suggest that the transport pathways, in addition to the emission sources, contribute to the observed differences in the organic aerosol oxidation.**"

Regarding the aerosol phase state, we estimated the glass transition temperature (DeRieux et al., 2018) for the identified molecular formulas using the extracted Global Forecast System (GFS) data for the FLEXPART retroplumes. Doing this showed that the majority of species identified in PMO-1 and PMO-3 had $T_g$ values that exceeded the ambient temperature, which suggests solid state during the last 5 days of transport. Considering the length of transport (~7-10 days), these samples demonstrated low O/C and $OS_C$ even compared to biomass burning samples analyzed after less transport (Bougiatioti et al., 2014). As described in several papers (Koop et al., 2011; Shrivastava et al., 2017; Zelenyuk et al., 2017), the solid phase is expected to have decreased susceptibility to oxidation and other chemical processes. For this reason, it was hypothesized that the phase state contributes to ambient observations made in this study.

In contrast to PMO-1 and PMO-3, the estimated $T_g$ values for compounds identified in PMO-2 were often lower than the ambient temperature, indicating a less viscous state and thus an increased susceptibility to oxidation and other chemical processes, relative to PMO-1 and PMO-3. This is consistent with other studies of anthropogenic and biogenic SOA in the planetary boundary layer (Maclean et al., 2017; Ye et al., 2016). Although there are uncertainties in the prediction (DeRieux et al., 2018), PMO-2 organic aerosol species were more likely to be semi-solid/liquid based on the ambient conditions than PMO-1 or PMO-3. Thus, we hypothesized that the higher O/C and oxidation observed in PMO-2 may be due to aqueous phase processing during transport, leading to increased oxidation of atmospheric organic matter (Ervens et al., 2008; Zhao et al., 2014). To support this hypothesis, we examined markers of potential cloud processing. We observed depleted nitrate (Dunlea et al., 2009), elevated sulfate and oxalate (Yu et al. 2005; Sorooshian et al. 2007) in PMO-2 as described in section 3.1.

*In the literature it is now well recognized that these two kinds of organic aerosol (BBOA vs SOA from anthropogenic and biogenic VOCs) are already very different on the molecular level at the time when emitted by their sources or formed in the atmosphere: While BBOA is largely composed of lignin- and cellulose-derived condensed aromatic / polyphenolic structures with low O/C ratios (Lin et al., 2016), numerous studies have shown that the auto-oxidation of (mostly biogenic) VOCs results in highly oxygenated molecules within seconds after the initial attack by an oxidant (Crounse et al., 2013; Ehn et al., 2012; Jokinen et al., 2014). Although it is not yet fully understood what happens to these compounds once they condense, the auto-oxidation mechanism still can explain high aerosol O/C from atmospheric oxidation of VOCs.*

We agree that the two types of organic aerosol (SOA vs. BBOA) are very different on the molecular level. Auto-oxidation as described by Crounse et al. (2013), Ehn et al. (2012), and Jokinen et al. (2014) does increase the O/C, but it also shows clear carbon number preferences associated with the oxidation of terpene precursors. This trend is consistent with our earlier work on condensed SOA where the concept of "auto-oxidation" was described as "oxygen-increasing-reactions" (Kundu et al., 2012). However, in the case of PMO-2, we did not observe carbon number preferences, which would indicate auto-oxidation. While this does not negate the possible influence of auto-oxidation, it does minimize its relative importance for these long range transported aerosol observations. On the other hand, aqueous phase processing as described by Lim et al. (2010) leads to SOA production with a greater array of carbon

numbers; the greater array of carbon numbers matches more closely with our observations of a continuum of carbon numbers from 2 to 33 in PMO-2.

We also compared the molecular composition of PMO-2 to that of cloud water from the Storm Peak Laboratory (Zhao et al., 2014) and Whiteface Mountain (Cook et al., 2017), and fog water from Fresno, California (Mazzoleni et al. 2010). In these comparisons, the common species observed only in cloud water and PMO-2 had higher O/C and oxidation than the species common to other samples. This likely indicates that at least some degree of aqueous phase processing has occurred. The results comparing our study to the Cook et al. (2017) and Mazzoleni et al. (2010) studies are new and have been added to the Supplement of this paper in Table S6. All three samples contain atmospheric water that was impacted by anthropogenic, biogenic, and biomass burning air plumes at one time or another. These results consistently show that high O/C species in PMO-2 are uniquely common with atmospheric organic matter in fog and cloud, indicating the influence of aqueous phase processing. We note that the comparisons of the molecular compositions from Zhao et al. (2014) and Mazzoleni et al. (2010) were more comprehensive because we had more complete datasets. The species common only to PMO-1 and the Cook et al. (2017) dataset also had high O/C, but the two formulas represent only 2% of the formulas used for comparison. On the other hand, the uniquely common formulas between the Cook et al. (2017) dataset and PMO-2 represent over 20% of the formulas used for comparison. As such it seems reasonable to say the same trend holds.

**Table S6. Number of molecular formulas and their average O/C values (unweighted O/C and RA weighted O/C (O/C$_w$)) uniquely common between this study and ambient aqueous organic matter (Mazzoleni et al., 2010; Zhao et al., 2015; Cook et al., 2017). Uniquely common means that the formula is common between only one of the PMO samples and the aqueous organic matter sample. CW indicates cloud water, the numbers in parentheses are the percentage of total formulas.**

| Sample | # Common Formula | O/C | O/C$_w$ |
|---|---|---|---|
| PMO and Fog (Mazzoleni et al., 2010) | | | |
| PMO-1 | 202 (6.4%) | 0.38 | 0.39 |
| PMO-2 | 48 (2.3%) | 0.5 | 0.55 |
| PMO-3 | 11 (0.60%) | 0.29 | 0.29 |
| PMO and CW (Cook et al., 2017) | | | |
| PMO-1 | 2 (0.063%) | 0.82 | 0.82 |
| PMO-2 | 23 (1.1%) | 0.8 | 0.81 |
| PMO-3 | 1 (0.055%) | 0.36 | 0.36 |
| PMO and CW (Zhao et al., 2015) | | | |
| PMO-1 | 197 (6.2%) | 0.42 | 0.42 |
| PMO-2 | 70 (3.3%) | 0.76 | 0.8 |
| PMO-3 | 42 (2.3%) | 0.38 | 0.38 |

In addition to Table S6, the following sentences were added to the manuscript (Lines 483-486):
**"Comparisons of the detailed molecular composition of the PMO samples with studies of cloud**

**(Zhao et al., 2013; Cook et al., 2017) and fog (Mazzoleni et al., 2010) water organic matter indicates that the formulas uniquely common to only PMO-2 and the literature atmospheric water samples have higher O/C consistent with aqueous processing during transport. These results are provided in Fig. S19 and Table S6."**

*My impression is that the authors do not adequately consider or discuss different reasons for their observations and overemphasize the possible link between atmospheric transport and aging efficiency at different altitudes.*

Our previous paper (Dzepina et al., 2015) provided a discussion of a biomass burning plume event intercepted at the PMO. In that paper, we considered several factors, but not phase state. We also did not have any non-biomass burning influenced aerosol samples like PMO-2 for comparison. The new set of samples provides an opportunity for a new perspective. In this work, we examined the meteorological variables from GFS that corresponded to the FLEXPART transport pathways and used the recently developed estimation methods for volatility and phase state to assist in the interpretation of the observations.

Since in Dzepina et al. 2015 we considered photochemical oxidation and fragmentation, it was deemed unnecessary to go over again in detail for this manuscript. For clarity, the following introductory sentences were revised (Lines 100 – 104): **"The low oxidation observed by Dzepina et al. (2015) was attributed to the dominance of persistent aerosol that resisted removal mechanisms, however it is possible that the phase state of the aerosol during transport played a significant role. The increased resistance to photodegradation (Lignell et al., 2014; Hinks et al., 2015) and water diffusivity (Berkemeier et al., 2014) of solid phase organic aerosol provide a basis for this hypothesis."**

*To be clear, I am not saying that aerosol phase state does not change aerosol transformation rates, but to extract this effect from ambient observations, one likely needs to consider a larger set of samples (which the authors apparently have).*

We would like to have a larger dataset to extract this valuable information. Unfortunately, there were several limiting factors that are not readily apparent:

(1) The Pico Mountain Observatory is a research post that was originally designed to be temporary, located on the caldera top of Pico Mountain in the Azores (see also Image 1 in the Supplement). The mountain-top site is one of the highest points for 1500 km in the North Atlantic. The site is only accessible by foot over rugged terrain (Honrath et al., 2004) and has limited infrastructure for aerosol chemical observations (Dzepina et al., 2015). Thus, we conducted a field study with filter collection and conducted off-line analyses.

(2) The field study and filter collection were limited by the meteorology. Thus, the pollution events were primarily influenced by biomass combustion from North America or dust events from Africa. We also lost samples/observations due to occasional nighttime lenticular cloud formation (Dzepina et al., 2015).

(3) The detailed molecular analyses were done with an ultrahigh resolution FT-ICR MS instrument at the Woods Hole Oceanographic Institution (WHOI). The WHOI instrument is made available with user fees. The costs of the instrument and travel limited our analyses to only those samples with adequate OC loading to ensure reliable measurements.

*(2) The authors argue that the aerosol that was captured on the PMO-2 sample travelled at altitudes below 2 km over Eastern U.S. and stayed below 2 km altitude until it reached PMO 2-4 days later (p. 7, l. 251-253). From Figure 1 (e), I cannot see that. For the upwind days 0-5, the mean height of the plume is consistently higher than 2 km. As stated on p.2, l. 65-67, the marine boundary layer (MBL) around PMO ranges between 500 m and 2 km, and thus below the mean height of the plume.*

We thank the reviewer for catching our mistake with Fig. 1. We immediately posted the corrected Fig. 1 in Author Comment #1 during the open discussion of this paper. We also added a few more FLEXPART retroplumes for our sampling periods to the Supplement (Figs S1-S3).

*However, the authors argue that **PMO-2 air masses travelled within the MBL layer** to PMO, explaining high relative humidity and a semi-solid phase state during transport. Another argument against the transport within the MBL is given by the authors, mentioning that PMO-2 does not reveal any chemical signature from the MBL (p. 6, l. 232-233). Furthermore, the mean height in PMO-3 appears even lower than PMO-2 for the last five days.*

We thank the reviewer for this comment. We noted that the MBL influence, as inferred from the estimated amount of sea salt sulfate and presence of methane sulfonic acid (MSA), does not seem to be the major influence on that sample. (lines 259-261). The relatively low influence of the MBL does not negate transport within the MBL, especially in this case, because the transport was very fast. Clarification regarding this point was added to lines 263-265. **"The influence of marine sources supports boundary layer transport. However, the results indicate that marine aerosol is not likely a major component of PMO-2, perhaps because the rate of PMO-2 transport was very fast."**

*(3) The discussed filters were selected because of their high mass loadings of organic carbon (>1 mg OC / quarter filter). After loading the water-soluble (WS) OC extract onto solid phase extraction (SPE) material for purification, the SPE was then eluted by 2 mL of MeCN/H2O and the extract was used for direct infusion. If we assume that half of the OC is WSOC and assume 100% SPE collection efficiency (neglecting losing the small, polar organic compounds), the concentration of the solution for direct injection ESI would be ~0.25 mg/mL. To me, this appears as a huge concentration in which ion source cluster formation (e.g. x-mers of analytes, clusters with solvents or solvent additives, impurities), can become a serious issue. I understand that SPE was done in order to reduce cluster formation with inorganic ions and that a separation technique was apparently not available. However, the authors could have done straight-forward tests to check the extent of cluster formation in these samples by: (1) sample dilution and checks for non-linear reduction of cluster-signal candidates and (2) MS/MS isolation and recording the fragmentation energy of cluster signal candidates.*

Yes, direct infusion of high concentrations can lead to ESI artifacts. We always dilute our samples to the lowest possible level to obtain a stable current during ESI. This has been described in our previous work (Putman et al., 2012). We also note that negative ion ESI is less prone to adduct artifacts. This is mainly because the low molecular weight negative ions are removed by the SPE procedure.

*Although, I understand that MS/MS cannot be done on all ion signals, there are some "suspicious" signals standing out in Figure 2 (the signals > 0.3 rel. abundance) that should have been checked using MS/MS when doing direct injection.*

Regarding the suspicious signals, we have done a thorough study using ultrahigh resolution FT-ICR MS/MS on PMO-1 similar to LeClair et al. (2012). The results of the MS/MS are the subject of a forthcoming manuscript. To answer the reviewer's concern, we re-examined several of the MS/MS

spectra. Overall, the mass spectra show fragmentation patterns consistent with covalently bonded molecules and not ion clusters. On the following pages, the MS/MS spectra for several of the tallest peaks present in PMO-1 are shown. Note that we cannot isolate 1 and only 1 m/z value. Instead we isolate a range of values (for example 230 ± 3). In this way, the ions of the entire isolated group are simultaneously fragmented.

[Figure]

P062713w_230w6_100k_CID #1-55   RT: 0.00-3.16   AV: 55   NL: 9.18E4
T: FTMS - p ESI Full ms2 230.00@cid33.00 [60.00-240.00]

[Figure]

P062713w_245w6_100k_CID #1-55   RT: 0.01-3.10   AV: 55   NL: 1.11E4
T: FTMS - p ESI Full ms2 245.00@cid32.00 [65.00-255.00]

P062713w_270w6_100k_CID #1-54   RT: 0.01-2.47   AV: 54   NL: 1.10E5
T: FTMS - p ESI Full ms2 270.00@cid35.00 [70.00-280.00]

[Figure]

P062713w_300w6_200k_CID #1-54   RT: 0.02-3.31   AV: 54   NL: 4.85E3
T: FTMS - p ESI Full ms2 300.00@cid38.00 [80.00-310.00]

[Figure]

*Ion source cluster formation would introduce a bias on the calculated glass transition temperature (Tg) by artificially increasing the average number of carbon per molecule. Furthermore, the overall Tg is already biased toward higher values since small molecules are very likely lost during the SPE procedure (l. 158-159). The manuscript misses in its current form a critical evaluation of these points and its implications on aerosol phase state and aging.*

As shown above, we do not have evidence for ion clustering. However, the lower MW species can be quite important for the average organic aerosol $T_g$. We chose to address this in two ways, first we report the distribution of $T_g$ values in Figure 7 and second, we estimated the influence of the known low MW compounds. Using the concentrations of organic acids (e.g., Table SM1), we found that the relative abundance weighted $T_g$ for each of the samples changed by $\leq 2.5\%$ when the abundance of the organic acids was considered (Table SM5). This was done by estimating the $T_g$ for each of the following organic acids (formic, acetic, oxalic, malonic, and lactic) using the Boyer-Kauzmann rule ($T_g = g* T_m$, $g = 0.7$, $T_m$ = melting temperature) (Shiraiwa et al. 2017; DeRieux et al. 2018). Oxalic acid is by far the most abundant of all organic acids in these samples and thus has the largest impact. The percentage of total organic mass that each acid made up was calculated by dividing their concentration by the concentration of organic mass, which was estimated by multiplying the OC value by 2 (El-Zanan et al., 2005). Then the individual low MW compound mass fractions were used to estimate their abundance relative to the sum of the total abundance of species identified by FT-ICR MS using the worst-case scenario assumption that the detected species in the mass spectra represent only 50 % of the total organic mass. These abundance values, along with the estimated $T_g$ values were then used to calculate the abundance weighted average dry $T_g$ for each sample. When compared to the original weighted $T_g$ values, the difference was $\leq 2.5\%$ for all samples, indicating that while the organic acids do impact the $T_g$ to some extent, the impact is not so significant as to change any of the conclusions of this study. A series of tables showing the values described here are provided in the Supplement (Tables SM1-SM5).

**Table SM1. The concentrations of the ions used for the estimation and the organic mass (OM) concentration. The values are in μg/m$^3$ air.**

| Ion | PMO-1 | PMO-2 | PMO-3 |
|---|---|---|---|
| Formate | 0.0289 ± 0.0003 | 0.00438 ± 0.00007 | 0.0119± 0.0001 |
| Acetate | 0.0519 ± 0.0001 | 0.004587 ± 0.000005 | 0.0071 ± 0.0002 |
| Oxalate | 0.0938 ± 0.00070 | 0.0897 ± 0.00181 | 0.0522 ± 0.00002 |
| Malonate | 0.00605 ± 0.0003 | 0.00548 ± 0.0007 | 0.0045± 0.0003 |
| Lactate | 0.0292 ± 0.0004 | 0.0019 ± 0.0001 | 0.00467 ± 0.0001 |
| OM | 4.14 ± 0.04 | 0.956 ± 0.052 | 1.74 ± 0.20 |

**Table SM5. Estimated average $T_g$ values when the ions are considered. The table contains the results for 3 assumptions of the organic mass fraction represented by the FT-ICR MS identified species (100%, 70%, 50%). The numbers in parentheses show the percent change in average $T_g$ from the $T_g$ without ions considered. All $T_g$ values are in K.**

| Sample | RA Weighted $T_g$ without Ions (100%) | Ions and RA Weighted $T_g$ (100%) | Ions and RA Weighted $T_g$ (70%) | Ions and RA Weighted $T_g$ (50%) |
|---|---|---|---|---|
| PMO-1 | 328.75 | 324.38 (1.33%) | 322.67 (1.85%) | 320.51 (2.51%) |
| PMO-2 | 326.45 | 324.43 (0.619%) | 323.71 (0.839%) | 322.85 (1.10%) |
| PMO-3 | 326.88 | 324.41 (0.756%) | 323.44 (1.05%) | 322.22 (1.43%) |

A comment about the estimated impact of small organic acids to the $T_g$ has been added to the manuscript as follows (lines: 164-171): **"The procedural loss of ionic low MW compounds such as oxalate can lead to an underprediction of the organic aerosol O/C and overprediction of the average glass transition temperatures ($T_g$). To investigate this, we used the concentrations of the prominent organic anions measured with ion chromatography to estimate the abundance of these compound relative to the compounds detected by FT-ICR MS. The low MW corrected average O/C values correlated with the trends of the original O/C values, however the significance of impacts varies with the measured analyte concentrations and the assumptions associated with the uncertain mass fraction of the molecular formula composition (Table SM4). When low MW organic anions were included in the estimated average dry $T_g$ values, they dropped by $\leq 2.5$ %, which was deemed relatively insignificant (Table SM5).**

In the supplement, a discussion of the estimation method was added (pg. 5). In addition, the tables shown above were added to the supplement as well.

*Finally, it would have been interesting to measure also in the positive ESI mode, in which one can observe levoglucosan or nitrogen-heterocycles that are expected in biomass burning aerosol samples.*

We agree with the reviewer, analyses with positive mode ESI would have been interesting. Time constraints limited our ability to do multiple ionization method and we were interested in the oxidation characteristics of long-range transported aerosol. Since oxidation leads to the addition of carboxylic acid groups and in general more polar molecular species, negative mode ESI was the most practical way to analyze the samples.

*Moreover, it is not clear how the blank signal of the DI/ESI-UHRMS was determined. I would expect a measurement of a blank filer that undergoes the whole procedure incl. transport from the field site, sample preparation procedure in laboratory (sonication, filtration, SPE, etc). Only a good blank measurement allows determining the significance level at which individual signals are present in the samples and identifying those signals that emerge from sample preparation. Here, it is especially important since the paper discusses the number of identified compounds between the different samples. If a compound was identified as "not present" in one sample, does that mean after blank subtraction? It is not described what were procedures involving a blank filter, nor are mentioned the criteria and the thresholds for this kind of filtering!*

Blank subtraction is a non-trivial matter in ultrahigh resolution MS (e.g., Zielinski et al., 2018) due to the difference in ion collection times for a sample compared to a blank, and the possibility of resuspension of

sample residues within the instrument when a blank (i.e., clean solvent) is infused. Ion trap instruments (including hybrid FT-ICR MS and Orbitrap MS instruments) also use an auto-gain control (AGC) to avoid space charge artifacts. In our analysis, the AGC was set at the recommended setting of $1 \times 10^6$ ions. Since samples and blanks generate ions at very different rates, the time necessary for the analysis varies and often a maximum injection period is required. Our maximum inject period was 500 ms for samples and 800 ms for blanks. The actual average injections times for samples were in the range of 20-80 ms, however the blanks "timed-out" at 800 ms before the mass analysis was performed. The injection time differences indicated that the blanks were very clean and the potential for resuspension was non-negligible due to increased accumulation time. Therefore, we compared the intensities of the analytes in the samples and blanks and used a ratio of 3 to determine whether or not a peak should be removed.

In this study, we had both technical instrument blanks and field blanks. We applied the ratio of 3 criteria to both types of blanks, where all of the analytes with a ratio < 3 relative to the technical blank were removed and those < 3 relative to the field blanks were flagged. This led to 2 formulas being flagged because they didn't meet the criteria in 1 of the 3 samples. If the 2 analytes were contamination, they should have appeared equally in all 3 samples, but they were not. Further the 2 analytes were part of a homologous series that was not otherwise in common with the field blanks. The two flagged analytes are $C_{17}H_{34}O_4$ and $C_{19}H_{38}O_4$, which showed very low intensity in that sample. We deemed this to be a fair assessment, especially in light of the very different amounts of time required for the ion injection.

The QA that was performed for the samples is consistent with what has been described in other studies from our group (Putman et al., 2012; Mazzoleni et al., 2012; Dzepina et al., 2015). In short, we removed formulas with extremely high or low O/C (>2, <0.1), H/C (>2.2, <0.3), and DBE (>20). We also removed known solvent contaminant peaks and isolated assignments that were not part of a $CH_2$ homologous series. After this was done, we aligned the two replicate analyses of the samples and kept only the formulas that were present in both replicates. If a formula is described as "not present" it means that formula was not present in the sample being referred to after the QA steps described above were performed. This description of the QA procedure was added to the Supplement of the manuscript. To the main manuscript we added the following (lines: 185-188): **"Specifically, two formulas ($C_{17}H_{34}O_4$ and $C_{19}H_{38}O_4$) observed in PMO-1 could not be classified as pertaining only to the field blank and so they were not removed. Further discussion about the blank subtraction is provided in the Supplement. To produce the final data set for each sample, the replicates were aligned and only the molecular formulas found in both replicates after QA were retained. "**

The description of the blank subtraction procedure given above will be added to the Supplement pg. 4.

*Overall, I cannot recommend the article to be published in ACP, since the conclusions reached remain far too speculative and are not convincingly supported by the presented data. I miss a more critical discussion and evaluation of other potentially important processes (both atmospheric and instrumental) throughout the manuscript. The description of the mass spectrometry analysis is not sufficiently complete and leaves the reader with open questions (e.g. What was the workflow of the data analysis? What did they use as blank samples?). Last but not least, the presentation and language is in many cases not precise.*

Based on the combination of reviewer and editor impressions, we substantially revised the discussion in section 3.5 regarding the observed organic aerosol and its glass transition temperatures ($T_g$). This aspect of the paper is one that is especially unique because we pulled out the GFS ambient conditions along the

FLEXPART retroplume to consider the role of the ambient conditions on the observed chemistry. We paired this with a discussion of the markers of aqueous phase chemistry.

To avoid unnecessary length, our original paper referenced several of our previous papers regarding instrumental method details. However, the reviewers have raised a few interesting questions that we answered more directly in the revised manuscript and the corresponding supplement.

A key point that is especially important to keep in mind is that very little research has been done on the chemistry of free tropospheric aerosol as opposed to the more extensive knowledge of aerosol chemistry from within the continental boundary layer. Our detailed analysis contributes much needed insight to the chemistry of free tropospheric aerosol, where the ambient conditions are colder and drier than in the boundary layer.

The manuscript has been edited for grammar corrections and clarity.

*Specific comments*
*p.3, l. 106-109: Please provide a reference stating that long-range transported aerosol is generally acidic in nature. Furthermore, negative ESI is not only sensitive to organic acids, but also to important biomass burning markers (e.g. nitro-phenols (Iinuma et al., 2010)). Have you seen nitro-phenols or similar biomass burning tracers in the biomass burning samples?*

Long range transported aerosol generally has an acidic nature as mentioned in a study by Bougiatioti et al. (2016). Additionally, it is known that during oxidation, carboxylic acid groups will be formed in organic aerosol (Iinuma et al., 2004), providing additional evidence that aged aerosol is generally acidic in nature. The following reference has been added to the manuscript for the citation of acidic transported aerosol at line 112: **(Bougiatioti et al., 2016)**

To clarify our use of negative mode ESI and the reasons for it, the following has been added to lines 175-176 of the manuscript: **"Negative polarity is effective for the deprotonation of polar organic molecules (Mazzoleni et al., 2010), which are expected to dominate the organic aerosol mass fraction and were the focus of this study."**

Formulas such as $C_6H_5NO_3$ (nitro-phenol) and $C_6H_{10}O_5$ (levoglucosan) were detected in the samples. Additionally, all but three of the CHNO and CHO negative mode ESI species connected to brown carbon by Lin et al. (2016) were detected in one or more of the aerosol samples in this study. PMO-1 contained all of them, which supports the biomass combustion source for this sample in particular. For a list of matching formulas see Table S4. This table was added to the Supplement.

**Table S4. Molecular formulas identified in brown carbon by Iinuma et al. 2010 and Lin et al. 2016.**

| Formula | Observed | Citation |
|---|---|---|
| | | Iinuma et al. 2010; |
| $C_7H_7NO_4$ | Yes | Lin et al. 2016 |
| $C_6H_5NO_3$ | Yes | Lin et al. 2016 |
| $C_6H_5NO_4$ | Yes | Lin et al. 2016 |
| $C_6H_6N_2O_6$ | No | Lin et al. 2016 |
| $C_6H_4NO_4$ | No | Lin et al. 2016 |
| $C_{10}H_9NO_3$ | No | Lin et al. 2016 |
| $C_8H_7NO_4$ | Yes | Lin et al. 2016 |
| $C_8H_7NO_3$ | Yes | Lin et al. 2016 |
| $C_9H_7NO_4$ | Yes | Lin et al. 2016 |
| $C_{10}H_7NO_4$ | Yes | Lin et al. 2016 |
| $C_8H_8O_3$ | Yes | Lin et al. 2016 |
| $C_9H_6O_3$ | Yes | Lin et al. 2016 |
| $C_{10}H_8O_4$ | Yes | Lin et al. 2016 |
| $C_{13}H_8O_5$ | Yes | Lin et al. 2016 |
| $C_{13}H_8O_6$ | Yes | Lin et al. 2016 |
| $C_{15}H_{10}O_6$ | Yes | Lin et al. 2016 |
| $C_{16}H_{12}O_6$ | Yes | Lin et al. 2016 |
| $C_{16}H_{12}O_7$ | Yes | Lin et al. 2016 |
| $C_{17}H_{14}O_8$ | Yes | Lin et al. 2016 |

*p. 5, l. 169: The two references describe different criteria for the molecular formula assignments: Dzepina et al.: max. 100 C, 400 H, 100 O, 3 N, and 1 S. Mazzoleni et al.: max. 70 C, 140 H, 25 O, 3 N, and 1 S.*

The maximum range for C, H, and O is a function of the molecular weight range. Our highest m/z value is 752.3636, thus assuming 100% C (752.3636/12 = 62.7), 50% O (752.3636/32 = 23.5) and an H/C = 2 (62.7 *2 = 125.4). Thus, it is not necessary to have higher maximum values.

*The elemental windows for nitrogen and sulfur seem very strict. The used limits exclude for example the identification of nitrogen-heterocycles with four nitrogen (e.g Kampf et al., 2012). Given the clear isotopic signature of sulfur, why was not more sulfur allowed? Were the isotopic patterns used to confirm the molecular formulas in case multiple elemental compositions appeared within the instrumental accuracy limits?*

Based on our observations over several iterations of molecular formula assignment with varied elemental tolerances, the number of unreliable (aka chemically unreasonable) molecular formula assignments increases with an increased number of N and S.

Isotopic patterns were used to provide confidence in the molecular formula assignment. Roughly 90% of all species identified were found to have a corresponding $^{13}C$ peak and roughly 70% of all sulfur containing formulas were found to have a corresponding $^{34}S$ peak. The molecular formulas without isotope confirmation had low relative abundances, thus the polyisotopic ions were likely below the noise threshold.

Reduced S and N (including heterocyclic compounds) are unlikely to be detected in negative ion ESI-MS. Furthermore, several studies have shown that the number of elements and especially the number of multivalent elements must be restricted to obtain reliable molecular formula assignment (Koch et al., 2007; Herzsprung et al., 2015). On the other hand, not allowing probable heteroatoms leads to incorrect assignments (Ohno et al., 2013).

*p. 5, l. 172: According to Putman et al., allowing nitrogen for compounds larger than 500 amu results in multiple results within 1 ppm. Does that mean that the number of elements allowed was chosen such strict that only one molecular formula per signal was obtained?*

In the Putman et al. (2012) paper we discussed the importance of having a *de novo* (aka first in series) cutoff of 500 u when assigning molecular formulas with heteroatoms such as nitrogen. This means that the molecular formula assignments above 500 u are restricted to homologous series of molecular formulas below 500 u. In Composer this relationship is based on Kendrick mass defects to identify homologous series of $CH_2$. This is necessary because the number of chemically reasonable formulas with N and S heteroatoms is greater than 1.

*p. 6, l. 205-222: As mentioned by the editor, this paragraph is not well structured and needs rewriting.*

Upon receipt of the editor's comments, we revised the paragraph from a single long paragraph to 3 shorter paragraphs. To further clarify the paragraph, we removed the following sentence (from Section 3.1) as it was slightly off topic: "**Generally, increased cloud processing is expected to lead to increased oxidation of atmospheric organic species (Ervens et al., 2008; Zhao et al., 2013), but has also been hypothesized that cloud scavenging of oxidized components could lead to lower overall oxidation by leaving behind reduced aerosol (Dzepina et al., 2015).**"

*p. 8, l. 308: Is the high O/C of the CHNO species potentially driven by organic nitrates?*

Yes, we expect the oxygen of nitrate functional groups to contribute to the O/C value. This is why the O/C ratios of CHO and CHNO species were never directly compared and contrasted within a single sample, instead they were only compared across the samples.

*p. 8, l. 310-311: Does the common number of identified molecules in a certain group (here CHNO) really tell us something about the similarity of samples? This is mentioned several times in the manuscript, and I don't understand why intensity of compounds is not given more weight in the discussions.*

The peak intensity is not the only consideration for these compounds, because intensity is not based entirely on the abundance of the compound in the sample. Ionization efficiency also plays an important role (this is mentioned in lines 324-326). For this reason, we tried to limit our reliance on interpreting the samples solely through abundance. However, because the ions represent a mixture of isomers, the trends associated with the molecular formulas are important. For example, although a majority of the CHO molecular formulas between the three samples were in common, we observed much higher normalized relative abundances of the CHO with higher O/C values in PMO-2. It is the same for CHNO species as well, the species with high O/C are more abundant in PMO-2 than in the other two samples.

To eliminate confusion, we moved the non-intensity weighted values to the supplement.

*Why not visualizing the similarity of two different samples by simple scatter plots of the intensity of all ions of sample A vs intensity of all ions of sample B. This would also allow determining the Pearson correlation coefficient.*

As requested, we made a scatter plot (see below). The scatter plots are consistent with our observations regarding the similarities between PMO-1 and PMO-3 and the relative difference between these two and PMO-2. The Pearson coefficients are shown in each plot and demonstrate PMO-1 and PMO-3 are more similar to each other than PMO-2.

[Figure]

Figure AR1. Scatter plot of the normalized relative abundance of one sample vs. the normalized relative abundance of another. The sample name is on each axis and the numbers denote the normalized relative abundance.

Another way of demonstrating the differences and similarities between these samples while considering their abundance can be done using difference mass spectra. We added 3 difference mass spectra to the Supplement (Fig. S23). In these plots, PMO-2 contains nearly all of the high O/C species, while PMO-1 and/or PMO-3 show the lower O/C species, as described in the manuscript. The plots also show where the species are more abundant. Likewise, similar abundances cancel each other toward zero.

[Figure]

**Figure S23. Difference mass spectra comparing the three PMO samples. The species more abundant in one sample or another are elevated in the correspondingly labeled half of the plot. PMO-1 vs. PMO-2 (a), PMO-1 vs. PMO-3 (b), and PMO-2 vs. PMO-3 (c).**

*p. 8, l. 319: Is the high O/C of the CHOS species potentially driven by organic sulfates? What would be the O/C after accounting for organic sulfates? Is it then still different from the CHO value?*

Yes, the O/C of the CHOS species is impacted by the presence of organic sulfates. If 4 oxygen are removed from the molecular formulas and the O/C is recalculated the O/C decreases to a level somewhat below that of the CHO group (O/C = 0.44 for PMO-2 and O/C = 0.27 for PMO-1 when "sulfate" is removed). This is why the O/C values of the CHOS compounds are not directly compared to the other groups.

*p. 9, l. 334 ff.: The oxygen that originates from organic nitrates and sulfates artificially increases the oxidation state of carbon.*

Yes, this is correct, and in Fig. 4 (manuscript) the $OS_C$ values used were calculated using the assumption that nitrogen and sulfur were both fully oxidized (Kroll et al., 2011). The average values reported in the tables calculated without this assumption were corrected. To clarify this, a sentence was added to lines 370-371: **"Additionally, we assumed all nitrogen and sulfur were present as nitrate and sulfate functional groups and calculated the $OS_C$ with the appropriate corrections (Equation S1)."**

*p. 10, l. 394: Was the glass transition temperature determined only for single molecules and not for the intensity weighted population of all ion signals? Obviously the atmospheric particles are mixtures and therefore only the glass transition temperature derived from the whole spectrum is meaningful.*

We thank the reviewer for this question. A similar question was raised by Reviewer 2. Prompting much additional consideration of this topic. Please see also the Authors Response to Reviewer 2.

First, we calculated the $T_g$ for all components because it provides the distribution of $T_g$ values. This can be useful because it is unlikely that all particles contain all the species identified (e.g. Riemer and West, 2013), and so some particles may contain species that are more likely to be liquid, while another may be more likely solid. A determination of the chemical mixing at the single particle level would require a different type of analysis (e.g., O'Brien et al., 2015) and is not possible with our samples. The single compound information is lost when all species are treated as if they were uniformly mixed by calculating a single average $T_g$.

We agree that the overall $T_g$ for a particle is what matters in determining the phase state of said particle, but as we have no way of knowing the exact composition of every aerosol particle we feel that showing the distribution of estimated $T_g$ values is more appropriate. In any case, we did calculate the compositional arithmetic average dry $T_g$ values for all three samples (RA Weighted $T_g$ without Ions (100%)), which is shown in the table below. This table has also been added to the Supplement as Table S5.

**Table SM5. Estimated average $T_g$ values when the ions are considered. The table contains the results for 3 assumptions of the organic mass fraction represented by the FT-ICR MS identified species (100%, 70%, 50%). The numbers in parentheses show the percent change in average $T_g$ from the $T_g$ without ions considered. All $T_g$ values are in K.**

| Sample | RA Weighted $T_g$ without Ions (100%) | Ions and RA Weighted $T_g$ (100%) | Ions and RA Weighted $T_g$ (70%) | Ions and RA Weighted $T_g$ (50%) |
|--------|------------------|------------------|------------------|------------------|
| PMO-1 | 328.75 | 324.38 (1.33%) | 322.67 (1.85%) | 320.51 (2.51%) |
| PMO-2 | 326.45 | 324.43 (0.619%) | 323.71 (0.839%) | 322.85 (1.10%) |
| PMO-3 | 326.88 | 324.41 (0.756%) | 323.44 (1.05%) | 322.22 (1.43%) |

*Accounting for the fact that small organics are lost during the SPE (which would reduce Tg of the mixture), what would be the effect on Tg of the mixture if one assumes that 10, 20 or 50% of total OC consist of small organics?*

It is correct that low MW species are lost during SPE and this can bias the $T_g$ towards higher values. However, the bias is very minor based on estimations made using the ion concentrations and OM concentrations for these samples, as discussed in the response to reviewer comment #3 (Table SM5). The most prevalent organic anion in all three samples, is oxalate, which has a $T_g$ estimated to be 324.21 according to the Boyer-Kauzmann rule ($T_g = g*T_m$, g= 0.7, $T_m$ = melting point) (Koop et al., 2011; Shiraiwa et al., 2017; DeRieux et al., 2018). If the assumption is made that half of the total OM is small organics, oxalate would likely make up the largest percentage of the total and would thus have the

greatest effect on the impact of small organics on the $T_g$. Just as a theoretical exercise, if we use the assumption that 50% of the OM is oxalate and the method described in detail above, the average dry $T_g$ drops ≤ 2.5 K. It would be lower if we assumed that formate or acetate were the major small organic being lost, but based on our measurements that is not expected.

As explained earlier, our intention was not to definitively calculate the glass transition temperature for the entire mixture in the particle. To clarify this we altered Figure 7 in the manuscript to instead show the distribution of estimated $T_g$ values based on the ambient relative humidity. In this plot, we included the $T_g$ and abundance of the most abundant low MW organic ions removed by SPE. This sheds light on their potential impact toward the overall $T_g$ of the particles. We also plotted the ambient temperature to guide the eye and illustrate where the estimated $T_g$ values exceed the ambient temperature, which implies a greater likelihood for solid state aerosol (Shiraiwa et al., 2017). Furthermore, the text of the manuscript has been revised to reflect these changes from discussing phase state predictions to estimates of $T_g$ and their implications.

*Figure 3: The three dots with H/C<0.6 are cyan in PMO-1, which should mean unique in PMO-1. Why are these three signals in PMO-2 and PMO-3 then grey (common signals)?*

We thank the reviewer for the comment, the figure caption was inaccurate, and the grey symbols were actually all CHNO formulas detected in the three samples. They were included in each plot to make it clear where the differences are between the three samples. The colored circles are the formulas that are unique to each sample, as stated in the figure caption. We clarified the wording by replacing **"common species"** with **"all identified CHNO species"**.

*Figure 1: The Figure quality is not appropriate.*

Corrected.

*Figure 2: X-axis ticks are missing for (a),(b),(d) and (e). It seems as the highest peaks in (b) and (c) are cut at the top, or are they all the same height?*

The tick marks on the bottom plot are accurate for all plots. The tallest peaks are cut off at the top in order to better show the lower intensity species. A comment to clarify this has been added to the caption for Figure 2: **"The tallest peaks in the mass spectra exceed the range, this was done to improve the visibility of the lower abundance species (see also Fig. S6)."** In addition, a plot that shows the entire range of abundance values was added to the Supplement (Fig. S6).

*Figure 4: X-axis ticks are missing for (a), (b), (d) and (e).*

Corrected

*Figure 4: The y-label is number of formulas. The caption is not clear on that. What is the "normalized abundance"? On what is it normalized?*

Thank you for catching this oversight. Instead of normalized abundance, it should have been number of formulas. However, we decided to use the histogram based on the normalized abundance instead of the one that was there, so the caption is now correct as written, and no additional changes are needed. The abundance is normalized to the total abundance of the assigned species.

*Figure 6: X-axis ticks are missing for PMO-1.*

Corrected

*Literature*

Crounse, J. D., Nielsen, L. B., Jorgensen, S., Kjaergaard, H. G., and Wennberg, P. O.: Autoxidation of Organic Compounds in the Atmosphere, Journal of Physical Chemistry Letters, 4, 3513–3520, doi:10.1021/jz4019207, 2013.

Ehn, M., Kleist, E., Junninen, H., Petaja, T., Lonn, G., Schobesberger, S., Dal Maso, M., Trimborn, A., Kulmala, M., Worsnop, D. R., Wahner, A., Wildt, J., and Mentel, T. F.: Gas phase formation of extremely oxidized pinene reaction products in chamber and ambient air, Atmos. Chem. Phys., 12, 5113–5127, doi:10.5194/acp-12-5113-2012, 2012.

Iinuma, Y., Boege, O., Graefe, R., and Herrmann, H.: Methyl-Nitrocatechols: Atmospheric Tracer Compounds for Biomass Burning Secondary Organic Aerosols, Environ. Sci. Technol., 44, 8453–8459, doi:10.1021/es102938a, 2010.

Jokinen, T., Sipila, M., Richters, S., Kerminen, V.-M., Paasonen, P., Stratmann, F., Worsnop, D., Kulmala, M., Ehn, M., Herrmann, H., and Berndt, T.: Rapid Autoxidation Forms Highly Oxidized RO2 Radicals in the Atmosphere, Angewandte Chemie-International Edition, 53, 14596–14600, doi:10.1002/anie.201408566, 2014.

Kampf, C. J., Jakob, R., and Hoffmann, T.: Identification and characterization of aging products in the glyoxal/ammonium sulfate system - implications for light-absorbing material in atmospheric aerosols, Atmos. Chem. Phys., 12, 6323–6333, doi:10.5194/acp-12-6323-2012, 2012.

Lin, P., Aiona, P. K., Li, Y., Shiraiwa, M., Laskin, J., Nizkorodov, S. A., and Laskin, A.: Molecular Characterization of Brown Carbon in Biomass Burning Aerosol Particles, Environ. Sci. Technol., 50, 11815–11824, doi:10.1021/acs.est.6603024, 2016.

**Additional References**

Aiken, A. C., DeCarlo, P. F., Kroll, J. H., Worsnop, D. R., Huffman, J. A., Docherty, K. S., Ulbrich, I. M., Mohr, C., Kimmel, J. R., Sueper, D., Sun, Y., Zhang, Q., Trimborn, A., Northway, M., Ziemann, P. J., Canagaratna, M. R., Onasch, T. B., Alfarra, M. R., Prevot, A. S. H., Dommen, J., Duplissy, Metzger, A., Baltensperger, U., and Jimenez J. L.: O/C and OM/OC Ratios of Primary, Secondary, and Ambient Organic Aerosols with High-Resolution Time-of-Flight Aerosol Mass Spectrometry, Environ. Sci. Technol., 42(12), 4478–4485, doi:10.1021/es703009q, 2008.

Bougiatioti, A., Stavroulas, I., Kostenidou, E., Zarmpas, P., Theodosi, C., Kouvarakis, G., Canonaco, F., Prevot, A. S. H., Nenes, A., Pandis, S. N. and Mihalopoulos, N.: Processing of biomass-burning aerosol in the eastern Mediterranean during summertime, Atmos. Chem. Phys., 14(9), 4793–4807, doi:10.5194/acp-14-4793-2014, 2014.

Bougiatioti, A., Nikolaou, P., Stavroulas, I., Kouvarakis, G., Weber, R., Nenes, A., Kanakidou, M. and Mihalopoulos, N.: Particle water and pH in the eastern Mediterranean: source variability and implications for nutrient availability, Atmos. Chem. Phys., 16(7), 4579–4591, doi:10.5194/acp-16-4579-2016, 2016.

Cook, R. D., Lin, Y.-H., Peng, Z., Boone, E., Chu, R. K., Dukett, J. E., Gunsch, M. J., Zhang, W., Tolic, N., Laskin, A. and Pratt, K. A.: Biogenic, urban, and wildfire influences on the molecular composition of dissolved organic compounds in cloud water, Atmos. Chem. Phys., 17(24), 15167–15180, doi:10.5194/acp-17-15167-2017, 2017.

DeRieux, W.-S. W., Li, Y., Lin, P., Laskin, J., Laskin, A., Bertram, A. K., Nizkorodov, S. A. and Shiraiwa, M.: Predicting the glass transition temperature and viscosity of secondary organic

material using molecular composition, Atmos. Chem. Phys., 18(9), 6331–6351, doi:10.5194/acp-18-6331-2018, 2018.

Dunlea, E. J., DeCarlo, P. F., Aiken, A. C., Kimmel, J. R., Peltier, R. E., Weber, R. J., Tomlinson, J., Collins, D. R., Shinozuka, Y., McNaughton, C. S., Howell, S. G., Clarke, A. D., Emmons, L. K., Apel, E. C., Pfister, G. G., van Donkelaar, A., Martin, R. V., Millet, D. B., Heald, C. L. and Jimenez, J. L.: Evolution of Asian aerosols during transpacific transport in INTEX-B, Atmos. Chem. Phys., 9(19), 7257–7287, doi:10.5194/acp-9-7257-2009, 2009.

Dzepina, K., Mazzoleni, C., Fialho, P., China, S., Zhang, B., Owen, R. C., Helmig, D., Hueber, J., Kumar, S., Perlinger, J. A., Kramer, L. J., Dziobak, M. P., Ampadu, M. T., Olsen, S., Wuebbles, D. J., and Mazzoleni, L. R.: Molecular characterization of free tropospheric aerosol collected at the Pico Mountain Observatory: a case study with a long-range transported biomass burning plume, Atmos. Chem. Phys., 15(9), 5047–5068, doi:10.5194/acp-15-5047-2015, 2015.

El-Zanan, H. S., Lowenthal, D. H., Zielinska, B., Chow, J. C. and Kumar, N.: Determination of the organic aerosol mass to organic carbon ratio in IMPROVE samples, Chemosphere, 60(4), 485–496, doi:10.1016/j.chemosphere.2005.01.005, 2005.

Ervens, B., Carlton, A. G., Turpin B. J., Altieri, K. E., Kreidenweis, S. M., and Feingold, G.: Secondary organic aerosol yields from cloud- processing of isoprene oxidation products, Geophy. Res. Lett., 35(2), doi:10.1029/2007gl031828, 2008.

Herzsprung, P., Tümpling, W. v, Hertkorn, N., Harir, M., Friese, K. and Schmitt-Kopplin, P.: High-Field FTICR-MS Data Evaluation of Natural Organic Matter: Are CHON5S2Molecular Class Formulas Assigned to13C Isotopicm/zand in Reality CHO Components?, Analytical Chemistry, 87(19), 9563–9566, doi:10.1021/acs.analchem.5b02549, 2015.

Honrath, R., Owen, R., Martín, M., Reid, J., Lapina, K., Fialho, P., Dziobak, M., Kleissl, J. and Westphal, D.: Regional and hemispheric impacts of anthropogenic and biomass burning emissions on summertime CO and O3 in the North Atlantic lower free troposphere, J. of Geophys. Res. Atmos., 109(D24), doi:10.1029/2004jd005147, 2004.

Iinuma, Y., Böge, O., Gnauk, T. and Herrmann, H.: Aerosol-chamber study of the α-pinene/O3 reaction: influence of particle acidity on aerosol yields and products, Atmos. Environ., 38(5), 761–773, doi:10.1016/j.atmosenv.2003.10.015, 2004.

Jimenez, J. L., Canagaratna, M. R., Donahue, N. M., Prevot, A. S. H., Zhang, Q., Kroll, J. H., DeCarlo, P. F., Allan, J. D., Coe, H., Ng, N. L., Aiken, A. C., Docherty, K. S., Ulbrich, I. M., Grieshop, A. P., Robinson, A. L., Duplissy, J., Smith, J. D., Wilson, K. R., Lanz, V. A., Hueglin, C., Sun, Y, L., Tian, J., Laaksonen, A., Raatikainen, T., Rautiainen, J., Vaattovaara, P., Ehn, M., Kulmala, M., Tomlinson, J. M., Collins, D. R., Cubison, M. J., Dunlea, E. J., Huffman, J. A., Onasch, T. B., Alfarra, M. R., Williams, P. I., Bower, K., Kondo, Y., Schneider, J., Drewnick, F., Borrmann, S., Weimer, S., Demerjian, K., Salcedo, D., Cottrell, L., Griffin, R., Takami, A., Miyoshi, T., Hatakeyama, S., Shimono, A., Sun, J. Y., Zhang, Y. M., Dzepina, K., Kimmel, J. R., Sueper, D., Jayne, J. T., Herndon, S. C., Trimborn, A. M., Williams, L. R., Wood, E. C., Middlebrook, A. M., Kolb, C. E., Baltensperger U. and Worsnop D. R.: Evolution of Organic Aerosols in the Atmosphere, Science, 326(5959), 1525–1529, doi:10.1126/science.1180353, 2009.

Koop, T., Bookhold, J., Shiraiwa, M. and Pöschl, U.: Glass transition and phase state of organic compounds: dependency on molecular properties and implications for secondary organic aerosols in the atmosphere, Phys. Chem. Chem. Phys., 13(43), 19238–19255, doi:10.1039/c1cp22617g, 2011.

Kroll, J. H., Donahue, N. M., Jimenez, J. L., Kessler, S. H., Canagaratna, M. R., Wilson, K. R., Altieri, K. E., Mazzoleni, L. R., Wozniak, A. S., Bluhm, H., Mysak, E. R., Smith, J. D., Kolb, C. E. and Worsnop, D. R.: Carbon oxidation state as a metric for describing the chemistry of atmospheric organic aerosol, Nat. Chem., 3(2), nchem.948, doi:10.1038/nchem.948, 2011.

Koch, B. P., Dittmar, T., Witt, M. and Kattner, G.: Fundamentals of Molecular Formula Assignment to Ultrahigh Resolution Mass Data of Natural Organic Matter, Anal Chem, 79(4), 1758–1763, doi:10.1021/ac061949s, 2007.

Kundu, S., Fisseha, R., Putman, A. L., Rahn, T. A. and Mazzoleni, L. R.: High molecular weight SOA formation during limonene ozonolysis: insights from ultrahigh-resolution FT-ICR mass spectrometry characterization, Atmos. Chem. and Phys., 12(12), 5523–5536, doi:10.5194/acp-12-5523-2012, 2012.

LeClair, J. P., Collett, J. L. and Mazzoleni, L. R.: Fragmentation Analysis of Water-Soluble Atmospheric Organic Matter Using Ultrahigh-Resolution FT-ICR Mass Spectrometry, Environ. Sci. Technol., 46(8), 4312–4322, doi:10.1021/es203509b, 2012.

Lim, Y. B., Tan, Y., Perri, M. J., Seitzinger, S. P. and Turpin, B. J.: Aqueous chemistry and its role in secondary organic aerosol (SOA) formation, Atmos. Chem. and Phys., 10(21), 10521–10539, doi:10.5194/acp-10-10521-2010, 2010.

Maclean, A. M., Butenhoff, C. L., Grayson, J. W., Barsanti, K., Jimenez, J. L. and Bertram, A. K.: Mixing times of organic molecules within secondary organic aerosol particles: a global planetary boundary layer perspective, Atmos. Chem. Phys., 17(21), 13037–13048, doi:10.5194/acp-17-13037-2017, 2017.

Mazzoleni, L. R., Ehrmann, B. M., Shen, X., Marshall, A. G. and Collett, J. L.: Water-Soluble Atmospheric Organic Matter in Fog: Exact Masses and Chemical Formula Identification by Ultrahigh-Resolution Fourier Transform Ion Cyclotron Resonance Mass Spectrometry, Environ. Sci. Technol., 44(10), 3690–3697, doi:10.1021/es903409k, 2010.

Mazzoleni, L. R., Saranjampour, P., Dalbec, M. M., Samburova, V., Hallar, A. G., Zielinska, B., Lowenthal, D. H. and Kohl, S.: Identification of water-soluble organic carbon in non-urban aerosols using ultrahigh-resolution FT-ICR mass spectrometry: organic anions, Environ. Chem., 9(3), 285–297, doi:10.1071/en11167, 2012.

O'Brien, R. E., Wang, B., Laskin, A., Riemer, N., West, M., Zhang, Q., Sun, Y., Yu, X., Alpert, P., Knopf, D. A., Gilles, M. K. and Moffet, R. C.: Chemical imaging of ambient aerosol particles: Observational constraints on mixing state parameterization, J. Geophys. Res. Atmos., 120(18), 9591–9605, doi:10.1002/2015jd023480, 2015.

Ohno, T. and Ohno, P. E.: Influence of heteroatom pre-selection on the molecular formula assignment of soil organic matter components determined by ultrahigh resolution mass spectrometry, Anal. Bioanal. Chem., 405(10), 3299–3306, doi:10.1007/s00216-013-6734-3, 2013.

Putman, A. L., Offenberg, J. H., Fisseha, R., Kundu, S., Rahn, T. A. and Mazzoleni, L. R.: Ultrahigh-resolution FT-ICR mass spectrometry characterization of α-pinene ozonolysis SOA, Atmos. Env., 46, 164–172, doi:10.1016/j.atmosenv.2011.10.003, 2012.

Riemer, N. and West, M.: Quantifying aerosol mixing state with entropy and diversity measures, Atmos. Chem. Phys., 13(22), 11423–11439, doi:10.5194/acp-13-11423-2013, 2013.

Shiraiwa, M., Li, Y., Tsimpidi, A., Karydis, V., Berkemeier, T., Pandis, S., Lelieveld, J., Koop, T. and Pöschl, U.: Global distribution of particle phase state in atmospheric secondary organic aerosols, Nat. Commun., 8, ncomms15002, doi:10.1038/ncomms15002, 2017.

Shrivastava, M., Lou, S., Zelenyuk, A., Easter, R., Corley, R., Thrall, B., Rasch, P., Fast, J., Simonich, S., Shen, H. and Tao, S.: Global long-range transport and lung cancer risk from polycyclic aromatic hydrocarbons shielded by coatings of organic aerosol, P. Natl. Acad. Sci., 114(6), 1246–1251, doi:10.1073/pnas.1618475114, 2017.

Sorooshian, A., Lu, M.-L., Brechtel, F., Jonsson, H., Feingold, G., Flagan, R. and Seinfeld, J.: On the Source of Organic Acid Aerosol Layers above Clouds, Environ. Sci. Technol., 41(13), 4647–4654, doi:10.1021/es0630442, 2007.

Ye, Q., Robinson, E. S., Ding, X., Ye, P., Sullivan, R. C. and Donahue, N. M.: Mixing of secondary organic aerosols versus relative humidity, P. Natl. Acad. of Sci., 113(45), 12649–12654, doi:10.1073/pnas.1604536113, 2016.

Yu, J. Z., Huang, X., Xu, J. and Hu, M.: When Aerosol Sulfate Goes Up, So Does Oxalate: Implication for the Formation Mechanisms of Oxalate, Environ. Sci. Technol., 39(1), 128–133, doi:10.1021/es049559f, 2005.

Zelenyuk, A., Imre, D. G., Wilson, J., Bell, D. M., Suski, K. J., Shrivastava, M., Beránek, J., Alexander, M. L., Kramer, A. L. and Massey-Simonich, S. L.: The effect of gas-phase polycyclic aromatic hydrocarbons on the formation and properties of biogenic secondary organic aerosol particles, Faraday Discuss., 200, 143–164, doi:10.1039/c7fd00032d, 2017.

Zhang, B., Owen, R. C., Perlinger, J. A., Helmig, D., Val Martín, M., Kramer, L., Mazzoleni, L. R. and Mazzoleni, C.: Ten-year chemical signatures associated with long-range transport observed in the free troposphere over the central North Atlantic, Elem. Sci. Anth., 5, doi:10.1525/elementa.194, 2017.

Zhao, Y., Hallar, A. G. and Mazzoleni, L. R.: Atmospheric organic matter in clouds: exact masses and molecular formula identification using ultrahigh-resolution FT-ICR mass spectrometry, Atmos. Chem. Phys., 13(24), 12343–12362, doi:10.5194/acp-13-12343-2013, 2013.

Zielinski, A. T., Kourtchev, I., Bortolini, C., Fuller, S. J., Giorio, C., Popoola, O. A. M., Bogialli, S., Tapparo, A., Jones, R. L. and Kalberer, M.: A new processing scheme for ultra-high resolution direct infusion mass spectrometry data, Atmos. Environ., doi:10.1016/j.atmosenv.2018.01.034, 2018.

---

## Author Comment (AC3) · 29 May 2018

**Author Responses to Reviewer #2**

The *Reviewer comments are in black italic font* and the Author responses are in blue font.

*Schum et al. present a unique dataset collected on Pico Mountain Observatory to study the physiochemical properties of aerosol in the remote marine free troposphere. They analyzed three aerosol samples that had elevated organic carbon concentration, and attributed the differences in their molecular and physical characteristics to emission sources as well as their transport pathways. They observed a lower O/C ratio in two samples that they believed were likely from biomass burning plumes that were transported mostly in the free troposphere, and the aerosols were in a solid state that resisted oxidation. Before this work is published in ACP, the authors need to provide careful clarification and further discussion of several important aspects in this manuscript. Please find the comments below.*

We thank the reviewer for their helpful comments. We made several changes to both the main paper and the supplemental information. In particular, we made major revisions to section 3.5.

*Major comments*
*1. The O/C values for PMO-1 and PMO-3 are surprisingly low for particles that had been transported for 7-10 days. In Section 2.3, the authors pointed out that "losses of highly water soluble, low molecular weight organic compounds are expected". Highly water soluble compounds are presumably quite polar and thus should have higher O/C. Authors need to address how the SPE artifacts affect the overall sample O/C. The same issue applies to the artifacts of water extraction that the water-soluble compounds in the samples were preferably collected for the subsequent analysis. Please provide a discussion of possible bias, what is roughly the fraction that had been extracted versus not-extracted, and how it might affect the results of the analysis.*

We thank the reviewer for this comment. The sample preparation step is necessary because electrospray ionization (ESI) is used to study complex organic matter. The soft ionization method is susceptible to salt adducts that can complicate the mass spectra. Furthermore, low molecular weight (MW) compounds can be analyzed using other analytical techniques (e.g., gradient anion chromatography is suitable for several common low MW organic anions).

As suggested by the reviewer, the loss of low MW compounds with high O/C values does likely correspond to a decrease in the total water-soluble organic carbon O/C value. The magnitude of this depends on how the O/C value is determined. An overall larger difference is associated with relative abundance weighted O/C values (RA-weighted or O/C$_w$). On the other hand, a negligible difference is associated with arithmetic mean values (due to the very high number of identified molecular formulas). For this reason, a majority of the values reported in the discussion paper were arithmetic mean values. For a more complete comparison of arithmetic mean values of O/C from several studies, we refer the reviewer to Table 3 from Dzepina et al. (2015) shown below.

**Table 3.** Chemical characterization of the molecular assignments detected in selected studies. All values are average (arithmetic mean).

| Sample name | Sample type | Measurement site | O/C | H/C | OM/OC | DBE | DBE/C | MW | Reference |
|---|---|---|---|---|---|---|---|---|---|
| Pico 9/24 | Aerosol | Free troposphere | 0.46 | 1.17 | 1.73 | 10.7 | 0.47 | 478 | This study |
| Pico 9/25 | Aerosol | Free troposphere | 0.42 | 1.28 | 1.67 | 9.4 | 0.42 | 462 | |
| Storm Peak Lab S4SXA | Aerosol | Remote | 0.53 | 1.48 | 1.91 | 6.2 | 0.34 | 414 | Mazzoleni et al. (2012) |
| Millbrook, NY[1] | Aerosol | Rural | 0.32 | 1.46 | 1.60 | 6.30 | 0.33 | 366 | Wozniak et al. (2008) |
| Harcum, VA[1] | Aerosol | Rural | 0.28 | 1.37 | 1.54 | 7.45 | 0.38 | 360 | |
| K-Puszta 2004 (KP2004)[2] | Aerosol | Rural | 0.48 | 1.40 | 1.84 | 7.36 | 0.37 | 408 | Schmitt-Kopplin et al. (2010) |
| K-Puszta 2005 (KP2005)[2] | Aerosol | Rural | 0.39 | 1.22 | 1.69 | 10.1 | 0.46 | 430 | |
| Pearl River Delta, China | Aerosol | Urban, Suburban, Rural, Regional | 0.46 | 1.34 | 1.85 | 5.3 | 0.45 | 265 | Lin et al. (2012a) |
| Atlantic Ocean[3] | Aerosol | Marine boundary layer | 0.35 | 1.59 | 1.67 | 4.37 | 0.28 | 317 | Schmitt-Kopplin et al. (2012) |
| North Atlantic Ocean – All[4] | Aerosol | Marine boundary layer | 0.42 | 1.49 | 1.74 | 6.76 | 0.32 | 445 | Wozniak et al. (2014) |
| North Atlantic Ocean – Aged Marine[4] | Aerosol | Marine boundary layer | 0.36 | 1.56 | 1.70 | 5.88 | 0.28 | 423 | |
| Storm Peak Lab CW1 | Cloud water | Remote | 0.62 | 1.46 | 2.08 | 6.3 | 0.38 | 402 | Zhao et al. (2013) |
| Storm Peak Lab CW2 | Cloud water | Remote | 0.61 | 1.46 | 2.06 | 6.3 | 0.38 | 400 | |
| Fresno fog | Fog water | Rural | 0.43 | 1.39 | 1.77 | 5.6 | 0.40 | 289 | Mazzoleni et al. (2010) |
| Camden and Pinelands, NJ[5] | Rainwater | Urban impacted | 1.02 | 1.49 | 2.73 | 3.24 | 0.44 | 220 | Altieri et al. (2009a, b) |

Values were calculated: [1] For each sample presented in Wozniak et al. (2008). [2] For only two samples (KP2004 and KP2005) presented in Schmitt-Kopplin et al. (2010). [3] For only one, marine aerosol, sample presented in Schmitt-Kopplin et al. (2012). [4] For all samples (and only one PCA group) presented in Wozniak et al. (2014). [5] By combining the negative mode FT-ICR MS data available in Altieri et al. (2009a) (CHO, CHOS and CHNOS) and Altieri et al. (2009b) (CHON).

After consideration of all of the comments (including those of Reviewer 1 and the editor), we opted to instead focus on the RA-weighted values consistently throughout the manuscript. These values do help distinguish important trends in the data (e.g., Fig. 2 and 4). However, we note both here, and in the manuscript (lines 327-328), that the RA is not expected to directly correspond to the analyte concentrations because the ionization efficiencies depend on several factors such polarity, surface activity, and pH (Cech and Enke, 2001).

We estimated the impact of the missing low MW species on the overall O/C using the five highest mass concentrations (oxalate, acetate, lactate, formate, and malonate) as measured using ion chromatography. The measured mass concentrations were converted to their percent abundance relative to the total organic mass (estimated using an OM/OC conversion of 2 (El-Zanan et al., 2005)). The total ion abundance identified using ultrahigh resolution FT-ICR MS was assumed to represent as little as 50% of the total WSOC. Then the individual low MW compound mass fractions were used to estimate their abundance relative to the sum of the total abundance of species identified by FT-ICR MS. These abundance values were then used to estimate the weighted average O/C value for each of the samples. The following table was added to the Supplement (Table SM4).

**Table SM4. Estimated average O/C values when the ions are considered. The table contains the results for 3 assumptions of the organic mass fraction represented by the FT-ICR MS identified species (100%, 70%, 50%). The numbers in parentheses show the percent change in average O/C from the O/C without ions considered.**

| Sample | RA Weighted O/C without Ions (100%) | Ions and RA Weighted O/C (100%) | Ions and RA Weighted O/C (70%) | Ions and RA Weighted O/C (50%) |
|---|---|---|---|---|
| PMO-1 | 0.48 | 0.53 (10.42%) | 0.55 (14.58%) | 0.58 (20.83%) |
| PMO-2 | 0.57 | 0.70 (22.81%) | 0.75 (31.58%) | 0.81 (42.11%) |
| PMO-3 | 0.45 | 0.52 (15.56%) | 0.54 (20.00%) | 0.57 (26.67%) |

PMO-2 is still by far the most oxidized sample overall. PMO-1 and PMO-3 were still somewhat un-oxidized relative to our pre-conceived expectations based on transport time (Bougiatioti et al., 2014; Aiken et al., 2008). Oxalate is by far the most abundant organic ion and has the highest O/C, thus it yields the largest impact. Note that these O/C values are likely the upper limit of the average high MW O/C for these samples. The ionization preferences associated with negative mode ESI favor more highly oxidized higher molecular weight species, thus high MW molecular species detected by other ionization methods would likely only decrease the O/C.

To clarify the impact of the missing anions on the average O/C, the following has been added (see lines 164-171): **"The procedural loss of ionic low MW compounds such as oxalate can lead to an underprediction of the organic aerosol O/C and overprediction of the average glass transition temperatures ($T_g$). To investigate this, we used the concentrations of the prominent organic anions measured with ion chromatography to estimate the abundance of these compound relative to the compounds detected by FT-ICR MS. The low MW corrected average O/C values correlated with the trends of the original O/C values, however the significance of impacts varies with the measured analyte concentrations and the assumptions associated with the uncertain mass fraction of the molecular formula composition (Table SM4). When low MW organic anions were included in the estimated average dry $T_g$ values, they dropped by $\leq 2.5$ %, which was deemed relatively insignificant (Table SM5)."**

A description of the estimation method and data discussed above were added to the Supplement (Page 5).

*2. The authors use the method developed by DeRieux et al. to estimate particle phase state and heavily rely on the result to explain their findings. However, the authors use this method without further comment and discussion, especially regarding its uncertainty. Solid, semisolid and liquid state are qualitative descriptions which do not provide much insight into diffusion time-scale of water or organic molecules into/out of particles. Diffusion is a key process that determines the evolution of particle composition, and the connection of phase state and diffusivity involves multiple-step estimations with large uncertainties, as shown in a couple of studies [1][2]. Is it possible that the uncertainty of the method is large enough that it changes the major conclusions of this paper? The authors need to provide a much more comprehensive discussion of these issues.*

We thank the reviewer for this comment.

This work expands the understanding of the long-range transported aerosol collected at the Pico Mountain Observatory (PMO) presented in previous studies (Dzepina et al., 2015, China et al., 2015; China et al. 2017; Zhang et al., 2015; Zhang et al., 2017). As described, the site is located in the North Atlantic free troposphere on the Azores archipelago, and as such, it is quite remote. Additionally, the site is uniquely well-suited for the observation of long range transported aerosol due to the low marine boundary layer which is frequently below the site (See also Image 1 in the Supplement for a photo of the site and the mountain). Specifically, this paper attempts to advance the interpretation of pollution events arriving from North America using the detailed molecular chemistry.

We agree that diffusion is a key process in determining the evolution of particle compositions; however according to the FLEXPART retroplumes, the aerosol have been aloft for several days and the compositions that we measured are mostly low volatility compounds (Fig. 6 of manuscript). Our intent was not to provide exact predictions of diffusion or viscosity for the aerosol collected during the sampling periods (predictions for which our available sample and measurements would not be appropriate), but to provide an estimate of the most probable phase state for the organic aerosol during transport using the GFS meteorological fields (specifically ambient T and RH) associated with the FLEXPART retroplumes for a few upwind days. We then used the ratio of glass transition temperatures ($T_g$) to the ambient temperature ($T_g/T$) coupled with chemical markers to assist in the interpretation of our observations of the samples.

At present, we do not have enough information to predict the diffusion of species in the aerosol particles. Insufficient knowledge on the composition of aerosol particles and a lack of available methods to accurately measure viscosity for ambient samples at low concentrations limit the possibility to estimate diffusion, as described in a recent review by Reid et al. (2018). Therefore, we used general literature ideas about phase state and its impact on diffusion and viscosity to support the hypothesis that phase state limits the atmospheric oxidation of organic aerosol, which is consistent with several other studies (Shrivastava et al. 2017, Berkemeier et al., 2014, Lignell et al., 2014, Zelenyuk et al., 2017). Research reported in Ye et al. (2016), has shown that low volatility compounds resist diffusion even at high RH.

To be more accurate and avoid confusion, we removed the classification of molecular species as "solid", "semi-solid", and "liquid", and instead show only the estimated $T_g$. We also focused our discussion on the uncertainties in the $T_g$ estimates with respect to the DeRieux et al. (2018) defined error and the range of meteorological conditions extracted from FLEXPART. In fact, the range of ambient conditions presents a larger range of estimated $T_g$ values. As such, Figure 7 from the revised manuscript has revised and an additional version of the plot which demonstrates the distribution of $T_g$ values using just the mean RH for the RH dependent $T_g$ was added to the supplement (Fig. S17).

[Figure]

**Figure 7. Panels a-c contain the ambient conditions extracted from the GFS analysis along the FLEXPART modeled path weighted by the residence time for PMO-1, PMO-2, and PMO-3, respectively. The line represents the mean value and the shading represents one standard deviation of values. Panels d-f contain the boxplot distributions of the relative humidity dependent $T_g$ values for molecular formulas using the maximum, mean, and minimum RH for PMO-1, PMO-2, and PMO-3, respectively. The $T_g$ values for the full composition of each sample were calculated using the maximum, mean, and minimum RH and then all three sets of data are combined and plotted as a single distribution for each time period. The open circles represent the abundance and Boyer-Kauzmann estimated $T_g$ for the acid forms of the three most abundant low MW organic ions, the bars around the circles represent the range of possible $T_g$ values for those compounds when the range of RH is considered. The red line demonstrates the ambient temperature at each time point, as extracted from GFS. The centerline of the boxplot represents the median, the top and bottom of the "box" represent the third and first quartiles, respectively. The "whiskers" represent Q3 + 1.5\* interquartile range (IQR, Q3-Q1) (maximum), and Q1 – 1.5\*(IQR) (minimum).**

[Figure]

**Figure S17. Boxplots showing the distributions of the relative humidity dependent $T_g$ values for each sample over the last five days of transport. The open circles represent the Boyer-Kauzmann estimated $T_g$ values for the acid forms of the three most abundant low MW organic ions not observed in FT-ICR mass spectra. The symbols are scaled by their ambient concentration. The red line represents the mean ambient temperature from the GFS analysis. The samples PMO-1, PMO-2, and PMO-3 are shown in panels (a), (b), and (c), respectively. The centerline of the boxplot represents the median, the top and bottom of the "box" represent the third and first quartiles, respectively. The "whiskers" represent Q3 + 1.5\* interquartile range (IQR, Q3-Q1) (maximum) and Q1 – 1.5\*(IQR) (minimum).**

According to DeRieux et al. 2018, the estimation of $T_g$ has an error of ±21 K when considering only a single compound. They also mention that when considered as a group, the error decreases substantially, owing largely to some species being overestimated and some being underestimated, leading to the final result being reasonably accurate. In order to test the limit of the potential error, we added and subtracted 21 K from all the estimated $T_g$ values and replotted the distributions as presented in the manuscript.

As expected, the range of $T_g$ values increased. However, despite this increase, the majority of the $T_g$ distribution was still below the ambient temperature for PMO-2. Figure S18 (below) illustrates the results of these tests. However, to have the equation under or overestimate the $T_g$ of all formulas by the maximum reported error and the same extent with the same direction is highly unlikely and so it seems likely that our results are robust. Furthermore, an individual molecular formula represents a mixture of isomers with slightly different $T_g$ values, thereby potentially decreasing the error consistent with the description by DeRieux et al. (2018). For example, ultrahigh resolution MS/MS work by LeClair et al. (2012) has shown that most molecular formulas have more functional group losses (neutral losses of hydroxyl, carboxyl, etc.) than could be expected from a single isomer of a molecule. This has also been observed in the MS/MS analysis of PMO-1 which is the subject of a forthcoming paper.

To make the potential error due to the estimation clear the following has been added to the manuscript (Lines 213-217): **"DeRieux et al. (2018) reported an uncertainty of ± 21 K for the prediction of any single compound, but the uncertainty is expected to decrease when a mixture of compounds is considered. Nonetheless, we assumed an uncertainty range of ± 21 K on $T_g$ and found that it did not significantly change the $T_g$ trends presented in Section 3.5. Further discussion the uncertainty on $T_g$ is provided in the Supplement."**

[Figure]

**Figure S18. Relative humidity dependent $T_g$ distribution box plots with ± 21 K uncertainty (DeRieux et al., 2018) applied. Panels (a), (b), and (c) show the distributions for PMO-1, PMO-2, and PMO-3, respectively. Three distributions were calculated for each sample, one with 21 K added to the dry $T_g$, one with 21 K subtracted from the dry $T_g$, and one with the original $T_g$ values. The three data sets were combined here. The inclusion of the ± 21 K uncertainty does not significantly impact the range of observations. The centerline of the boxplot represents the median, the top and bottom of the "box" represent the third and first quartile respectively. The "whiskers" represent Q3 + 1.5* interquartile range (IQR, Q3-Q1) (maximum) and Q1 – 1.5*(IQR) (minimum).**

*Minor comments*
*1. In line 20, "This suggests that biomass burning emissions injected into the free troposphere are longer-lived than emissions in the boundary layer." The term "longer lived" is vaguely used here, as well as in a couple places in the main text. Do the authors mean the particles from biomass burning have lower oxidation state, or the authors are referring to the chemical life time of the compounds from biomass burning that were transported in the free troposphere?*

The intent was to indicate that aerosol in the free troposphere appear to be more resistant to removal, due in part to the ambient conditions. In this case, we specifically contrast this finding to previously reported lifetimes of biomass burning brown carbon species, which were predicted to have a lifetime of ~1 day within the boundary layer (Forrister et al., 2016; Laing et al., 2016). To clarify, we changed "long lived" to "persistent", everywhere as appropriate in the manuscript.

*2. In Section 3.1, chloride is presented in Table 1 but not discussed in the main text. Some studies show that biomass burning can produce chlorine-containing particles [3][4].*

We thank the reviewer for this interesting observation. To reflect this, we added the following to Section 3.1 (lines 250-251) of the paper: **"Chloride was also present in PMO-1 and PMO-3, which has been shown in some studies to be a minor product of biomass burning, depending on the fuel burned (Levin et al., 2010; Liu et al., 2017)."**

*3. In Figure 1 (c), the air mass spent a couple of days over Europe, and based on (f), the height of the air mass was quite low during those days. Could there be any influence from emissions from Europe on the sample?*

We thank the reviewer for this comment. First, the altitude profile plots were inadvertently misplaced, so the altitude of the airmass over Europe was not as low as shown there. We corrected this as soon as we realized the mistake during the discussion. We also corrected the plot in the final manuscript and added additional retroplumes with 12 hour time differences. Looking at the correct plots, the altitude was still somewhat low and the RH increased indicating potential influence from Europe. However, the molecular species identified in PMO-3 were much more similar to the more strongly influenced biomass burning sample (PMO-1) than they were to the anthropogenic, albeit North American, influenced sample (PMO-2). There is the possibility of European influence roughly 5 days before reaching the PMO, shown by the spike in altitude and RH during that time period, but it does not seem to be a major component based on the chemical comparisons of the three samples. Additionally, the source apportionment modeling did not predict European influence for that sample.

*4. In Section 3.2, regarding the CO source apportionment in Figure S1, what is the uncertainty associated with the CO modeling?*

We add a few sentences in Section 2.4 (highlighted in bold below) to discuss the uncertainty and features associated with the FLEXPART CO simulations. We would like to point out that the FLEXPART CO tracer does not reproduce the actual CO concentration at the site because FLEXPART only simulates the transport of emissions but not the chemistry or deposition. FLEXPART CO was set to have a cutoff lifetime of 20 days in the model, but in reality, the CO lifetime varies from weeks to months depending on the location and atmospheric conditions. In this work, FLEXPART CO simulations were used as an indicator to show the relative contributions from anthropogenic and biomass burning emissions rather than an estimate of CO concentrations at Pico.

Manuscript excerpt (Lines 191-207):

"FLEXPART  was used to determine the sources, ages, and transport pathways of the aerosol samples collected at PMO. FLEXPART backward simulations (also called retroplumes) were driven by meteorology fields from the Global Forecast System (GFS) and its Final Analysis (FNL) with 3-hour

temporal resolution, 1° horizontal resolution, and 26 vertical levels. The output was saved in a grid with a horizontal resolution of 1° latitude by 1° longitude, and eleven vertical levels from the surface to 15,000 m a.s.l. **For each simulation, 80 thousand air parcels were released from the receptor and transported backwards for 20 days to calculate a source-receptor relationship (in units of s kg⁻¹, Seibert and Frank, 2004)**. FLEXPART retroplumes  are **then** multiplied with CO emission inventories **(kg s⁻¹)** from the Emissions Database for Global Atmospheric Research (EDGAR version 3.2 (Olivier and Berdowski, 2001)) and the Global Fire Assimilation System (Kaiser et al., 2012) to estimate the influence from anthropogenic and wildfire sources, respectively. **The FLEXPART CO tracer calculated with this approach indicates the relative contributions from anthropogenic and biomass burning emissions. Since CO chemistry and dry deposition are not considered in the FLEXPART setup, the absolute FLEXPART CO value does not reproduce the actual CO concentrations at Pico. FLEXPART does not consider the background CO accumulated in the atmosphere. The difference between FLEXPART CO and the actual CO largely depends on these factors. In previous applications of this approach, FLEXPART CO was able to estimate the episodes of CO enhancement due to transport of emissions (e.g., Brown et al., 2009; Stohl et al., 2007; Warneke et al., 2009).** This  approach has been used in several PMO studies **and successfully captured elevated CO periods** (e.g., Dzepina et al., 2015; Zhang et al., 2014, 2017) **and it is used here to assist in** the interpretation of the chemical composition in this work"

*5. In Section 3.3, line 285-287, 78% of the formulas in PMO-2 are found to be common with sample from the boundary layer aerosol, and PMO-3 has similarity of 76%. Are 78% and 76% significantly different? This piece of information might not be a strong evidence to support the conclusion that PMO-2 was largely influenced by North America outflow transported within the boundary layer while PMO-3 was not.*

As mentioned in the manuscript, there are many species (especially CHO molecular formulas) that are present in all samples. In addition, PMO-3 does not have a large number of unique species relative to PMO-1 or PMO-2. This is largely due to the sampled air as shown in the retroplume being somewhat more diffuse with a less certain path of transport and also origin, than either PMO-1 or PMO-2. Despite the large percentage of common species between PMO-3 and the boundary layer sample from Storm Peak Laboratory (SPL), PMO-3 had much more in common with a previous free troposphere wildfire sample (91%) (September 24, 2012, Dzepina et al., 2015). This suggests PMO-3 is more similar to PMO-1 than PMO-2, but there is the potential for a non-negligible influence from the European boundary layer. The point of the percentages, was to show that PMO-1 and PMO-3 have more in common with free tropospheric wildfire aerosol, than they do with the continental boundary layer aerosol of somewhat mixed sources (SPL aerosol).

To clarify, we revised the text (lines 313 – 318): **"In fact, when we compared the molecular formula composition of the free tropospheric aerosol sample "9/24" from the study by Dzepina et al. (2015) to the free tropospheric samples in this study (PMO-1 and PMO-3), we observed that 86% and 91% of the formulas are common. FLEXPART simulations from both studies suggested these samples were all affected by wildfire emissions, contributing to their similarity. In contrast, only 75% of the formulas found in the boundary layer sample (PMO-2) were common with those in Dzepina et al. (2015). These comparisons are provided in Table S2."**

*6. In Figure 2, an obvious difference of the three spectra is the much higher fraction of high molecular weight materials in PMO-2. Little is discussed about the sources of the high molecular weight compounds in the text. Are they from oligomerization? In contrast, Lee et al. [5] observed abundant high molecular weight compounds from biomass burning in Canada using an aerosol mass spectrometer.*

Considering the percentage of species with a mass greater than 350, PMO-2 actually has the smallest percentage of its total formulas in that range both by number of formulas and percentage of total abundance. PMO-1 has 70% of its formulas above 350 and PMO-3 has 71% of it formulas in that range, and PMO-2 has 64% of the formulas in this range. In terms of percentage of total abundance, they make up 63% of PMO-1, 59% of PMO-2, and 65% of PMO-3. These numbers are admittedly similar, but PMO-1 and PMO-3 are more similar and are both somewhat higher than PMO-2. These results may support the observations made in Lee et al. (2016) regarding high molecular weight compounds from biomass burning in Canada.

The main reason why those species (m/z > 350) stand out so much is due to their normalized relative abundance, where each measured intensity was normalized by the total ion intensity of the assigned molecular formulas in each sample. The implications of this increased O/C and subsequently, oxidation is the major focus of this paper and is discussed several times. The tall peaks that really stand out (norm. RA > 0.1) only make up 136 of 1349 masses above m/z 350 in PMO-2. Additionally, while analyzing the samples, we investigated the potential of SOA type oligomerization, and were unable to find any clear evidence, and thus did not include it in this manuscript. Also, interestingly, the fire studied by Lee et al. (2016) is likely the same fire that impacted the air mass that intercepted PMO on June 27-28, 2013 (PMO-1), so the relative increase in higher molecular mass compounds is consistent between the two studies.

*7. In section 3.5, line 387, "Volatility can also play a role in the phase state". This expression is vague. Do the authors mean phase state depends on volatility? Or they both relate to structures of molecules in particles? Please make clarification.*

The sentence was replaced with the following (Lines 424-426): **"In general, lower volatility typically inversely correlates with $T_g$ (Shiraiwa et al., 2017) and viscosity. As such it was important to estimate the volatility of the PMO aerosol."**

*8. In section 3.5, line 392, "This highlights the correlation between O/C and volatility, where volatility is expected to decrease as O/C increases." What about fragmentation?*

Fragmentation can definitely contribute to both a decrease in O/C and an increase in volatility which requires other chemical changes in the compounds, namely a decrease in mass. The general mass ranges for all three samples is consistent and so fragmentation is unlikely to be the source of lower O/C in PMO-1 or PMO-3. Fragmentation may have occurred in PMO-2, helping to contribute to the increased O/C value observed, but it does not change the observation that the predicted volatility for the high abundance species in PMO-2 was lower than for the high abundance species in PMO-1 or PMO-3, or that the high abundance species in PMO-2 were also those with elevated O/C. Furthermore, the estimation of volatility includes a term regarding the carbon and oxygen interactions (Donahue et al., 2011; Li et al., 2016) indicating a relationship between O/C and volatility. Also, studies have shown a relationship between O/C and volatility before (Ng et al., 2011).

To address this, the following has been added (Lines 429-432):
**"This highlights the relationship between O/C and volatility, where volatility is expected to decrease as O/C increases when the mass range is constant (Ng et al., 2011); the relationship between oxygen and carbon and its effect on volatility is used by both Donahue et al. (2011) and (Li et al., 2016) to estimate volatility. Similarly, lower volatility is expected to lead to lower diffusivity in aerosol even at elevated RH as demonstrated by Ye et al. (2016)."**

*9. Lastly, how generalizable are these findings in the paper in terms of predicting the oxidation state of aerosols having different transport pathways?*

We thank the reviewer for this interesting question. The idea of phase state having an impact is likely fairly generalizable because it has been shown in several studies to have an impact on the rate of chemical reaction in aerosol samples (Koop et al., 2011; Berkemeier et al., 2014; Lignell et al., 2014; Shrivastava et al., 2017; Zelenyuk et al., 2017). Thus, it is fair to say that aerosol traveling high in the atmosphere, effectively since emission, can be anticipated to have a relatively low oxidation state. However, more samples are needed to be studied using multiple ionization modes to get a more complete analysis of what is present in these samples.

Regarding this question we added the following to the manuscript (lines 512-514):

**"More work is needed to better constrain the molecular composition of long range transported aerosol and the processes that affect it during transport. The presented results have broader implications for the aging of long range transported biomass burning organic aerosol rapidly convected to the free troposphere."**

---

## Referee Report (RR1)

The revised manuscript „Molecular and physical characteristics of aerosol at a remote marine free troposphere site: Implications for atmospheric aging" by Schum et al. has improved compared to the initial submission. The authors immediately corrected the misassigned back-trajectory analysis in Fig.1. They evaluated the possible bias in calculating the glass transition temperature by small weight molecules, which are lost during sample preparation. The authors show that small weight molecules, such as oxalic acid, only marginally influence the overall glass transition temperature. However, other small organic acids (acetic acid, formic acid) do have a pronounced influence on the glass transition temperature.

One main point noticed in the earlier discussion was that the authors discuss the differences of DI-ESI-signatures of samples with different emission sources, and conclude that the conditions during transport to PMO are the cause of the different chemical signature. The authors have addressed this point by acknowledging that the emission sources on their own do play a role (l. 504). However, in my opinion, the authors still do not adequately discuss this aspect throughout their current manuscript (except the parenthesis in l. 504). What is still missing is a discussion on possible secondary gas phase processes in an anthropogenic pollution plume (sample PMO-2), in which high NOx, high SO2, and secondary ozone can also result in completely different conditions, compared to the biomass burning plumes (PMO-1 and 3). In their response to the reviewer´s comments, the authors argue that aqueous phase processing "leads to SOA production with a greater array of carbon numbers; the greater number of carbon numbers matches more closely with our observations of a continuum of carbon numbers from 2 to 33 in PMO-2". This argument does not seem convincingly to me, since there is a clear continuum of carbon numbers in the same range for PMO-1 and PMO-3 (Fig. 4 (a)-(c)), as well. The continuum of carbon numbers is actually the largest in PMO-1 (the biomass burning sample), resulting in ion signals up to 700 amu (Fig. 2 (a)). Thus, it looks like the emission source rather dominates the observed carbon number array.

I like the idea of comparing the PMO samples with samples of cloud and fog water (Table S.6) to identify compounds that are unique markers for aqueous phase processing. While the comparison between Cook et al. (2017) and PMO shows in fact the largest number of signals in common with PMO-2, the comparison with Zhao et al. (2015) shows the majority of common signals with PMO-1. We see that O/C of the comparison is highest for the sample PMO-2, but it remains unclear whether this is driven by the stronger presence of high O/C compounds in PMO2, which are not present in the samples PMO1/3. Thus, to me it remains elusive that this comparison is a clear indication for the "influence of aqueous phase processing" in sample PMO2.

Concerning ESI-artifacts: I do believe that negative ESI is less prone to adduct artifacts than positive ESI. The authors mentioned that samples are diluted to the lowest possible level to obtain a stable current during ESI. In principle, ESI should form always a stable current- even without any analyte present (e.g. in blank measurements). The authors mention that their AGC target of 1e6 ions was reached after 20-80 ms. I compared this to injection times on a Q-Exactive System (AGC target 1e6) after chromatographic separation – same ballpark. Thus, it seems that the samples were in fact reasonably diluted, however, it would also be interesting to see the actual ion count rates. I am asking for these numbers, since the NL value of the MS/MS experiments seem to me rather low (e.g. 5e3 for m/z 300 +/- 3). Such low value can be an indication for the presence of cluster ions, provided that the ion signal in the full scan shows a much higher signal (not possible to evaluate here, since the numbers are not reported).

Overall, I see the strength of this paper in relating the extracted conditions (T and RH) during transport with the observed chemical composition. But, I do miss a solid chain of reasoning toward the hypothesis of aqueous-phase aging in the MBL in contrast to slow aging rates in the FT when the aerosol is in a glassy state. It all goes back to the fact that the samples have a completely different source (anthropogenic pollution vs biomass burning), and the chemical signature of the aerosol, when it is close to its emission source (t_zero of aging), is not known. Thus, I do not see point 5 of the ACP review criteria fulfilled (*Are the results sufficient to support the interpretations and conclusions?*). Still, the paper provides a novel and interesting concept: the extraction of glass transition temperatures from molecular ion measurements. Also the atmospheric implications are highly relevant: slow aging rates of BB aerosol and slow decomposition of BrC in the free troposphere. Therefore, I recommend the article to be published in ACP after a more critical discussion regarding the fact that the initial aerosol composition in plume PMO2 (when it is still over the continent) is not known, and that other processes in the plume than only aqueous-phase oxidation during transport could explain the high O/C of the sample PMO-2.

Specific comments:

l. 13-16: Here the authors conclude that "environmental factors during transport" are responsible for the higher O/C in PMO-2. I am missing any mentioning of the possibility of different chemical composition (e.g. already higher O/C) of the PMO2 plume right at its source. Also, the term "environmental factors during transport" is too blurry.

l. 20: The same issue here: the comparison between PMO1/3 and PMO2 has to be discussed with more caution. The data presented do not allow the conclusion that solely Tg/T is the cause of higher oxidized aerosol in PMO2. I recommend something like "[…] and therefore less susceptible to oxidative aging than the organic aerosol transported in the boundary layer."

l. 28: […] cloud droplet and ice nucleation activity […]?

l. 34-36: The biomass burning studies cited here might not be the best, since they all refer to biomass burning events from grassland fires. Grass-lignin is different from softwood (coniferous) lignin, resulting in different biomass burning marker molecules (Simoneit et al., 1993). Refering to studies on biomass burning plumes from boreal forest fires (e.g. Corrigan et al. (2013)) might be more appropriate in the context of discussing plumes from Canadian boreal forests.  It is well recognized that the most prominent organic biomass burning marker (levoglucosan) undergoes fast photooxidation, however, I did not find a connection between biomass burning markers and their oxidative degradation in the paper by Vakkari et al., 2014. Here, a paper on the degradation kinetics of levoglucosan would fit much better (e.g. Lai et al. (2014), Arangio et al. (2015))

l. 38: The authors might find our paper on long-range-(low-altitude)-transported biomass burning aerosol from wildfires in Russia to the SMEAR station in Hyytiälä, Finland of interest. During a pollution plume event we observed highly oxidized aerosol (O/C ~ 0.70), while the average O/C during this campaign ranged around 0.5. We also observed high-molecular weight organic matter in the aerosol phase during this BB event, indicated by molecular ion signals up to *m/z* 800 (Vogel et al., 2013).

l. 244: A low oxalate/sulfate ratio is reported for PMO-2, likely obscured by high sulfate concentrations. The authors argue that oxalate is also high in PMO-2 due to aqueous phase processing. To further search for evidence that cloud processing has really occurred for PMO-2, it would be of interest to compare the oxalate/sulfate ratio in sample PMO2 with reported oxalate/sulfate ratios in the Eastern US close to the emission sources.

l. 307: Is the North American SOA solely anthropogenic? I think there are several studies reporting dominant biogenic SOA in East America (e.g. isoprene SOA in South East US?).

l. 353-366: The authors describe a higher O/C in PMO2 for the CHOS group compared to PMO1, and argue at the end of the section that this observation highlights the enhanced aging during transport of PMO2. While I believe that there is indeed a higher oxygen content in the sample PMO2, it does matter to which atom the oxygen is bonded. If it appears as oxygen-sulfur bonds (as organosulfates), then the increased oxygen content (higher O/C) in PMO-2 rather tells us that the different source emissions of the PMO2 plume allowed enhanced formation of organosulfates. This observation would be in line with the observation of higher inorganic sulfate in PMO2, as well as the expectation that SO2 emissions over North East US are higher than in the remote Canadian boreal forests.

l. 380: It is too speculative to talk about the CCN ability of the sampled aerosol particles when their size distribution is not known. Also the speculation about the amount of less volatile components seems to me ambiguous, since the total mass concentration available for gas-to-particle partitioning will also affect the fraction of higher-volatility compounds in the particle phase.

l. 472: What is about the role of multiphase and heterogeneous oxidation of the aerosol additionally to aqueous-phase processing of cloud droplets?

l. 480-494: The section is missing a more critical discussion, including the fact that the initial chemical composition of the plume sampled in PMO2 is not known. Again, PMO2 is compared here against PMO1 and 3. The fact that higher sulfate is observed in PMO2 compared with PMO1/3 rather goes back to the emission source, where you expect more SO2 being emitted in North America than in the boreal forest. The relatively higher abundance of sulfate in PMO2 against PMO1/3 hence does not necessarily support cloud processing.

l. 488-491: These lines are redundant with section 3.1 (l.235 ff.).

l. 508: Only aqueous-phase oxidation?

**References**

Arangio, A. M., Slade, J. H., Berkemeier, T., Pöschl, U., Knopf, D. A., and Shiraiwa, M.: Multiphase chemical kinetics of OH radical uptake by molecular organic markers of biomass burning aerosols: humidity and temperature dependence, surface reaction, and bulk diffusion, J. Phys. Chem. A, 119, 4533–4544, doi:10.1021/jp510489z, 2015.

Corrigan, A. L., Russell, L. M., Takahama, S., Äijälä, M., Ehn, M., Junninen, H., Rinne, J., Petäjä, T., Kulmala, M., Vogel, A. L., Hoffmann, T., Ebben, C. J., Geiger, F. M., Chhabra, P., Seinfeld, J. H., Worsnop, D. R., Song, W., Auld, J., and Williams, J.: Biogenic and biomass burning organic aerosol in

a boreal forest at Hyytiälä, Finland, during HUMPPA-COPEC 2010, Atmos. Chem. Phys., 13, 12233–12256, doi:10.5194/acp-13-12233-2013, 2013.

Lai, C., Liu, Y., Ma, J., Ma, Q., and He, H.: Degradation kinetics of levoglucosan initiated by hydroxyl radical under different environmental conditions, Atmospheric Environment, 91, 32–39, doi:10.1016/j.atmosenv.2014.03.054, 2014.

Simoneit, B.R.T., Rogge, W. F., Mazurek, M. A., Standley, L. J., and Hildemann, L.M., and Cass, G. R.: Lignin pyrolysis products, lignans, and resin acids as specific tracers of plant classes in emissions from biomass combustion, Environmental Science & Technology, 27, 2533–2541, 1993.

Vogel, A. L., Äijälä, M., Corrigan, A. L., Junninen, H., Ehn, M., Petäjä, T., Worsnop, D. R., Kulmala, M., Russell, L. M., Williams, J., and Hoffmann, T.: In situ submicron organic aerosol characterization at a boreal forest research station during HUMPPA-COPEC 2010 using soft and hard ionization mass spectrometry, Atmos. Chem. Phys., 13, 10933–10950, doi:10.5194/acp-13-10933-2013, 2013.

---

## Author Response (AR2)

**Aggregated Author Response Document**

We thank Editor Donahue for the opportunity to further refine and improve the manuscript.

In summary, we have:

- made several modifications to clarify the limitations of the current study,
- provided additional description of the oxidation pathways relevant for PMO-2, and
- improved the overall readability of the manuscript.

Overall despite the study limitations, the observations reported here have important implications for atmospheric aging and transport of organic aerosol in the free troposphere. We believe our unique sample set and detailed molecular analytical approach provides a unique opportunity to consider the implications on aerosol aging associated with respect to the transport pathways, meteorological conditions, and aerosol phase state.

A point-by-point response to each of the reviewer comments and a tracked changes version of the manuscript are provided below.

**Author Responses to Reviewer #1**

We thank Reviewer #1, Alexander Vogel, for his evaluation of the revised ACPD manuscript and useful comments, which helped us further improve the manuscript. The referee agrees that this paper provides a novel approach for assessing the aging of aerosol with highly relevant atmospheric implications, and that it should be published after including a more critical discussion on aerosol composition and processing.

In this reply *the Reviewer comments are given in black italic font* and the Authors responses are given in blue font. Manuscript text that we added or revised to address the comments is given in **bold font**. All page and line numbers are for the final revised manuscript (note, the line numbers in the tracked changes version of the manuscript are different due to inline tracking).

*The revised manuscript „Molecular and physical characteristics of aerosol at a remote marine free troposphere site: Implications for atmospheric aging" by Schum et al. has improved compared to the initial submission. The authors immediately corrected the misassigned back-trajectory analysis in Fig.1. They evaluated the possible bias in calculating the glass transition temperature by small weight molecules, which are lost during sample preparation. The authors show that small weight molecules, such as oxalic acid, only marginally influence the overall glass transition temperature. However, other small organic acids (acetic acid, formic acid) do have a pronounced influence on the glass transition temperature.*

*One main point noticed in the earlier discussion was that the authors discuss the differences of DI-ESI-signatures of samples with different emission sources, and conclude that the conditions during transport to PMO are the cause of the different chemical signature. The authors have addressed this point by acknowledging that the emission sources on their own do play a role (l. 504). However, in my opinion, the authors still do not adequately discuss this aspect throughout their current manuscript (except the parenthesis in l. 504). What is still missing is a discussion on possible secondary gas phase processes in an anthropogenic pollution plume (sample PMO-2), in which high NOx, high SO2, and secondary ozone can also result in completely different conditions, compared to the biomass burning plumes (PMO-1 and 3).*

To address the reviewer's concern that we did not sufficiently clarify the significance of different emission sources, we added/revised the following text:

- Lines 308-314: "**The North American boundary layer outflow of organic aerosol captured in PMO-2 was likely influenced by SOA (Zhang et al., 2007) and thus** is expected to have a higher

**initial** O/C value compared **to pyro-convected wildfire emissions of organic** aerosol (e.g., Aiken et al., 2008; Jimenez et al., 2009; Bougiatioti et al., 2014). **Although the initial compositions are unknown, we anticipated that the samples with longer transport times (~ 1 week for PMO-1 and PMO-3) would be at least similar or perhaps more oxidized than PMO-2 which had a much shorter transport time (~ 3 days). This** expectation was based on **literature describing** secondary organic aerosol **formation and aging** (Volkamer et al., 2006; Jimenez et al., 2009) and **the reported molecular composition of** continental boundary layer aerosol (Mazzoleni et al., 2012; Huang et al., 2014).**"**

- Lines 366-370: "**The increased number of sulfur species observed in PMO-2 is likely associated with the anthropogenic emission sources in the North American boundary layer. Overall, the observed** differences in the O/C ratios between the boundary layer transported aerosol (PMO-2) compared to the free troposphere transported aerosol (PMO-1 and PMO-3) highlight differences in the aging and lifetime of aerosol relative to its transport pathway **and emission source.**"

- Lines 516-517: "PMO-2 aerosol were transported primarily through the boundary layer over the Northeast continental U.S. and the North Atlantic Ocean **and was largely influenced by anthropogenic and biogenic sources.**"

To address the reviewer's concern that we did not sufficiently discuss the secondary gas phase processes, we added the following text:

- Lines 486-492: "**However, the exact oxidation pathways that led to the increased oxidation observed for PMO-2 and its initial composition are unclear. Both gas phase and aqueous phase reactions lead to SOA formation, where aqueous SOA components can have higher O/C values than gas phase SOA components (Lim et al., 2010; Ervens et al., 2011). The high numbers of CHNO and CHOS molecular formulas observed here are consistent with secondary components associated with an emission plume likely enriched in $SO_2$, NOx, and $O_3$ pertaining to its expected anthropogenic influence. All three of these reactive species have been shown to lead to production and oxidation of SOA in the atmosphere (Hoyle et al., 2016; Bertrand et al., 2018).**"

- Lines 506-509: "**While clearly gas phase SOA cannot be excluded, several lines of evidence suggest that aqueous phase oxidation likely influenced the chemical and physical characteristics of the PMO-2 aerosol to a larger extent than those of PMO-1 and PMO-3 based on the observed molecular characteristics, major ion concentrations (Fig. S20), and the model simulated transport pathways and GFS meteorology.**"

*In their response to the reviewer's comments, the authors argue that aqueous phase processing "leads to SOA production with a greater array of carbon numbers; the greater number of carbon numbers matches more closely with our observations of a continuum of carbon numbers from 2 to 33 in PMO-2". This argument does not seem convincingly to me, since there is a clear continuum of carbon numbers in the same range for PMO-1 and PMO-3 (Fig. 4 (a)-(c)), as well. The continuum of carbon numbers is actually the largest in PMO-1 (the biomass burning sample), resulting in ion signals up to 700 amu (Fig. 2 (a)). Thus, it looks like the emission source rather dominates the observed carbon number array.*

The reviewer appears to have misunderstood our comment in the previous response document.

The original response in Author Responses to Reviewer #1 (last paragraph on page 2) was "Auto-oxidation as described by Crounse et al. (2013), Ehn et al. (2012), and Jokinen et al. (2014) does increase the O/C, but it also shows clear carbon number preferences associated with the oxidation of terpene precursors. This trend is consistent with our earlier work on condensed SOA where the concept of "auto-oxidation" was described as "oxygen-increasing-reactions" (Kundu et al., 2012). However, in the case of PMO-2, we did not observe carbon number preferences, which would indicate auto-oxidation. While this does not negate the possible influence of auto-oxidation, it does minimize its relative importance for these long range transported aerosol observations. On the other hand, aqueous phase processing as described by Lim et al. (2010) leads to SOA production with a greater array of carbon numbers; the greater array of carbon numbers matches more closely with our observations of a continuum of carbon numbers from 2 to 33 in PMO-2. "

Thus, the phrase "a greater array of carbon numbers" specifically referred to the dispersion of ion intensity over the range of carbon numbers for aqueous phase SOA vs. auto-oxidation. It by no means compares the range of carbon values for emission sources. The well-known polymeric signatures at specific carbon numbers is routinely observed with terpene SOA, but not aqueous SOA and not combustion emissions. Therefore, a continuum of carbon numbers does not imply an emission source.

*I like the idea of comparing the PMO samples with samples of cloud and fog water (Table S.6) to identify compounds that are unique markers for aqueous phase processing. While the comparison between Cook et al. (2017) and PMO shows in fact the largest number of signals in common with PMO-2, the comparison with Zhao et al. (2015) shows the majority of common signals with PMO-1. We see that O/C of the comparison is highest for the sample PMO-2, but it remains unclear whether this is driven by the stronger presence of high O/C compounds in PMO2, which are not present in the samples PMO1/3. Thus, to me it remains elusive that this comparison is a clear indication for the "influence of aqueous phase processing" in sample PMO2.*

This line of evidence, in conjunction with the other lines of evidence that we have put forward (back trajectories, RH, and ion concentrations), supports our hypothesis for the influence of aqueous processing on the observed oxidation of PMO-2. Furthermore, Figure S19 shows the species that are uniquely common between each of the PMO samples and the SPL cloud water. In it, the formulas in PMO-2 are clearly focused in the more oxidized region of the plot, while the majority of the unique formulas for PMO-1 and PMO-3 are in less oxidized, lower O/C regions of the plot. The rest of the formulas are common between two or more of the samples are thus fairly well accounted for when calculated the average O/C. In other words, the greatly increased O/C observed for the species common between PMO-2 and the cloud water is driven by the presence of high O/C formulas that are not present in the other samples.

Note we assumed the reviewer intended to refer to Zhao et al. (2013), as we've previously cited, instead of (2015).

*Concerning ESI-artifacts: I do believe that negative ESI is less prone to adduct artifacts than positive ESI. The authors mentioned that samples are diluted to the lowest possible level to obtain a stable current during ESI. In principle, ESI should form always a stable current- even without any analyte present (e.g. in blank measurements). The authors mention that their AGC target of 1e6 ions was reached after 20-80 ms. I compared this to injection times on a Q-Exactive System (AGC target 1e6) after chromatographic separation – same ballpark. Thus, it seems that the samples were in fact reasonably diluted, however, it would also be interesting to see the actual ion count rates. I am asking for these numbers, since the NL value of the MS/MS experiments seem to me rather low (e.g. 5e3 for m/z 300 +/- 3). Such low value can be an indication for the presence of cluster ions, provided that the ion signal in the full scan shows a much higher signal (not possible to evaluate here, since the numbers are not reported).*

The normalization levels (NL) shown in the mass spectra are arbitrary values corresponding to the ion intensity of the base peak and not the total number of ions (corresponding to the auto-gain control (AGC) setting). The NL levels on the FT-ICR MS (FT Ultra, Thermo Scientific) are a few orders of magnitude lower than they are in the Orbitrap MS (e.g., Orbitrap Elite, Thermo Scientific). The FT-ICR MS instruments are significantly less sensitive than the Orbitrap MS instruments; thus, co-addition of 200 recorded transient spectra is routinely done with FT-ICR MS (as described in lines 180-182). We have observed NL values of $10^3 - 10^5$ on the FT-ICR MS and NL values of $10^7 - 10^8$ on the Orbitrap Elite. We do not know how either of these instruments compare to the Q-Exactive System.

The mass spectra we studied do not show evidence of ESI artifacts. We believe this is because we analyze dilute solutions using negative ion ESI after reverse phase SPE isolation to remove the inorganic salts and low molecular weight organic salts that are possible adducting species (e.g., lines 158-164). Further analysis of the MS/MS spectra is beyond the scope of the current manuscript.

*Overall, I see the strength of this paper in relating the extracted conditions (T and RH) during transport with the observed chemical composition. But, I do miss a solid chain of reasoning toward the hypothesis of aqueous-phase aging in the MBL in contrast to slow aging rates in the FT when the aerosol is in a glassy state. It all goes back to the fact that the samples have a completely different source (anthropogenic pollution vs biomass burning), and the chemical signature of the aerosol, when it is close to its emission source (t_zero of aging), is not known. Thus, I do not see point 5 of the ACP review criteria fulfilled (Are the results sufficient to support the interpretations and conclusions?). Still, the paper provides a novel and interesting concept: the extraction of glass transition temperatures from molecular ion measurements. Also the atmospheric implications are highly relevant: slow aging rates of BB aerosol and slow decomposition of BrC in the free troposphere. Therefore, I recommend the article to be published in ACP after a more critical discussion regarding the fact that the initial aerosol composition in plume PMO2 (when it is still over the continent) is not known, and that other processes in the plume than only aqueous-phase oxidation during transport could explain the high O/C of the sample PMO-2.*

We thank the reviewer for highlighting the strengths of the manuscript and the recommendation to be published in ACP. To address the reviewer's concern regarding the need for additional discussion on the initial aerosol composition and other potentially present processes, we additionally revised the following paragraph (lines 484-509) describing the oxidation of PMO-2. Here we include the entire paragraph with all changes in bold, although some of the sentences from this paragraph were already mentioned above.

**"As described above, the most obvious difference in the molecular composition of PMO-2 vs. PMO-1 and PMO-3 is the increased extent of oxidation. In fact, most of the unique species observed in PMO-2 are in the highly oxidized region of the van Krevelen plot (Fig. 8). However, the exact oxidation pathways that led to the increased oxidation observed for PMO-2 and its initial composition are unclear. Both gas phase and aqueous phase reactions lead to SOA formation, where aqueous SOA components can have higher O/C values than gas phase SOA components (Lim et al., 2010; Ervens et al., 2011). The high numbers of CHNO and CHOS molecular formulas observed here are consistent with secondary components associated with an emission plume likely enriched in $SO_2$, NOx, and $O_3$ pertaining to its expected anthropogenic influence. All three of these reactive species have been shown to lead to the production and oxidation of SOA in the atmosphere (Hoyle et al., 2016; Bertrand et al., 2018). Cloud and aqueous phase processing have also** been shown to increase the oxidation of atmospheric organic matter (e.g., Ervens et al., 2008; Zhao et al., 2013; Cook et al., 2017; **Brege et al., 2018**). Comparisons of the detailed molecular composition of the PMO samples with studies of cloud (Zhao et al., 2013; Cook et al., 2017) and fog (Mazzoleni et al., 2010) organic matter indicate that the formulas uniquely common to only PMO-2 have higher O/C, **which supports** aqueous phase processing during transport. These results are provided in Fig. S19 and Table S6. **Studies have shown that the reactive species emitted from anthropogenic plumes ($SO_2$, NOx, $O_3$) can play a role in the oxidation of the**

**organic species that are dissolved in water (Blando and Turpin, 2000; Chen et al., 2008; Ervens et al., 2011). Furthermore, studies have shown that aerosol liquid water content contributes to aqueous production of SOA (Volkamer et al., 2009; Lim et al., 2010). The elevated RH extracted from the GFS for this plume (Fig. 7) indicates the presence of aerosol liquid water and is consistent with its ubiquitous nature (Nguyen et al., 2016).** Additionally, PMO-2 had a strongly elevated non-sea salt sulfate concentration relative to PMO-1 and PMO-3, which **also indicates aqueous phase** processing **(Crahan et al., 2004; Yu et al., 2005; Sorooshian et al., 2007; Hoyle et al., 2016)**. Oxalate, **another well-known** marker of aqueous phase processing **(Warneck 2003; Crahan et al., 2004;** Yu et al., 2005; Sorooshian et al., 2007; **Carlton et al., 2007)**, was also elevated in PMO-2. The organic mass fraction of oxalate was 9.4 % in PMO-2 compared to 2.3 % and 3.0 % in PMO-1 and PMO-3. The nitrate concentration in PMO-2 was very low compared to PMO-1 or PMO-3 (Table 1), also supporting aqueous phase processed aerosol in PMO-2. **While clearly gas phase formation of SOA cannot be excluded, several lines of evidence suggest that aqueous phase oxidation likely influenced the chemical and physical characteristics of the PMO-2 aerosol to a larger extent than those of PMO-1 and PMO-3 based on the observed** molecular characteristics, major ion concentrations (Fig. S20)**, and the model simulated transport pathways and GFS meteorology."**

We modified the abstract to note directly that we have a limited number of observations (lines 21-23): "**Although the number of observations is limited,** the results suggest that biomass burning organic aerosol injected into the free troposphere are more persistent than **organic aerosol** in the boundary layer having broader implications for aerosol aging."

The following changes were also applied to clarify the potential effect of emission sources:

- Lines 120-121: "We observed key molecular differences **pertaining to** the **extent of** oxidation **likely** related to the **combination of** transport pathways and their apparent emission sources."

- Lines 308-312: "**The North American boundary layer outflow of organic aerosol captured in PMO-2 was likely influenced by SOA (Zhang et al., 2007) and thus** is expected to have a higher **initial** O/C value compared to **pyro-convected** wildfire emissions of organic aerosol (e.g., Aiken et al., 2008; Jimenez et al., 2009; Bougiatioti et al., 2014). **Although the initial compositions are unknown,** we **anticipated** that the samples with longer transport times **(~ 1 week** for PMO-1 and PMO-3) would be **at least similar or perhaps** more oxidized than PMO-2 which had a **much** shorter transport time **(~ 3 days**)."

- Lines 366-368: "**The increased number of sulfur species observed in PMO-2 is likely associated with the anthropogenic emission sources in the North American boundary layer.**"

*Specific comments:*

*l. 13-16: Here the authors conclude that "environmental factors during transport" are responsible for the higher O/C in PMO-2. I am missing any mentioning of the possibility of different chemical composition (e.g. already higher O/C) of the PMO2 plume right at its source. Also, the term "environmental factors during transport" is too blurry.*

To clarify this, the sentence was replaced with the following: "**To better understand the difference between free tropospheric** transport and **boundary layer transport, the** meteorological conditions **along the FLEXPART simulated transport pathways were** extracted from the Global Forecast System analysis for model grids." (Lines 15-17)

*l. 20: The same issue here: the comparison between PMO1/3 and PMO2 has to be discussed with more caution. The data presented do not allow the conclusion that solely Tg/T is the cause of higher oxidized*

*aerosol in PMO2. I recommend something like "[...] and therefore less susceptible to oxidative aging than the organic aerosol transported in the boundary layer."*

We agree that PMO-1 and PMO-3 are less susceptible to oxidative aging and thank the reviewer for his suggestion of how to clarify the text. We changed the mentioned sentence which now reads (lines 19-21): "Comparisons of the $T_g$ to the ambient temperature indicated that a majority of the organic aerosol components transported in the free troposphere were more viscous and therefore **less susceptible to oxidation** than the organic aerosol components transported in the boundary layer."

*l. 28: [...] cloud droplet and ice nucleation activity [...]?*

We added the word **"activity"** to mentioned sentence and its beginning now reads (lines 28-29): "Oxidation of organic aerosol impacts its lifetime, cloud droplet and ice nucleation **activity**, …"

*l. 34-36: The biomass burning studies cited here might not be the best, since they all refer to biomass burning events from gassland fires. Grass-lignin is different from softwood (coniferous) lignin, resulting in different biomass burning marker molecules (Simoneit et al., 1993). Refering to studies on biomass burning plumes from boreal forest fires (e.g. Corrigan et al. (2013)) might be more appropriate in the context of discussing plumes from Canadian boreal forests. It is well recognized that the most prominent organic biomass burning marker (levoglucosan) undergoes fast photooxidation, however, I did not find a connection between biomass burning markers and their oxidative degradation in the paper by Vakkari et al., 2014. Here, a paper on the degradation kinetics of levoglucosan would fit much better (e.g. Lai et al. (2014), Arangio et al. (2015))*

The references illustrate that rapid oxidation occurs in both anthropogenic and biomass burning aerosol within the boundary layer. Corrigan et al., 2013 is consistent with these, so we added it (lines 35-37): "Rapid oxidation was also observed in studies of biomass burning organic aerosol in Africa (Capes et al., 2008; Vakkari et al., 2014), over the Mediterranean Sea (Bougiatioti et al., 2014**), and Hyytiälä, Finland (Corrigan et al., 2013; Vogel et al., 2013).**"

To clarify the rapid oxidation of levoglucosan the following sentence was added (lines 37-40): **"Other studies focused on the oxidation of molecular tracers such as levoglucosan have shown that they can be degraded rapidly after emission, depending on the atmospheric conditions (Lai et al., 2014; Slade et al., 2014; Arrangio et al., 2015; Bertrand et al., 2018). These studies all demonstrate** the importance of oxidation to the aging of organic aerosol and provide motivation for studies of long-range transported organic aerosol."

*l. 38: The authors might find our paper on long-range-(low-altitude)-transported biomass burning aerosol from wildfires in Russia to the SMEAR station in Hyytiälä, Finland of interest. During a pollution plume event we observed highly oxidized aerosol (O/C ~ 0.70), while the average O/C during this campaign ranged around 0.5. We also observed high-molecular weight organic matter in the aerosol phase during this BB event, indicated by molecular ion signals up to m/z 800 (Vogel et al., 2013).*

We thank the reviewer for pointing out this interesting paper, which is highly relevant for this work.

*l. 244: A low oxalate/sulfate ratio is reported for PMO-2, likely obscured by high sulfate concentrations. The authors argue that oxalate is also high in PMO-2 due to aqueous phase processing. To further search for evidence that cloud processing has really occurred for PMO-2, it would be of interest to compare the oxalate/sulfate ratio in sample PMO2 with reported oxalate/sulfate ratios in the Eastern US close to the emission sources.*

It was difficult to obtain oxalate/sulfate ratios for fresh aerosol in the Eastern US, but we did find some ratios for aerosol from Helsinki, Southern Africa, Hong Kong, and the Amazon (Zhou et al., 2015). The ratios from PMO-2 compared to the non-BB ratios from Zhou et al. noticeably exceed those reported for Helsinki, Southern Africa, and Hong Kong (Note, only BB aerosol values were reported for the Amazon sample). Based on the studies by Yu et al., (2005) and Sorooshian et al., (2007) the observed oxlate/sulfate ratio further supports our hypothesis of aqueous processing for this sample.

*l. 307: Is the North American SOA solely anthropogenic? I think there are several studies reporting dominant biogenic SOA in East America (e.g. isoprene SOA in South East US?).*

We thank the reviewer this comment and agree with the reviewer it is likely that biogenic SOA are a significant component of the aerosol. As stated above, to make this more clear we added the following text (lines 308-310): "**The** North American **boundary layer** outflow of organic aerosol **captured in PMO-2 was likely influenced by SOA (Zhang et al., 2007) and thus** is expected to have a higher **initial** O/C value compared to **pyro-convected** wildfire emissions of organic aerosol (e.g., Aiken et al., 2008; Jimenez et al., 2009; Bougiatioti et al., 2014)."

*l. 353-366: The authors describe a higher O/C in PMO2 for the CHOS group compared to PMO1, and argue at the end of the section that this observation highlights the enhanced aging during transport of PMO2. While I believe that there is indeed a higher oxygen content in the sample PMO2, it does matter to which atom the oxygen is bonded. If it appears as oxygen-sulfur bonds (as organosulfates), then the increased oxygen content (higher O/C) in PMO-2 rather tells us that the different source emissions of the PMO2 plume allowed enhanced formation of organosulfates. This observation would be in line with the observation of higher inorganic sulfate in PMO2, as well as the expectation that SO2 emissions over North East US are higher than in the remote Canadian boreal forests.*

Yes, because of the ESI analytical bias the sulfur containing formulas in our samples are likely organosulfates. We also carried this assumption through the calculation of the average oxidation state of carbon as stated in lines 374-375: "Additionally, we assumed all nitrogen and sulfur were present as nitrate and sulfate functional groups and calculated the $OS_C$ with the appropriate corrections (Equation S1)."

The presence of organosulfates indicates that reactive sulfur was present, which we described as being due in part to anthropogenic emissions (line 246). To strengthen the connection between the CHOS formulas and the emission source we added the following text (lines 366-370): **"The increased number of sulfur species observed in PMO-2 are likely associated with the anthropogenic emission sources in the North American boundary layer.** Overall, **the observed** differences in the O/C ratios between the boundary layer transported aerosol (PMO-2) compared to the free troposphere transported aerosol (PMO-1 and PMO-3) highlight differences in the aging and lifetime of aerosol relative to its transport pathway **and emission source."**

Regarding an artificially increased O/C value, we repeat our original response to this concern in Author Responses to Reviewer #1 (second paragraph on page 16): "Yes, the O/C of the CHOS species is impacted by the presence of organic sulfates. If 4 oxygen (atoms) are removed from the molecular formulas and the O/C is recalculated the O/C decreases to a level somewhat below that of the CHO group (O/C = 0.44 for PMO-2 and O/C = 0.27 for PMO-1 when "sulfate" is removed). This is why the O/C values of the CHOS compounds are not directly compared to the other (elemental) groups."

*l. 380: It is too speculative to talk about the CCN ability of the sampled aerosol particles when their size distribution is not known. Also, the speculation about the amount of less volatile components seems to me*

*ambiguous, since the total mass concentration available for gas-to-particle partitioning will also affect the fraction of higher-volatility compounds in the particle phase.*

We did not intend to imply that we knew the CCN ability of the sampled aerosol, only that based on the observed characteristics the aerosol would have a higher CCN ability. Likewise, the most abundant components were less volatile in PMO-2, as demonstrated by the volatility plots (Figs. 6, S13, S14). To make this more clear, we have made the following changes to the sentence: "Conversely, the overall higher oxidation of PMO-2 implies that the sampled aerosol was likely more hygroscopic, included more efficient cloud condensation nuclei (Massoli et al., 2010), **or** had **components of a** less volatile **nature** (Ng et al., 2011) **than PMO-1 and PMO-3**." (Lines 383-385)

*+l. 472: What is about the role of multiphase and heterogeneous oxidation of the aerosol additionally to aqueous-phase processing of cloud droplets?*

In this case, we referred to oxidation processes in general and only exemplify aqueous phase oxidation because the predicted RH during transport varied strongly between the boundary layer and the free troposphere. This implies that aqueous-phase oxidation/processing would be especially limited in the free troposphere. Note, we do not have trace gas concentrations and thus we chose to avoid speculation regarding the gas-phase oxidation process differences. As such, we decided to leave this sentence as is.

*l. 480-494: The section is missing a more critical discussion, including the fact that the initial chemical composition of the plume sampled in PMO2 is not known. Again, PMO2 is compared here against PMO1 and 3. The fact that higher sulfate is observed in PMO2 compared with PMO1/3 rather goes back to the emission source, where you expect more SO2 being emitted in North America than in the boreal forest. The relatively higher abundance of sulfate in PMO2 against PMO1/3 hence does not necessarily support cloud processing.*

To directly address the reviewers concern about not knowing the initial composition, we added the following: "**However, the exact oxidation pathways that led to the increased oxidation observed for PMO-2 and the initial composition are unclear.**" (Lines 486-487).

As for the reviewer's concern about sulfate in PMO-2 vs. PMO-1/3, we mentioned several times that the sulfate concentration is related to the emission source. Manuscript text in lines 234-235, 246, 285, and 308-310 all makes direct or oblique mention of the impact of anthropogenic emissions on the concentration of sulfate or the observed oxidation. Additionally, as many studies have noted (Crahan et al., 2004; Carlton et al., 2007; Zhou et al., 2015), the production of sulfate is strongly tied to aqueous and cloud phase processing of $SO_2$ including when the emission source of $SO_2$ is anthropogenic.

In any case (and as described above), we have made several minor changes to this section to better address the concerns about initial composition and gas phase reactions.

*l. 488-491: These lines are redundant with section 3.1 (l.235 ff.).*

We removed the following sentence: "Furthermore, nitrate is known to be scavenged during cloud processing (Dunlea et al., 2009), leading to its decrease in recently cloud processed aerosol."

*l. 508: Only aqueous-phase oxidation?*

No, it should be all oxidation, the sentence has been changed accordingly (Line 526).

[revised manuscript text omitted]